# CTDG-SSM: Continuous-time Dynamic Graph State-Space Models for Long Range Propagation

## Abstract

Continuous-time dynamic graphs (CTDGs) provide a richer framework to capture fine-grained temporal patterns in evolving relational data. Long-range information propagation is a key challenge in learning representations for CTDGs, wherein it is important to retain and update information over long temporal horizons. Existing approaches restrict models to capture one-hop or local temporal neighborhoods and fail to capture multi-hop or global structural patterns. To mitigate limitations of the current approaches, we derive the state-space modelling framework for continuous-time dynamic graphs (CTDG-SSM) from first principles. We first introduce continuous-time Topology-Aware higher order polynomial projection operator (CTT-HiPPO), a novel memory-based reformulation of HiPPO to jointly encode temporal dynamics and graph structure, where solution for memory representations from CTT-HiPPO are obtained by projecting the classical HiPPO solution through a polynomial of the Laplacian matrix, yielding topology-aware memory updates that admit an equivalent state-space formulation for CTDGs (CTDG-SSM). This is then discretized (e.g., using the zero-order hold method) for practical implementation. We further provide theoretical guarantees demonstrating the robustness of memory representations under graph structure perturbations. Across benchmarks on dynamic link prediction, dynamic node classification, and sequence classification, CTDG-SSM achieves state-of-the-art performance. Notably, it achieves large performance gains on dynamic link prediction and sequence classification tasks, specifically on datasets that require long range temporal (LRT) and spatial reasoning.[1]

## 1 Introduction

Continuous-time dynamic graphs (CTDGs) provide a principle framework for modeling evolving relational data as a continuous stream of timestamped events, with each event capturing interactions between entities at a specific time instance (Rossi et al., 2020). Unlike discrete-time dynamic graphs (DTDGs), which rely on coarser snapshot intervals (Kazemi et al., 2020), CTDGs preserve fine-grained temporal information, making them especially well-suited for tasks such as dynamic link prediction and dynamic node classification (Ding et al., 2024; Rossi et al., 2020). These capabilities have made CTDGs increasingly important in domains including finance, e-commerce, and social network analysis, to name a few. Despite initial efforts in representation learning for CTDGs, existing approaches still face two primary challenges: (1) *long-range temporal dependencies (LRT)*: the ability to preserve and use node states and interactions over extended time horizons; and (2) *long-range spatial dependencies (LRS)*: the ability to capture multi-hop structural interactions beyond immediate neighborhoods in dynamic graphs.

Based on these challenges, existing models for CTDGs can be broadly categorized into two types: *event-driven models* and *sequence-based models*. *Event-driven models* update node states at the arrival of each interaction and capture structural context through mechanisms such as temporal random walks and graph neural networks-based message passing (Wang et al., 2021b; Rossi

---

[1] Code to reproduce the results is available at: https://anonymous.4open.science/r/CTDG-SSM-7D78

et al., 2020; Xu et al., 2020). While computationally efficient, such models mainly capture short-term temporal patterns and are weak at preserving LRT (Yu et al., 2023). The second category includes *sequence-based models*, which explicitly target LRT using sequence models such as Transformer or Mamba. These methods construct temporal sequences of node features and their 1-hop temporal neighbors, patch them, and process them with either Transformer or Mamba layers (Yu et al., 2023; Ding et al., 2024). Although effective for LRT, these models inherently restrict structural context to the local neighborhood, limiting their capacity to capture LRS (Gravina et al., 2024) and global spatial patterns in dynamic graphs. Modeling LRS is particularly important in domains such as financial fraud detection, where money laundering typically spans long transaction chains rather than isolated local interactions (Altman et al., 2023).

To overcome the limitations of existing methods while ensuring both LRS and LRT, we introduce a continuous-time dynamic graph state-space model (`CTDG-SSM`)-a unified spatiotemporal state-space framework that integrates temporal memory compression through a temporal polynomial basis and graph structure through graph filters that are polynomials of the graph Laplacian. To begin with, we derive a continuous-time, topology-aware higher-order polynomial projection operator (`CTT-HiPPO`), in which time-varying node signals are expressed jointly through temporal and spatial polynomial bases. The resulting coefficients of `CTT-HiPPO` are computed by minimizing the discrepancy between the observed node features and their graph-filtered polynomial approximations, thereby extending `GHiPPO` (Li et al., 2024b) to the continuous-time dynamic graph setting. As a result, `CTT-HiPPO` captures both temporal evolution and graph-induced structural patterns, providing a principled way to construct a structure-aware state matrix for the SSM.

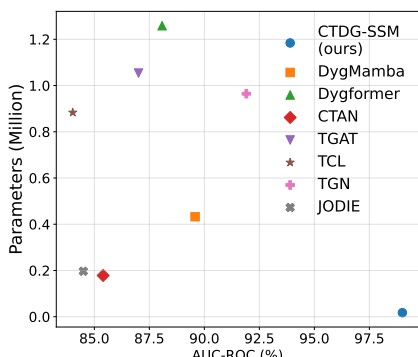

Figure 1: Efficiency of `CTDG-SSM` in terms of performance and number of learnable parameters

To implement `CTDG-SSM` efficiently, we discretize the continuous-time formulation using zero-order hold (ZOH), yielding the discrete counterpart of the model. The resulting `CTDG-SSM` remains lightweight, with only a small set of learnable parameters-primarily the coefficients of the graph polynomial filter and the system matrices governing state evolution. On the LRT task with the MOOC dataset, Fig. 1 shows that the model captures temporal patterns effectively despite its small parameter count. Using AUC-ROC and number of parameters as metrics, `CTDG-SSM` achieves top performance while using roughly one-tenth the parameters of competing methods.

**Contributions and main results.** We summarize the main contributions of the paper as follows:

- We first develop `CTT-HiPPO`, a HiPPO-based memory mechanism for CTDGs that efficiently compresses historical information from all events while maintaining LRT and LRS. Further, leveraging the relation between the classical HiPPO coefficients and the coefficients of `CTT-HiPPO`, we derive an equivalent SSM, `CTDG-SSM`, that governs the evolution of `CTT-HiPPO`.

- We derive a discrete form of `CTDG-SSM` using ZOH discretization that enables efficient implementation with diagonal parameterization for scalable and stable computation.

- We provide theoretical guarantees characterizing the robustness of `CTT-HiPPO` coefficients to graph perturbations and establish the permutation equivariance property of `CTDG-SSM`. These properties are crucial for real-world scenarios where continuous data stream collection and processing are susceptible to errors and failures.

We conduct extensive experiments to assess the ability of our model to preserve both LRT and LRS. For temporal long-range dependency, we benchmark `CTDG-SSM` on dynamic graph learning tasks such as link prediction and node classification, where it outperforms state-of-the-art methods on LRT benchmarks, including LastFM, Enron, and MOOC. To evaluate spatial long-range dependency, we conduct the sequence classification experiment (Gravina et al., 2024), demonstrating the model's capacity to capture LRS through node states generated using spatiotemporal updates.

## 2 RELATED WORKS

**Learning with DTDGs.** Learning on dynamic graphs can be broadly categorized into two subareas: learning for DTDGs and CTDGs. DTDGs represent data as a sequence of graph snapshots observed at discrete time intervals. Most learning algorithms for DTDGs extend static graph learning methods, such as graph convolutional networks (GCNs), to each snapshot and employ recurrent neural networks (RNNs) to capture temporal dependencies (Pareja et al., 2020; Chen et al., 2022). Recently, efforts have been made to extend SSMs to the DTDGs to capture LRT dependencies (Li et al., 2024b). However, it assumes a fixed graph structure within each interval and then combines node embeddings from these snapshots using GNNs. A direct extension of this approach to CTDGs is challenging, since it involves continuous graph evolution, where the set of nodes evolves over time, and edges occur at irregular intervals. Moreover, representing event streams using DTDGs rather than CTDGs inevitably leads to a loss of fine-grained temporal information (Rossi et al., 2020; Kumar et al., 2019; Trivedi et al., 2018).

**Learning with CTDGs.** CTDGs represent dynamic graphs as streams of time-stamped events. Existing learning methods typically focus on either short-range or LRT dependencies, and are based on random walks, message passing, or sequence modeling with Transformer or mamba layers. Representative approaches include temporal random walks (Nguyen et al., 2018; Starnini et al., 2012), message passing architectures such as TGAT (Xu et al., 2020), and memory-based methods such as TGN and JODIE (Rossi et al., 2020; Kumar et al., 2019). Memory-based models that rely on RNNs often suffer from gradient instability (vanishing or exploding), which limits their ability to capture long-range dependencies (Rossi et al., 2020). To address this, recent architectures such as DyGFormer and DyGmamba employ Transformers and Mamba, respectively (Yu et al., 2023; Ding et al., 2024). However, these methods pre-process temporal data by restricting attention to one-hop temporal neighborhoods before transformation, thereby limiting their ability to capture multi-hop information. In contrast, our proposed method learns node representations without imposing such structural constraints, enabling richer modeling of both temporal and spatial dependencies. Furthermore, the proposed method, primarily developed for CTDGs, can also handle DTDGs, given the equivalence between CTDGs and DTDGs (Souza et al., 2022).

## 3 CONTINUOUS-TIME DYNAMIC GRAPHS

In this section, we describe continuous-time dynamic graphs (CTDGs) and the notation used throughout the paper.

Consider a *continuous-time* observation $\mathcal{G}(t) = (u, v, t)$, which represents a temporal edge between node $u$ and $v$ at time $t$. A CTDG (Rossi et al., 2020), denoted by $\mathcal{G}$, is an ordered sequence of temporal interactions $\mathcal{G} = \{\mathcal{G}(t_1), \mathcal{G}(t_2), \ldots\}$ appearing at time instances $t_1 < t_2 < \cdots$. It should be noted that the same subset of nodes may appear in $\mathcal{G}(t_i)$ and $\mathcal{G}(t_j)$ for $i \neq j$. In what follows, we capture those unique subsets of nodes that appear within a temporal window and define their underlying graph operator.

**Active node set.** For a given time $\tau \in \mathbb{R}_+$, we define the subgraph $\mathcal{G}_\tau$ of $\mathcal{G}$ as the collection of temporal interactions that occur up to time $\tau$. Formally, $\mathcal{G}_\tau = \{\mathcal{G}(t_i) \mid t_i \leq \tau\}$. The set of *active nodes* at time $\tau$ is then the set of nodes that participate in any interaction in $\mathcal{G}_\tau$, and is denoted by $\mathcal{V}_\tau = \{u \mid u \in \mathcal{G}(t_i), t_i \leq \tau\}$. Let us denote the number of nodes in $\mathcal{V}_\tau$ by $N_\tau = |\mathcal{V}_\tau|$.

**Subgraph operator and filters.** The temporal interactions of the active nodes in $\mathcal{G}_\tau$ is captured by the subgraph adjacency matrix $\boldsymbol{A}_\tau \in \mathbb{R}^{N_\tau \times N_\tau}$ with entries $A_\tau[u, v] = \sum_{t_i \leq \tau} \mathbb{I}(\{u, v\} \in \mathcal{G}(t_i))$, where $\mathbb{I}(\cdot)$ is the indicator function defined as $\mathbb{I}(\{u, v\} \in \mathcal{G}(t_i)) = 1$ if $\{u, v\} \in \mathcal{G}(t_i)$, and 0 otherwise. We use the degree normalized Laplacian matrix defined as $\boldsymbol{L}_\tau = \boldsymbol{I} - \boldsymbol{D}_\tau^{-1/2} \boldsymbol{A}_\tau \boldsymbol{D}_\tau^{-1/2}$, where $\boldsymbol{D}_\tau$ is the corresponding degree matrix $\boldsymbol{D}_\tau = \mathrm{diag}(\boldsymbol{A}_\tau \boldsymbol{1})$.

Graph filters are expressed as matrix polynomials of the normalized Laplacian matrix. We define a $K$th-order filter as $p(\boldsymbol{L}_\tau) = \sum_{k=0}^{K-1} \alpha_k \boldsymbol{L}_\tau^k$, where $\{\alpha_k\}_{k=1}^{K-1}$ are learnable filter coefficients. Applying a $K$th-order filter aggregates information from up to $K$-hop neighborhoods in the subgraph $\mathcal{G}_\tau$. Specifically, as $\tau$ evolves continuously with time in CTDGs, both $\boldsymbol{A}_\tau$ and $\boldsymbol{L}_\tau$ evolve sequentially, and thus the corresponding filters $p(\boldsymbol{L}_\tau)$ adapt to the temporal evolution of the graph structure.

Each node $u$ in the subgraph $\mathcal{G}_\tau$ is associated with a feature vector $\boldsymbol{x}_u(t) \in \mathbb{R}^{D_n}$. Collecting the node features over the subgraph yields the graph-level feature matrix $\boldsymbol{X} \in \mathbb{R}^{N_\tau \times D_n}$.

# 4 THE PROPOSED STATE-SPACE MODELS FOR CTDGS

In this section, we develop SSMs for CTDGs, with the objective of compressing historical event information into compact latent memory representations. We first present a HiPPO matrix (Gu et al., 2020) computation that incorporates graph structure as an inductive bias within latent memory representations. Specifically, we decompose node signals as graph-aware transformations of signals represented in an orthogonal polynomial space. Subsequently, we develop a novel SSM model for CTDG and derive its discrete counterpart, which is useful for practical implementation.

To begin with, we describe the HiPPO projection for graph data, drawing inspiration from (Li et al., 2024b). Let us define an orthogonal polynomial $\boldsymbol{g}(t) \in \mathbb{R}^{d \times 1}$ and a coefficient matrix $\boldsymbol{H}_{i,\tau} \in \mathbb{R}^{N_\tau \times d}$. We then model the $i^{\text{th}}$ features $\boldsymbol{X}[:,i](t) \in \mathbb{R}^{N_\tau \times 1}$ on $\mathcal{V}_\tau$ as

$$\boldsymbol{X}[:,i](t) = p(\boldsymbol{L}_\tau)\boldsymbol{H}_{i,\tau}\boldsymbol{g}(t) + \boldsymbol{r}_i(t), \quad \forall t < \tau, \tag{1}$$

for $i = 1, \cdots, D_n$, where $p(\boldsymbol{L}_\tau)$ is the polynomial of the normalized Laplacian and the error $\boldsymbol{r}_i(t) \in \mathbb{R}^{N_\tau \times 1}$ accounts for any model mismatch. Here, the graph filter $p(\boldsymbol{L}_\tau)$ incorporates the topology structure in $\boldsymbol{H}_{i,\tau}$ by aggregating the HiPPO coefficients based on the temporal graph-structure in $\mathcal{G}_\tau$.

Then the coefficients $\boldsymbol{H}_{i,\tau}$ are obtained by minimizing the residual in the temporal window $[0, \tau]$ as

$$\min_{\boldsymbol{H}_{i,\tau}} \int_0^\tau \|\boldsymbol{X}[:,i](t) - p(\boldsymbol{L}_\tau)\boldsymbol{H}_{i,\tau}\boldsymbol{g}(t)\|_2^2 \, d\mu(t), \tag{2}$$

where $\mu(t)$ is the measure under which the orthogonality of $\boldsymbol{g}(t)$ is defined. Although the above formulation provides a general framework with a learnable $K$th-order graph filter for modeling the HiPPO coefficients for graph data, it is related to the one in (Li et al., 2024b) that instead uses a quadratic Laplacian regularizer in equation 2 with $p(\boldsymbol{L}_\tau) = \boldsymbol{I}$, whereas the classical HiPPO formulation (without any graph structure) Gu et al. (2020) uses $p(\boldsymbol{L}_\tau) = \boldsymbol{I}$ in equation 2.

Now, to find the optimal set of coefficients $\boldsymbol{H}_{i,\tau}$, we use the first-order optimality condition (detailed derivation can be found in Appendix B) to obtain

$$p(\boldsymbol{L}_\tau)\boldsymbol{H}_{i,\tau} = \int_0^\tau \boldsymbol{X}[:,i](t)\boldsymbol{g}(t)^\top \, d\mu(t) = \boldsymbol{H}_{i,\tau}^{\text{(HiPPO)}} \tag{3}$$

$$\boldsymbol{H}_{i,\tau} = p(\boldsymbol{L}_\tau)^{-1}\boldsymbol{H}_{i,\tau}^{\text{(HiPPO)}} \tag{4}$$

where $\boldsymbol{H}_{i,\tau}^{\text{(HiPPO)}}$ denotes the solution to the classical HiPPO formulation without any graph structure (Gu et al., 2020), and by the choice of $\boldsymbol{L}_\tau$, $p(\boldsymbol{L}_\tau)^{-1}$ is well-defined. From equation 3, it can be seen that the `CTT-HiPPO` coefficients $\boldsymbol{H}_{i,\tau}$ are essentially the graph-aware extension of the classical HiPPO coefficients, obtained by projecting $\boldsymbol{H}_{i,\tau}^{\text{(HiPPO)}}$ through the inverse polynomial graph filter. Although we provide the solution $\boldsymbol{H}_{i,\tau}$ for a single feature $i$, it can be easily extended to multiple features along the lines as above. Henceforth, for brevity, we drop the subscript $i$ in $\boldsymbol{H}_{i,\tau}$ and $\boldsymbol{X}[:,i](t)$ and simply use $\boldsymbol{H}_\tau$ and $\boldsymbol{X}(t)$.

## 4.1 THE CTDG STATE-SPACE MODEL

We now present the main result of the paper, i.e., the state-space formulation that governs the evolution of the memory coefficients $\boldsymbol{H}_\tau$. In SSMs, the temporal dynamics of an input signal are modeled through the progression of latent memory representations (state space vectors). We now describe the evolution of the representations of CTDGs over time through the evolution of the memory coefficient matrix $\boldsymbol{H}_\tau$, which jointly captures both temporal and topological structures. We refer to the proposed SSM for CTDG as `CTDG-SSM`, whose model is described in the next theorem.

**Theorem 4.1** (`CTDG-SSM`). *Consider a interval $s \in [\tau, \tau_+)$ with CTDGs $\mathcal{G}_\tau$ and $\mathcal{G}_{\tau_+}$. Let $\mathcal{G}_\tau$ denote a CTDG at time $\tau$, and for new a observation $\mathcal{G}(\tau_+)$ with corresponding CTDG $\mathcal{G}_{\tau_+}$. The evolution of the memory coefficients $\boldsymbol{H}_s$ for $s \in [\tau, \tau_+)$ admits the following state-space representation:*

$$\frac{d\boldsymbol{H}_s}{ds} = -\boldsymbol{H}_s\frac{\boldsymbol{A}^\top}{M(s)} - p(\boldsymbol{L}_s)^{-1}\frac{dp(\boldsymbol{L}_s)}{ds}\boldsymbol{H}_s + p(\boldsymbol{L}_s)^{-1}\boldsymbol{X}(s)\frac{\boldsymbol{B}^\top}{M(s)}, \tag{5}$$

*where $\boldsymbol{A} \in \mathbb{R}^{d \times d}$ is the state-transition matrix that depends on the choice of the orthogonal polynomial $\boldsymbol{g}(\cdot)$, $\boldsymbol{B} \in \mathbb{R}^{d \times 1}$ is the input matrix, and $M(s) : \mathbb{R}_+ \to \mathbb{R}_+$ is a normalization term that depends on the choice of the measure $\mu(t)$. Here, $\boldsymbol{L}_s \in \mathbb{R}^{N_{\tau_+} \times N_{\tau_+}}$, $\boldsymbol{X}(s) \in \mathbb{R}^{N_{\tau_+} \times 1}$, $p(\boldsymbol{L}_s) = \frac{\tau - s}{\tau - \tau_+} p(\boldsymbol{L}_{\tau_+}) + \frac{\tau_+ - s}{\tau_+ - \tau} p(\boldsymbol{L}_\tau)$, and $\boldsymbol{H}_s \in \mathbb{R}^{N_{\tau_+} \times d}$ for $s \in [\tau, \tau_+)$.* [2]

The proof of this theorem is relegated to Appendix C.1. The result directly follows from the equivalence between the classical HiPPO coefficients and a linear ODE (Theorem 1 in (Gu et al., 2020)) characterized by the state matrix $\boldsymbol{A}$ and input matrix $\boldsymbol{B}$, and more importantly, incorporating the fact that $\boldsymbol{L}_\tau$ depends on $\tau$ in CTDGs. We end this subsection with the following remark that explicitly connects CTDG-SSM to (Gu et al., 2020) and (Li et al., 2024b).

**Remark.** *Equation 5 shows that the graph filter $p(\boldsymbol{L}_\tau)$ modifies the classical HiPPO dynamics by introducing time-dependent graph-aware terms that account for the change in temporal evolution of the graph. When the polynomial of Laplacian is static or fixed as in (Li et al., 2024b), we have $\frac{dp(\boldsymbol{L}_s)}{ds} = 0$. Thus, CTDG-SSM reduces to the SSM variant in (Li et al., 2024b).*

*When there is no graph, i.e., $p(\boldsymbol{L}_\tau) = \boldsymbol{I}$, CTDG-SSM reduces exactly to classical SSM (Gu et al., 2020).*

### 4.2 THE DISCRETE VERSION OF CTDG-SSM

We now describe the discrete-time version of CTDG-SSM in this section. In particular, we discretize CTDG-SSM using the ZOH approach.

In practice, the continuous-time SSM is discretized using ZOH, which assumes piecewise constant inputs, i.e., $\boldsymbol{X}(t) = \boldsymbol{X}[k]$ for $t \in [t_{k-1}, t_k)$ with step size $\Delta[k] = t_k - t_{k-1}$. Here $t_{k-1}$ is $\tau$ and $t_k$ is $\tau_+$. Where $\boldsymbol{L}[k] \in \mathbb{R}^{N_{\tau_+} \times N_{\tau_+}}$ is obtained using a subgraph $\mathcal{G}_{\tau_+}$ and $\boldsymbol{L}[k-1] \in \mathbb{R}^{N_{\tau_+} \times N_{\tau_+}}$ is obtained by removing the newly observed edges in $\Delta[k]$ time interval from $\boldsymbol{L}[k]$.

**Theorem 4.2** (*Discrete CTDG-SSM*). *Let $\boldsymbol{X}[k]$ denote the input at time $t_k$, and let the temporal graph structures at times $t_k$ and $t_{k-1}$ be represented by the Laplacians $\boldsymbol{L}[k]$ and $\boldsymbol{L}[k-1]$, respectively. Then for $\Delta[k] = t_k - t_{k-1}$, the memory update of the proposed CTDG-SSM model is governed by the following discrete-time recursion:*

$$\boldsymbol{H}[k+1] = \bar{\boldsymbol{A}}_{\boldsymbol{L}[k]} \, \boldsymbol{H}[k] \, \bar{\boldsymbol{A}} + \bar{\boldsymbol{B}}(\boldsymbol{L}[k], \boldsymbol{X}[k]) \tag{6}$$

*Here, $\bar{\boldsymbol{A}}_{\boldsymbol{L}[k]} = \exp\left(-p(\boldsymbol{L}[k])^{-1} \left(p(\boldsymbol{L}[k]) - p(\boldsymbol{L}[k-1])\right)\right)$, $\bar{\boldsymbol{A}} = \exp(\Delta[k]\boldsymbol{A}^\top)$, and $\bar{\boldsymbol{B}}(\boldsymbol{L}[k], \boldsymbol{X}[k]) = \int_0^1 (\bar{\boldsymbol{A}}_{\boldsymbol{L}[k]})^s \, p(\boldsymbol{L}[k])^{-1} \boldsymbol{X}[k] \boldsymbol{B}^\top (\bar{\boldsymbol{A}})^s \, \Delta[k] \, ds$.*

We present the detailed proof in Appendix C.2. The proof proceeds by first simplifying Equation equation 5 to standard state-space form with system and input matrices, leveraging the properties of the Kronecker structure. We then apply the ZOH discretization to this form and subsequently factorize the discretized equations to obtain the final expression.

The discrete memory update in equation 6 is structurally analogous to the vanilla mamba update (Gu & Dao, 2024), with two key distinctions: there are two state-transition matrices that jointly operate on the state variable, and the input-dependent component $\bar{\boldsymbol{B}}(\boldsymbol{L}[k], \boldsymbol{X}[k])$ does not admit a closed-form solution.

**Remark.** *The invertibility of $p(\boldsymbol{L}[k])$ matrix involved in equation 6 is ensured by choosing all graph filter coefficients to be strictly positive i.e., $\alpha_i > 0 \, \forall i = 0, 1, \ldots, K - 1$. Since $\boldsymbol{L}[k]$ is a normalized graph Laplacian, the spectrum satisfies $\lambda[k] \in [0, 2]$. Therefore with $\alpha_i > 0$ we have $p(\lambda[k]) > 0$, which implies that $p(\boldsymbol{L}[k])$ is positive definite and therefore invertible. It is more important to notice that complexity of this operation is influenced by the batch size as it directly influences the number of the active nodes (more details in the Section 5).*

## 5 ARCHITECTURE

In this section, we introduce the proposed architecture that implements discrete CTDG-SSM. The overall modular design is illustrated in Fig. 2. It mainly consists of three blocks: (a) Subgraph sam-

---

[2]To match the dimension of $\boldsymbol{L}_\tau$ and $\boldsymbol{L}_{\tau_+}$ in equation 5, we construct $\boldsymbol{L}_\tau$ by removing the edges observed in $\mathcal{G}(\tau+)$ from $\boldsymbol{L}_{\tau_+}$.

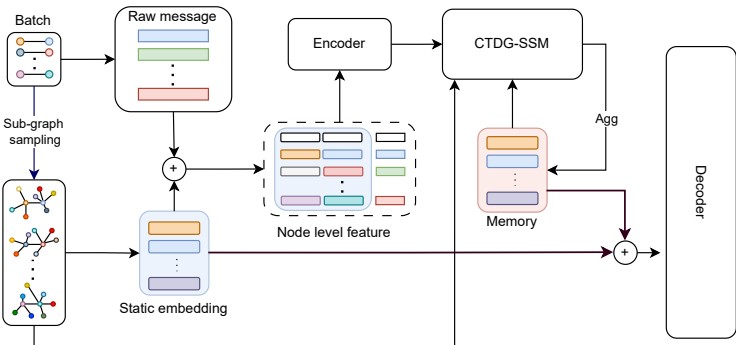

Figure 2: Architecture of the `CTDG-SSM` framework. A batch of events is subgraph-sampled to produce the batch graph. Raw messages and static embeddings are combined to form node-level features, which are encoded and processed by the CTDG-SSM module to update dynamic memory. The updated memory and static embeddings are then aggregated to form the final node representations used by the decoder.

pler: constructs $N_u$-temporal neighborhoods for each node. (b) Node feature encoder: integrates node, edge, and temporal information into node feature representations. (c) `CTDG-SSM` module: generates memory representations that capture LRT dependencies and structural context.

**Subgraph sampling.** At each training step, we construct a mini-batch of temporal interactions by grouping together $B$ chronologically consecutive events. From this batch, we develop a batch level Laplacian $\boldsymbol{L}_B[k] \in \mathbb{R}^{N_B \times N_B}$ by generating subgraphs via a neighborhood-based sampling strategy: for every node participating in an event, we sample up to $N_u$ of its most recent neighbors, where $N_u$ defines the spatial context size. To estimate $\boldsymbol{L}_B[k-1]$ we remove the current batch interaction edges from $\boldsymbol{L}_B[k]$ while preserving the neighborhood edges of the $N_u$ neighbors. This subgraph-based approach is motivated by two factors: (i) it captures information from the multi-hop temporal neighborhood, and (ii) it enables the model to update states for nodes beyond those directly involved in the observed interactions, thereby incorporating both local structural dependencies and broader temporal context.

**Node feature encoder.** We construct input features $\boldsymbol{X}_B \in \mathbb{R}^{N_B \times D_B}$ for the current batch by concatenating node-specific features, temporal neighbor features, edge attributes, and the corresponding timestamp information of events in the batch. For an interaction event $\mathcal{G}(t_i) = (u, v, t_i)$ with edge feature $\boldsymbol{x}_{uv}$, the feature vectors for the participating nodes are defined as $\boldsymbol{X}_B[u,:] = [\boldsymbol{x}_u(t_i) || \boldsymbol{x}_v(t_i) || \boldsymbol{x}_{u,v} || \phi(\Delta t_i)]$, and $\boldsymbol{X}_B[v,:] = [\boldsymbol{x}_u(t_i) || \boldsymbol{x}_v(t_i) || \boldsymbol{x}_{u,v} || \phi(\Delta t_i)]$. Here, $\boldsymbol{x}_u$ and $\boldsymbol{x}_v$ denote the static embeddings of nodes $u$ and $v$ concatenated with their raw features, and $\phi(\cdot)$ denotes a fixed (non-trainable) time-encoding function. The term $\Delta t_i$ corresponds to the inter-event time since the last occurrence of $(u, v)$; for first-time interactions, $\Delta t_i$ is assigned a large constant following prior works (Ding et al., 2024).

**Encoder.** The encoder $h_\theta$ takes the input feature matrix $\boldsymbol{X}_B$ and projects it into a latent space of $d$-dimension. These projected features are then used to update the memory representation through the `CTDG-SSM` recurrence. In experimentation, we implement the encoder as a 2-layer neural network and represent augmented and projected node features as $h_\theta(\boldsymbol{X}_B) = \tilde{\boldsymbol{X}}[k] \in \mathbb{R}^{N_B \times d}$.

**Learnable CTDG-SSMs.:** The `CTDG-SSM` block computes node memory representations according to equation 6. While a single-layer `CTDG-SSM` is sufficient to capture linear state-space dynamics, stacking multiple layers enables the model to learn richer temporal feature transformations. To enhance representational capacity, we incorporate residual connections, `RMSNorm` normalization, and the `GeLU` activation within our `CTDG-SSM` architecture, following design principles from mamba (Gu & Dao, 2024) (see Figure 3).

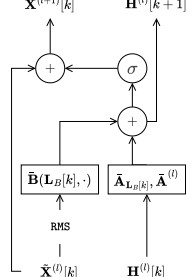

Figure 3: Illustration of single layer `CTDG-SSM`

Therefore, given the output of $(l-1)$-th layer denoted as $\tilde{\boldsymbol{X}}^{(l)}[k]$, the $l$-th layer performs the following sequence of operations: $\boldsymbol{H}^{(l)}[k+1] = \texttt{CTDG-SSM}(\text{RMS}(\tilde{\boldsymbol{X}}^{(l)}[k]), \boldsymbol{L}_B[k])$, and $\tilde{\boldsymbol{X}}^{(l+1)}[k] =$

$\tilde{\boldsymbol{X}}^{(l)}[k] + \sigma\big(\boldsymbol{H}^{(l)}[k+1]\big)$, where $\mathrm{RMS}(\cdot)$ denotes RMS normalization and $\sigma$ is a nonlinear activation function. We use the `GeLU` activation, which promotes stable training and ensures well-scaled feature transformations. The input for the first layer, i.e, $\tilde{\boldsymbol{X}}^{(0)}[k] = \tilde{\boldsymbol{X}}[k]$, is the projected node features. For nodes participating in multiple events within the same batch, we apply a *mean* aggregator to obtain a single consolidated representation.

**Memory**: The memory module maintains the latent representations of all nodes. These are initialized as zero vectors of dimension $d$. After each batch, the memory is updated with the newly computed representations of the nodes involved in the current interactions and the sampled nodes.

**Decoder.** For downstream tasks such as link prediction and node classification, the decoder operates on the memory representations of the target nodes.

*Link Prediction.* Given a query of the form $(u, v, T)$, we first retrieve the static embeddings and dynamic memory states of nodes $u$ and $v$, denoted $\boldsymbol{h}_u$ and $\boldsymbol{h}_v$. This representation is augmented with a learnable temporal embedding $\psi(\Delta t)$, where $\Delta t = T - t_{\text{last}}$ and $t_{\text{last}}$ denotes the most recent interaction time between $u$ and $v$. The concatenated vector $\big[\,\boldsymbol{h}_u \,\|\, \boldsymbol{h}_v \,\|\, \psi(\Delta t)\,\big]$ is then passed through a linear layer to produce an edge score.

*Node Classification.* For a query of the form $(u, v, T)$ or $(u, T)$, only the representation of node $u$ is used. The decoder applies a linear mapping to $\boldsymbol{h}_u$, optionally concatenated with available temporal information, to produce a multi-class probability vector corresponding to the predicted node label.

## 6 THEORETICAL CHARACTERIZATION

In this section, we derive the robustness and permutation equivariance properties of `CTDG-SSM`. In particular, robustness property characterizes the stability of memory representations under structural perturbations and is crucial given that real-world temporal graphs may include spurious edges.

**Theorem 6.1** (*Robustness property*). *Let $\bar{\boldsymbol{L}} = \boldsymbol{L} + \Delta\boldsymbol{L}$ be the perturbed graph Laplacian with $\|\Delta\boldsymbol{L}\|_2 \leq \epsilon$. Then the error between the perturbed and true coefficients is bounded linearly in terms of the energy of the perturbed graph Laplacian as $\frac{\|\hat{\boldsymbol{H}}_{i,\tau} - \boldsymbol{H}_{i,\tau}\|_2}{\|\boldsymbol{H}_{i,\tau}\|_2} \leq \epsilon\Gamma$, where $\Gamma = \frac{\lambda_2\lambda_c}{\lambda_1^2}$ with $\lambda_1 := \min_{y \in [0,2]} |p(y)| > 0$, $\lambda_2 := \max_{y \in [0,2]} |p(y)|$, and $\lambda_c := \max_{y \in [0,2]} |\frac{dp(y)}{dy}|$.*

We relegate the proof to Appendix C.3. The derivations follows by using the triangle inequality and exploiting spectral bounds of the normalized graph Laplacian. The derived error bound shows that the deviation between the perturbed and true coefficients scales linearly with the energy of the perturbed Laplacian $\Delta\boldsymbol{L}$. In other words, this implies that small structural perturbations in the underlying graph induce only proportionally small deviations in the coefficients. Hence, the representations produced by `CTT-HiPPO` are stable and robust with respect to perturbations.

**Theorem 6.2** (*Permutation Equivariance*). *Let $\mathcal{P} = \{\,\boldsymbol{\Pi} \in \{0,1\}^{N_\tau \times N_\tau} : \boldsymbol{\Pi}^\top\boldsymbol{\Pi} = \boldsymbol{\Pi}\boldsymbol{\Pi}^\top = \boldsymbol{I}_{N_\tau}\}$ be the set of all $N_\tau \times N_\tau$ permutation matrices. Then under the permutation of the graph Laplacian $\boldsymbol{L}[k]$ and node-features $\boldsymbol{X}$ by any $\boldsymbol{\Pi} \in \mathcal{P}$, the representations from `CTDG-SSM` also modifies as $\bar{\boldsymbol{H}}[k+1] = \boldsymbol{\Pi}\boldsymbol{H}[k+1]$.*

We relegate the proof to the Appendix C.4. The permutation equivariance property guarantees that, when the nodes in the observed CTDGs and their associated signals are permuted, the representations by `CTDG-SSM` permute in exactly the same way, thereby preserving equivariance.

## 7 NUMERICAL EXPERIMENTS

We evaluate the proposed algorithm on two downstream temporal graph learning tasks, namely dynamic link prediction and node classification. Further, to assess the model's ability to preserve long-range information, we test it on a sequence classification task.

**Baseline models.** For all the three tasks, we compare the performance of our model against the following state of the art algorithms, namely, `JODIE` (Kumar et al., 2019), `DyRep` (Trivedi et al., 2018), `TGN` (Rossi et al., 2020), `TGAT` (Xu et al., 2020), `GraphMixer` (Cong et al., 2023), `DyGFormer` (Yu et al., 2023), `CTAN` (Gravina et al., 2024), `DyGmamba` (Ding et al., 2024). For dy-

Table 1: AUC-ROC of dynamic link prediction with random negative sampling under T: Transductive, and I: Inductive setup. Best-performing model per dataset is shown in bold.

| Setup | Datasets | JODIE | DyRep | TGAT | TGN | CAWN | TCL | GraphMixer | DyGFormer | CTAN | DyGmamba | CTDG-SSM |
|---|---|---|---|---|---|---|---|---|---|---|---|---|
| T | LastFM | 70.89 ± 1.97 | 71.40 ± 2.12 | 71.47 ± 0.14 | 76.64 ± 4.66 | 85.92 ± 0.16 | 71.09 ± 1.48 | 73.51 ± 0.14 | 93.03 ± 0.11 | 85.12 ± 0.77 | 93.31 ± 0.18 | **93.79 ± 0.22** |
| | Enron | 87.77 ± 2.43 | 83.09 ± 2.20 | 68.57 ± 1.46 | 88.72 ± 0.95 | 90.34 ± 0.23 | 83.33 ± 0.93 | 84.16 ± 0.34 | 93.20 ± 0.12 | 87.09 ± 1.51 | 93.34 ± 0.23 | **94.98 ± 2.92** |
| | MOOC | 84.50 ± 0.60 | 84.50 ± 0.87 | 87.01 ± 0.16 | 91.91 ± 0.82 | 80.48 ± 0.41 | 84.02 ± 0.59 | 84.04 ± 0.12 | 88.08 ± 0.50 | 85.40 ± 2.67 | 89.58 ± 0.12 | **99.00 ± 0.33** |
| | Reddit | 98.29 ± 0.05 | 98.13 ± 0.04 | 98.50 ± 0.01 | 98.61 ± 0.05 | 99.02 ± 0.00 | 97.67 ± 0.01 | 97.17 ± 0.02 | 99.15 ± 0.01 | 97.24 ± 0.75 | 99.27 ± 0.01 | **99.48 ± 0.02** |
| | Wikipedia | 96.36 ± 0.14 | 94.43 ± 0.32 | 96.60 ± 0.07 | 98.37 ± 0.10 | 98.54 ± 0.01 | 97.27 ± 0.06 | 96.89 ± 0.04 | 98.92 ± 0.03 | 97.00 ± 0.21 | 99.08 ± 0.02 | **99.33 ± 0.08** |
| | UCI | 90.35 ± 0.51 | 69.46 ± 2.66 | 78.76 ± 1.10 | 92.03 ± 0.69 | 93.81 ± 0.23 | 85.49 ± 0.82 | 91.62 ± 0.52 | 94.45 ± 0.22 | 76.25 ± 2.83 | **94.77 ± 0.18** | 89.24 ± 0.43 |
| | Social Evo. | 92.13 ± 0.20 | 90.37 ± 0.52 | 94.93 ± 0.06 | 95.31 ± 0.27 | 87.34 ± 0.10 | 95.45 ± 0.21 | 95.21 ± 0.07 | 96.25 ± 0.04 | Timeout | 96.38 ± 0.02 | **99.10 ± 0.49** |
| | **Avg. Rank** | 7.93 | 9.36 | 7.86 | 4.57 | 5.71 | 8.00 | 7.71 | 3.00 | 7.50 | 2.00 | **1.86** |
| I | LastFM | 83.13 ± 1.19 | 83.47 ± 1.06 | 78.40 ± 0.30 | 81.18 ± 3.27 | 89.33 ± 0.06 | 81.38 ± 1.53 | 82.07 ± 0.31 | 94.17 ± 0.10 | 60.40 ± 3.01 | 94.42 ± 0.21 | **94.49 ± 0.27** |
| | Enron | 78.97 ± 1.59 | 73.97 ± 3.00 | 66.67 ± 1.07 | 78.76 ± 1.69 | 86.30 ± 0.56 | 82.61 ± 0.61 | 75.55 ± 0.81 | 89.62 ± 0.27 | 74.61 ± 1.64 | 89.67 ± 0.27 | **93.66 ± 4.67** |
| | MOOC | 80.57 ± 0.52 | 80.50 ± 0.68 | 85.28 ± 0.30 | 88.01 ± 1.48 | 81.32 ± 0.42 | 82.28 ± 0.99 | 81.38 ± 0.17 | 87.05 ± 0.51 | 64.99 ± 2.24 | 88.64 ± 0.08 | **98.67 ± 0.46** |
| | Reddit | 96.43 ± 0.16 | 95.89 ± 0.26 | 97.13 ± 0.04 | 97.41 ± 0.12 | 98.62 ± 0.01 | 95.01 ± 0.10 | 95.24 ± 0.08 | 98.83 ± 0.02 | 80.07 ± 2.53 | 98.97 ± 0.01 | **99.13 ± 0.03** |
| | Wikipedia | 94.91 ± 0.32 | 92.21 ± 0.29 | 96.26 ± 0.12 | 97.81 ± 0.18 | 98.27 ± 0.02 | 97.48 ± 0.06 | 96.61 ± 0.04 | 98.58 ± 0.01 | 93.58 ± 0.65 | 98.77 ± 0.03 | **99.06 ± 0.10** |
| | UCI | 79.73 ± 1.48 | 58.39 ± 2.38 | 79.10 ± 0.49 | 87.81 ± 1.32 | 92.61 ± 0.35 | 84.19 ± 1.37 | 91.17 ± 0.29 | 94.45 ± 0.13 | 49.78 ± 5.02 | **94.76 ± 0.19** | 87.43 ± 0.79 |
| | Social Evo. | 91.72 ± 0.66 | 89.10 ± 1.90 | 91.47 ± 0.10 | 90.74 ± 1.40 | 79.83 ± 0.14 | 92.51 ± 0.11 | 91.89 ± 0.05 | 93.05 ± 0.10 | Timeout | 93.13 ± 0.05 | **98.60 ± 0.14** |
| | **Avg. Rank** | 7.29 | 9.00 | 8.00 | 6.00 | 5.29 | 6.57 | 6.71 | 3.00 | 10.57 | 1.86 | **1.71** |

Table 2: Performance comparison on the dynamic node classification task with AUC-ROC as a metric.

| Dataset | JODIE | DyRep | TGAT | TGN | CAWN | TCL | GraphMixer | DyGFormer | CTAN | DyGmamba | CTDG-SSM |
|---|---|---|---|---|---|---|---|---|---|---|---|
| Wikipedia | 88.10 ± 1.57 | 87.41 ± 1.94 | 83.42 ± 2.92 | 85.51 ± 3.28 | 84.59 ± 1.16 | 79.03 ± 1.18 | 85.60 ± 1.73 | 86.35 ± 2.19 | 87.38 ± 0.14 | 87.44 ± 0.82 | **88.61 ± 0.64** |
| Reddit | 59.53 ± 3.18 | 63.12 ± 0.51 | 69.31 ± 2.18 | 63.21 ± 3.00 | 65.22 ± 0.79 | 68.04 ± 2.00 | 64.42 ± 1.15 | 67.67 ± 1.39 | 67.29 ± 0.15 | 67.70 ± 1.32 | **69.50 ± 0.82** |
| **Avg. Rank** | 7.14 | 8.86 | 7.14 | 3.86 | 4.86 | 7.29 | 7.14 | 2.14 | 7.29 | 1.14 | **1.00** |

namic link prediction and node classification tasks, we also consider models `Edgebank` (Poursafaei et al., 2022), `CAWN` (Wang et al., 2021b), and `TCL`(Wang et al., 2021a) for comparison.

## 7.1 DYNAMIC LINK PREDICTION

In this section, we present results on dynamic link prediction where the task is to predict the existence of an edge between two nodes at a given time. We evaluate the proposed algorithm in both transductive (test nodes are observed during training) and inductive (test nodes are unseen during training) settings, under different sampling strategies (random, historical, and inductive) for generating negative samples. Experiments are performed on benchmark temporal link prediction datasets (Poursafaei et al., 2022) details are provided in Appendix D.1.

**Results.** In Table 1, we present the results with AUC-ROC as a metric calculated for 5 independent trials on transductive and inductive settings with random negative sampling (more experiments with different metrics and different sampling criteria are relegated to Appendix D.2). It can be seen that on LRT benchmarks such as LastFM, MOOC, and Enron, our method consistently outperforms state-of-the-art baselines due to the model's ability in jointly encoding structural information via graph polynomials that capture multi-hop neighborhood interactions and temporal evolution through a state-space formulation. Further, importantly `CTDG-SSM` exhibits only a minor performance drop in inductive setting, highlighting its ability to effectively capture global structural and temporal patterns instead of learning local structural patterns.

## 7.2 DYNAMIC NODE CLASSIFICATION

For dynamic node classification, the goal is to predict the class label of nodes participating in an interaction $\mathcal{G}(T)$ at time $T$. We evaluate our model on the Wikipedia and Reddit datasets with 2 classes. We follow the dataset splits and preprocessing strategy outlined in Yu et al. (2023). The model is trained for 200 epochs with early stopping, and memory representations are updated as described in Section 5. During testing, we combine the memory states with static embeddings and temporal encodings, which are then passed through an MLP decoder for classification.

In Table 2, we report the mean AUC-ROC over 5 independent runs. The results demonstrate that `CTDG-SSM` consistently outperforms state-of-the-art approaches, highlighting the effectiveness of jointly capturing LRS and LRT dependencies.

## 7.3 SEQUENCE CLASSIFICATION

In this section, we present results on the sequence classification task, primarily designed to test the model's ability to capture LRS and LRT (Gravina et al., 2024) dependency. The task involves predicting the label of the initial node after traversing a long path, where each new node was connected

Table 3: Performance comparison on the sequence classification task. Best-performing model is shown in bold; second-best is underlined.

| | $n = 3$ | $n = 9$ | $n = 15$ | $n = 20$ | Avg. Rank |
|---|---|---|---|---|---|
| DyRep | 100.0±0.0 | 47.93±2.73 | 48.60±2.48 | 50.47±2.88 | 7.25 |
| GraphMixer | 100.0±0.0 | 52.80±5.56 | 52.49±15.36 | 52.04±8.20 | 6.75 |
| JODIE | $100.0 \pm 0.0$ | $\mathbf{100.0 \pm 0.0}$ | 60.0±14.91 | 50.87±2.46 | 3.75 |
| TGAT | 100.0±0.0 | 47.87±2.72 | 50.53±2.15 | 49.07±1.55 | 7.50 |
| TGN | 100.0±0.0 | 48.13±1.63 | 48.67±2.76 | 50.13±2.17 | 7.00 |
| CTAN | 100.0±0.0 | $99.93 \pm 0.21$ | 93.47±8.78 | 88.93±12.06 | 3.25 |
| TU-SSM | 47.0±1.12 | $\underline{50.73 \pm 1.74}$ | $52.26 \pm 2.44$ | $54.46 \pm 0.73$ | 8.00 |
| DyGFormer | 100.0±0.0 | 53.02±6.06 | 42.80±16.25 | 42.79±19.62 | 9.25 |
| DyGmamba | 100.0±0.0 | 54.01±6.06 | 45.60±12.25 | 45.29±17.62 | 8.25 |
| CTDG-SSM (FO) | 100.0±0.0 | $97.06 \pm 0.44$ | $\underline{97.40 \pm 0.20}$ | $\underline{97.13 \pm 0.89}$ | $\underline{2.75}$ |
| CTDG-SSM (SO) | 100.0±0.0 | $98.13 \pm 0.58$ | $\mathbf{97.80 \pm 0.58}$ | $\mathbf{98.60 \pm 0.29}$ | $\mathbf{2.25}$ |

to the node from the previous event, as illustrated in Fig. 4. We generate the data using the procedure in (Gravina et al., 2024).

In this experiment, to depict the importance of aggregating the information from one-hop and multi-hop and also to see the importance of aggregating the information using the structural change term we present three variants `CTDG-SSM (FO)`, employs a first-order polynomial filter of the form $\boldsymbol{I} + \alpha_1 \boldsymbol{L}_\tau$, `CTDG-SSM (SO)`, uses a second-order polynomial filter defined as $p(\boldsymbol{L}_\tau) = \boldsymbol{I} + \alpha_1 \boldsymbol{L}_\tau + \alpha_2 \boldsymbol{L}_\tau^2$ and Topology unaware SSM (`TU-SSM`) fixes the spatial system matrix $\boldsymbol{A}_{\boldsymbol{L}[k]}$ to the identity, thereby isolating its contribution in learning structural patterns.

**Results.** Table 3 reports results for the sequence classification task, where prediction accuracy is defined as the ratio of correctly classified sequences to the total number of sequences. We observe that removing the structural update term in the memory update (`TU-SSM`) leads to a substantial drop in performance, underscoring the importance of modeling the time-varying graph structure in CTDGs. Furthermore, incorporating higher-order polynomials for multi-hop aggregation in `CTDG-SSM (SO)` yields clear gains over the single-order variant, which primarily captures local patterns. Finally, the proposed method achieves significant improvements over state-of-the-art baselines, particularly on longer sequence lengths, highlighting its effectiveness in capturing LRS.

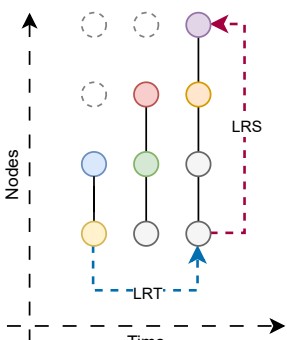

Figure 4: Illustration of LRT and LRS dependency in sequence classification task .

### 7.4 IMPLEMENTATION DETAILS

For link prediction and node classification, we follow the experimental protocol of Yu et al. (2023) and compare `CTDG-SSM` with established baselines. For sequence classification, we adopt the setup from Gravina et al. (2024). The model is trained with binary cross-entropy using the Adam optimizer; additional hyperparameter details are provided in Appendix D.2. We train for up to 200 epochs with early stopping and select the best validation model for testing. Experiments are conducted on two machines equipped with NVIDIA A6000 and RTX 8000 GPUs (48 GB).

## 8 CONCLUSIONS

In this work, we proposed `CTDG-SSM`, a novel representation learning framework for continuous-time dynamic graphs that preserves long-range information across both spatial and temporal dimensions. Our approach formulates a SSM for CTDGs. In particular, we introduced `CTT-HiPPO` that yields memory representations that are topology aware obtained by projecting HiPPO coefficients through a polynomial of graph Laplacian. Leveraging this we proposed a SSM for CTDGs where the memory representations are governed using the evolving topology. We further established theoretical guarantees on the robustness and permutation equivariance of `CTDG-SSM`. Extensive experiments on diverse temporal graph learning tasks-including link prediction, node classification, and sequence classification-demonstrate the effectiveness of our model in jointly capturing LRT and LRS dependencies.

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

## A  STATE-SPACE MODELS

State-space models (SSMs) are widely used for sequence modeling due to their ability to capture long-range dependencies through latent state evolution while remaining computationally efficient compared to Transformers (Gu et al., 2022). For an input signal $\boldsymbol{x}(t)$, an SSM evolves latent states $\boldsymbol{h}(t) \in \mathbb{R}^d$ according to a linear ordinary differential equation (ODE), producing output $\boldsymbol{y}(t)$ as (Gu et al., 2020; 2022; Smith et al., 2023):

$$\frac{d\boldsymbol{h}(t)}{dt} = \boldsymbol{A}(t)\boldsymbol{h}(t) + \boldsymbol{B}(t)\boldsymbol{x}(t) \quad \text{and} \quad \boldsymbol{y}(t) = \boldsymbol{C}(t)\boldsymbol{h}(t) + \boldsymbol{D}(t)\boldsymbol{x}(t), \tag{7}$$

where $\boldsymbol{A}(t)$, $\boldsymbol{B}(t)$, $\boldsymbol{C}(t)$, and $\boldsymbol{D}(t)$ are the state transition, input, output, and feedforward matrices, respectively. In practice, the continuous-time model is discretized using zero-order hold (ZOH), which assumes piecewise constant inputs: $\boldsymbol{x}(t) = \boldsymbol{x}[k]$ for $t \in [t_k, t_{k+1})$ with step size $\Delta[k] = t_{k+1} - t_k$. This yields the discrete-time system:

$$\boldsymbol{h}[k+1] = \bar{\boldsymbol{A}}\boldsymbol{h}[k] + \bar{\boldsymbol{B}}\boldsymbol{x}[k] \quad \text{and} \quad \boldsymbol{y}[k] = \bar{\boldsymbol{C}}\boldsymbol{h}[k] + \bar{\boldsymbol{D}}\boldsymbol{x}[k], \tag{8}$$

where $\bar{\boldsymbol{A}} = e^{\Delta[k]\boldsymbol{A}}$, $\bar{\boldsymbol{B}} = \int_0^{\Delta[k]} e^{\boldsymbol{A}\tau}\boldsymbol{B}\,d\tau$, $\bar{\boldsymbol{C}} = \boldsymbol{C}$, and $\bar{\boldsymbol{D}} = \boldsymbol{D}$.

Research in SSM design has significantly enhanced its effectiveness for sequence modeling. HiPPO (Gu et al., 2020) introduced principled initialization strategies for capturing long-range dependencies. S4 (Gu et al., 2022) extended these with structured parameterizations of the system matrix $\boldsymbol{A}$ (e.g., diagonal plus low-rank decompositions) to enable efficient computation. More recent work like mamba (Gu & Dao, 2024) has further improved scalability and selectivity by making the input and output matrices $\boldsymbol{B}$ and $\boldsymbol{C}$ input-dependent and the system matrix diagonal, enhancing model expressivity while maintaining computational efficiency.

## B  DERIVATION FOR CTT-HiPPO COEFFICIENTS

We derive the solution for the representations/coefficients ($\boldsymbol{H}_{i,\tau}$) for CTT-HiPPO. Recall, to obtain the coefficients we minimize the residual in the observed time interval. To begin with the $\boldsymbol{r}_i(t)$ can be equivalently expressed as

$$\int_0^\tau \|\boldsymbol{r}_i(t)\|_2^2 \, d\mu(t) = \int_0^\tau \|\boldsymbol{X}[:,i](t) - p(\boldsymbol{L}_\tau)\boldsymbol{H}_\tau \boldsymbol{g}(t)\|_2^2 \, d\mu(t),$$

$$= \int_0^\tau \text{Tr}\left[\left(\boldsymbol{X}[:,i](t) - p(\boldsymbol{L}_\tau)\boldsymbol{H}_{i,\tau}\boldsymbol{g}(t)\right)\left(\boldsymbol{X}[:,i](t) - p(\boldsymbol{L}_\tau)\boldsymbol{H}_{i,\tau}\boldsymbol{g}(t)\right)^\top\right] d\mu(t). \tag{9}$$

Using the first optimality condition i.e., $\frac{\partial \|\boldsymbol{r}_i(\tau)\|_2^2}{\partial \boldsymbol{H}_\tau} = 0$ and on simplifying we have

$$\frac{\partial}{\partial \boldsymbol{H}_{i,\tau}} \int_0^\tau \text{Tr}\left[\left(\boldsymbol{X}[:,i](t) - p(\boldsymbol{L}_\tau)\boldsymbol{H}_{i,\tau}\boldsymbol{g}(t)\right)\left(\boldsymbol{X}[:,i](t) - p(\boldsymbol{L}_\tau)\boldsymbol{H}_{i,\tau}\boldsymbol{g}(t)\right)^\top\right] d\mu(t) = 0,$$

$$\int_0^\tau \frac{\partial}{\partial \boldsymbol{H}_{i,\tau}} \left[\text{Tr}\left(p(\boldsymbol{L}_\tau)\boldsymbol{H}_{i,\tau}\boldsymbol{g}(t)\boldsymbol{g}(t)^\top \boldsymbol{H}_{i,\tau}^\top p(\boldsymbol{L}_\tau)^\top\right) - 2\,\text{Tr}\left(p(\boldsymbol{L}_\tau)\boldsymbol{H}_{i,\tau}\boldsymbol{g}(t)\boldsymbol{X}[:,i](t)^\top\right)\right] d\mu(t) = 0,$$

$$\int_0^\tau 2p(\boldsymbol{L}_\tau)^\top p(\boldsymbol{L}_\tau)\boldsymbol{H}_{i,\tau}\boldsymbol{g}(t)\boldsymbol{g}(t)^\top - 2p(\boldsymbol{L}_\tau)^\top \boldsymbol{X}[:,i](t)\boldsymbol{g}(t)^\top \, d\mu(t) = 0,$$

$$\int_0^\tau p(\boldsymbol{L}_\tau)^\top p(\boldsymbol{L}_\tau)\boldsymbol{H}_{i,\tau}\boldsymbol{g}(t)\boldsymbol{g}(t)^\top d\mu(t) = \int_0^\tau p(\boldsymbol{L}_\tau)^\top \boldsymbol{X}[:,i](t)\boldsymbol{g}(t)^\top d\mu(t). \tag{10}$$

For fixed set of orthogonal polynomials we have $\int_0^\tau \boldsymbol{g}(t)\boldsymbol{g}(t)^\top d\mu(t) = \boldsymbol{I}$, then equation 10 can be simplified as

$$p(\boldsymbol{L}_\tau)\boldsymbol{H}_\tau = \int_0^\tau \boldsymbol{X}[:,i](t)\boldsymbol{g}(t)^\top d\mu(t),$$

$$\boldsymbol{H}_{i,\tau} = p(\boldsymbol{L}_\tau)^{-1} \int_0^\tau \boldsymbol{X}[:,i](t)\boldsymbol{g}(t)^\top d\mu(t),$$

$$\boldsymbol{H}_{i,\tau} = p(\boldsymbol{L}_\tau)^{-1}\boldsymbol{H}_{i,\tau}^{(\text{HiPPO})}. \tag{11}$$

where $\boldsymbol{H}_{i,\tau}$ corresponds to the solution from `CTT-HiPPO`, where it is obtained by projecting the HiPPO solution through graph-aware polynomial.

## C  PROOF OF THEOREMS

This section presents detailed proofs of the theorems from the main text.

### C.1  PROOF FOR THEOREM 4.1

*Proof.* We derive the SSM for CTDGs (`CTDG-SSM`) that governs the evolution of memory representations. To begin with, we consider the relation between structural HiPPO coefficients $\boldsymbol{H}_\tau$ and HiPPO coefficients given in equation 3 and obtain the evolution of memory states as follows: For an event observed at $\tau_+$ and corresponding CTDG $\mathcal{G}_{\tau_+}$, we define the polynomial as in the interval $[\tau, \tau_+)$ as $p(L_s) = \frac{\tau - s}{\tau - \tau_+}p(\boldsymbol{L}_{\tau_+}) + \frac{\tau_+ - s}{\tau_+ - \tau}p(\boldsymbol{L}_\tau)$ for $s \in [\tau, \tau_+)$. Here $\boldsymbol{L}_{\tau_+} \in \mathbb{R}^{N_{\tau_+} \times N_{\tau_+}}$ and $\boldsymbol{L}_\tau$ is calculated by removing the newly observed edges from $\boldsymbol{L}_{\tau_+}$. Then the derivative of the coefficients for $s \in [\tau, \tau_+)$ is given as :

$$\frac{d}{ds}\left(p(\boldsymbol{L}_s)\boldsymbol{H}_s\right) = \frac{d\boldsymbol{H}_s^{(HiPPO)}}{ds},$$

$$\frac{dp(\boldsymbol{L}_s)}{ds}\boldsymbol{H}_s + p(\boldsymbol{L}_s)\frac{d\boldsymbol{H}_s}{ds} = \frac{d\boldsymbol{H}_s^{(HiPPO)}}{ds}. \tag{12}$$

This can be equivalently expressed on multiplying with $p(\boldsymbol{L}_s)^{-1}$ as

$$\frac{d\boldsymbol{H}_s}{ds} = p(\boldsymbol{L}_s)^{-1}\frac{d\boldsymbol{H}_s^{(HiPPO)}}{ds} - p(\boldsymbol{L}_s)^{-1}\frac{dp(\boldsymbol{L}_s)}{d\tau}\boldsymbol{H}_s$$

To obtain an equivalent SSM for CTDGs, we leverage the established result from (Gu et al., 2020) that relates the evolution of HiPPO coefficients to a linear ordinary equation as

$$\frac{d\boldsymbol{H}_s^{(HiPPO)}}{ds} = -\boldsymbol{H}_s^{(HiPPO)}\frac{\boldsymbol{A}^\top}{M(s)} + \boldsymbol{x}(s)\frac{\boldsymbol{B}^\top}{M(s)},$$

where $\boldsymbol{A} \in \mathbb{R}^{d \times d}$ is a state transition matrix, $\boldsymbol{B} \in \mathbb{R}^{d \times 1}$ input matrix and $M(\tau) : \mathbb{R}^+ \to \mathbb{R}^+$ is a scalar that depends on the choice of bases polynomial and weigh function $\mu(t)$. The continuous SSM for CTDGs for $s \in [\tau, \tau_+)$ is given by

$$\frac{d\boldsymbol{H}_s}{ds} = -p(\boldsymbol{L}_s)^{-1}\boldsymbol{H}_s^{(HiPPO)}\frac{\boldsymbol{A}^\top}{M(s)} - p(\boldsymbol{L}_s)^{-1}\frac{dp(\boldsymbol{L}_s)}{ds}\boldsymbol{H}_s + p(\boldsymbol{L}_s)^{-1}\boldsymbol{X}(s)\frac{\boldsymbol{B}^\top}{M(s)},$$

$$\frac{d\boldsymbol{H}_s}{ds} = -\boldsymbol{H}_\tau\frac{\boldsymbol{A}^\top}{M(s)} - p(\boldsymbol{L}_s)^{-1}\frac{dp(\boldsymbol{L}_s)}{ds}\boldsymbol{H}_s + p(\boldsymbol{L}_s)^{-1}\boldsymbol{x}(s)\frac{\boldsymbol{B}^\top}{M(s)}. \tag{13}$$

$\square$

We can further simplify equation 13 to express it in a standard first-order state-space model. To do so, we apply vectorization operation on equation 13 and use the identity $\text{vec}(\boldsymbol{ABC}) = (\boldsymbol{C}^\top \otimes \boldsymbol{A})\text{vec}(\boldsymbol{B})$, where $\otimes$ is a Kronecker product. Then we obtain

$$\frac{d\boldsymbol{h}_s}{ds} = -\left(\frac{\boldsymbol{A}}{M(s)} \oplus \left(p(\boldsymbol{L}_s)^{-1}\frac{dp(\boldsymbol{L}_s)}{ds}\right)\right)\boldsymbol{h}_s + \frac{\boldsymbol{B}}{M(s)} \otimes p(\boldsymbol{L}_s)^{-1}(\boldsymbol{x}(s))$$

$$\frac{d\boldsymbol{h}_s}{ds} = \boldsymbol{A}_g(s)\boldsymbol{h}_s + \boldsymbol{B}_g(s)\boldsymbol{X}(s), \quad s \in [\tau, \tau_+) \tag{14}$$

where $\boldsymbol{h}_\tau = \mathrm{vec}(\boldsymbol{H}_\tau) \in \mathbb{R}^{N_T d \times 1}$, $\boldsymbol{A}_g(\tau)$ and $\boldsymbol{B}_g(\tau)$ denote the time-dependent system and input matrices, respectively. Here $\oplus$ denotes the Kronecker sum. The evolution of memory coefficients of nodes so far characterizes the continuous time-variant SSM that jointly encodes dynamic graphs' structural and temporal information.

**Corollary** (Reduction to Classical HiPPO). *Let the graph Laplacian be static ($\boldsymbol{L}_\tau = \boldsymbol{L}$) and let the filter satisfy $p(\boldsymbol{L}_\tau) = \boldsymbol{I}$. Then, the* CTDG-SSM *dynamics equation 5 reduce to the classical HiPPO state-space dynamics:*

$$\frac{d\boldsymbol{H}_\tau}{d\tau} = -\boldsymbol{H}_\tau \frac{\boldsymbol{A}^\top}{M(\tau)} + \boldsymbol{X}(\tau)\frac{\boldsymbol{B}^\top}{M(\tau)}.$$

*This shows that* CTDG-SSM *is a strict generalization of classical HiPPO: it recovers standard memory evolution when the graph is static or the filter is the identity, while naturally incorporating dynamic graph information when $p(\boldsymbol{L}_\tau)$ varies over time.*

**Remark.** *The expression,*

$$\min_{\boldsymbol{H}_{i,\tau}} \int_0^\tau \|\boldsymbol{X}[:,i](t) - p(\boldsymbol{L}_\tau)\boldsymbol{H}_{i,\tau}\boldsymbol{g}(t)\|_2^2 \, d\mu(t),$$

*is the CTDG-SSM formulation, where the polynomial operator $p(\boldsymbol{L}_\tau)$ specifies how the graph structure influences the reconstruction.*

*When $p(\boldsymbol{L}_\tau) = \boldsymbol{I}$, the Laplacian dependence vanishes, yielding the classical HiPPO (Gu et al., 2020) objective:*

$$\min_{\boldsymbol{H}_{i,\tau}} \int_0^\tau \|\boldsymbol{X}[:,i](t) - \boldsymbol{H}_{i,\tau}\boldsymbol{g}(t)\|_2^2 \, d\mu(t),$$

*which matches the standard HiPPO setting.*

*However, when a quadratic Laplacian regularizer is introduced to enforce smoothness over the reconstructed signal, we obtain the GraphSSM (Li et al., 2024b) objective:*

$$\min_{\boldsymbol{H}_{i,\tau}} \int_0^\tau \|\boldsymbol{X}[:,i](t) - \boldsymbol{H}_{i,\tau}\boldsymbol{g}(t)\|_2^2 \, d\mu(t) + \int_0^\tau \left(\boldsymbol{H}_{i,\tau}\boldsymbol{g}(t)\right)^\top \boldsymbol{L}_t \left(\boldsymbol{H}_{i,\tau}\boldsymbol{g}(t)\right) d\mu(t).$$

## C.2 PROOF FOR THEOREM 4.2

*Proof.* We present the equivalent discrete-time SSM for CTDG using ZOH technique. Recall from equation 14 the continuous-time evolution for CTDGs is given as

$$\frac{d\boldsymbol{h}_i[k]}{dt} = \boldsymbol{A}_g(k)\boldsymbol{h}_i[k] + \boldsymbol{B}_g(k)\boldsymbol{X}[:,i][k]. \tag{15}$$

Following equation 8, the equivalent discrete update is given as

$$\boldsymbol{h}_i[k+1] = \exp\left(\boldsymbol{A}_g(t_k)\Delta[k]\right)\boldsymbol{h}_i[k] + \int_0^{\Delta[k]} \exp\left(\boldsymbol{A}_g(k)s\right)\boldsymbol{B}_g(t_k)\boldsymbol{X}[:,i][k] \, ds, \tag{16}$$

where $\Delta[k] = t_{k+1} - t_k$. Recall $\boldsymbol{A}_g(t_k) = -\boldsymbol{A} \oplus \left(-p(\boldsymbol{L}[k])^{-1}\frac{p(\boldsymbol{L}[k])-p(\boldsymbol{L}[k-1])}{\Delta[k]}\right)$, $\boldsymbol{B}_g(t_k) = \boldsymbol{B} \otimes p(\boldsymbol{L}[k])^{-1}$ Although one can directly apply equation 8 as discussed in the preliminaries to obtain a discrete equivalent for equation 14, this approach incurs significant computational overhead in implementation, since the Kronecker-structured matrices $\boldsymbol{A}_g$ and $\boldsymbol{B}_g$ involved in equation 14 are of large dimensions $(N_\tau d \times N_\tau d)$ and $(N_\tau d \times N_\tau)$. To alleviate this complexity, we exploit algebraic properties of the Kronecker product to derive an equivalent update rule as

$$\boldsymbol{h}[k+1] = \left(e^{-\boldsymbol{A}\Delta[k]} \otimes e^{-p(\boldsymbol{L}[k])^{-1}\frac{p(\boldsymbol{L}[k])-p(\boldsymbol{L}[k-1])}{\Delta[k]}\Delta[k]}\right)\boldsymbol{h}[k],$$

$$+ \int_0^1 \left(e^{-\boldsymbol{A}\Delta[k]s} \otimes e^{-p(\boldsymbol{L}[k])^{-1}\frac{p(\boldsymbol{L}[k])-p(\boldsymbol{L}[k-1])}{\Delta[k]}\Delta[k]s}\right)\left(\boldsymbol{B} \otimes p(\boldsymbol{L}[k])^{-1}\right)\boldsymbol{X}[:,i][k]\,\Delta[k]ds, \tag{17}$$

where equation 17 follows by using the following identities $e^{\boldsymbol{A} \oplus \boldsymbol{B}} = e^{\boldsymbol{A}} \otimes e^{\boldsymbol{B}}, \mathrm{vec}(\boldsymbol{ABH}) = (\boldsymbol{H}^\top \otimes \boldsymbol{A})\,\mathrm{vec}(\boldsymbol{B}), (\boldsymbol{A} \otimes \boldsymbol{B})(\boldsymbol{H} \otimes \boldsymbol{D}) = (\boldsymbol{AH}) \otimes (\boldsymbol{BD})$. This can be equivalently expressed as

$$
\boldsymbol{H}[k+1] = e^{-\,p(\boldsymbol{L}[k])^{-1}\frac{p(\boldsymbol{L}[k])-p(\boldsymbol{L}[k-1])}{\Delta[k]}\,\Delta[k]}\,\boldsymbol{H}[k]\,e^{-\boldsymbol{A}^\top \Delta[k]},
$$
$$
+ \int_0^1 e^{-\,p(\boldsymbol{L}[k])^{-1}\frac{p(\boldsymbol{L}[k])-p(\boldsymbol{L}[k-1])}{\Delta[k]}\,\Delta[k]s}\,p(\boldsymbol{L}[k])^{-1}\boldsymbol{X}[:,i][k]\boldsymbol{B}^\top\,e^{-\boldsymbol{A}^\top \Delta[k]s}\,\Delta[k]\,ds.
$$
$$(18)$$

$$
\boldsymbol{H}[k+1] = \bar{\boldsymbol{A}}_{\boldsymbol{L}[k]}\,\boldsymbol{H}[k]\,\bar{\boldsymbol{A}} + \bar{\boldsymbol{B}}(\boldsymbol{L}[k], \boldsymbol{x}[k]), \tag{19}
$$

where $\bar{\boldsymbol{A}}_{\boldsymbol{L}[k]} = \exp(-p(\boldsymbol{L}[k])^{-1}\frac{(p(\boldsymbol{L}[k])-p(\boldsymbol{L}[k-1]))}{\Delta[k]}\Delta[k])$, $\bar{\boldsymbol{A}} = \exp(-\boldsymbol{A}^\top\Delta[k])$, $\bar{\boldsymbol{B}}(\boldsymbol{L}[k], \boldsymbol{X}[:,i][k]) = \int_0^1 (\bar{\boldsymbol{A}}_{\boldsymbol{L}[k]})^s\,p(\boldsymbol{L}[k])^{-1}\boldsymbol{X}[:,i][k]\boldsymbol{B}^\top(\bar{\boldsymbol{A}})^s\,\Delta[k]\,ds$. $\qquad\square$

### C.3 PROOF OF THEOREM 6.1

Consider the $\bar{\boldsymbol{H}}_\tau$ and $\boldsymbol{H}_\tau$ as the memory representations obtained with the perturbed graph Laplacian and true Laplacian. For brevity, we call the solution from HiPPO as $\boldsymbol{H}_H$. Then the error between the representations is given as

$$
\begin{aligned}
\|\bar{\boldsymbol{H}}_\tau - \boldsymbol{H}_\tau\|_2 &= \|P(\bar{\boldsymbol{L}}_\tau)^{-1}\boldsymbol{H}_H - P(\boldsymbol{L}_\tau)\boldsymbol{H}_H\|_2 \\
&\overset{(a)}{\le} \|P(\bar{\boldsymbol{L}}_\tau)^{-1} - P(\boldsymbol{L}_\tau)^{-1}\|_2\|\boldsymbol{H}_H\|_2 \\
&\overset{(b)}{\le} \|P(\bar{\boldsymbol{L}}_\tau)^{-1}(P(\boldsymbol{L}_\tau) - P(\bar{\boldsymbol{L}}_\tau))P(\boldsymbol{L}_\tau)^{-1}\|\|\boldsymbol{H}_H\|_2 \\
&\overset{(c)}{\le} \|P(\bar{\boldsymbol{L}}_\tau)^{-1}\|_2\|P(\boldsymbol{L}_\tau) - P(\bar{\boldsymbol{L}}_\tau)\|_2\|P(\boldsymbol{L}_\tau)^{-1}\|\|\boldsymbol{H}_H\|_2,
\end{aligned}
$$

where equation 20(a), (b), (c) follow from the norm inequalities. Recall $\boldsymbol{L}$ is a normalized Laplacian, therefore the spectrum is bounded in the range $\lambda \in [0, 2]$. Let us call $\lambda_1 := \min_{\lambda \in [0,2]} |p(\lambda)| > 0$, $\lambda_2 := \max_{\lambda \in [0,2]} |p(\lambda)|$, and $\lambda_c := \max_{\lambda \in [0,2]} |p(y)'|$. Then we have

$$
\begin{aligned}
\|\bar{\boldsymbol{H}}_\tau - \boldsymbol{H}_\tau\| &\le \tfrac{1}{\lambda_1^2}\|P(\boldsymbol{L}_\tau) - P(\bar{\boldsymbol{L}}_\tau)\|_2\|\boldsymbol{H}_{i,H}\|_2, \\
&\le \tfrac{\lambda_c}{\lambda_1^2}\|\boldsymbol{L}_\tau - \bar{\boldsymbol{L}}_\tau\|_2\|\boldsymbol{H}_{i,H}\|_2, \\
&\le \tfrac{\lambda_c}{\lambda_1^2}\|_2\boldsymbol{L}_\tau - \bar{\boldsymbol{L}}_\tau\|\|_2 P(\boldsymbol{L}_\tau)(\boldsymbol{H}_\tau)\|_2, \\
&\le \tfrac{\lambda_c}{\lambda_1^2}\|\boldsymbol{L}_\tau - \bar{\boldsymbol{L}}_\tau\|_2\|P(\boldsymbol{L}_\tau)\|_2\|\boldsymbol{H}_\tau\|_2.
\end{aligned} \tag{20}
$$

The normalized error given by

$$
\begin{aligned}
\frac{\|\bar{\boldsymbol{H}}_\tau - \boldsymbol{H}_\tau\|_2}{\|\boldsymbol{H}_\tau\|_2} &\le \tfrac{\lambda_c}{\lambda_1^2}\|\boldsymbol{L}_\tau - \bar{\boldsymbol{L}}_\tau\|\|_2 P(\boldsymbol{L}_\tau)\|_2, \\
&\le \tfrac{\lambda_2 \lambda_c}{\lambda_1^2}\|\boldsymbol{L}_\tau - \bar{\boldsymbol{L}}_\tau\|_2, \\
&\overset{(a)}{\le} \epsilon H,
\end{aligned} \tag{21}
$$

where equation 21(a) since energy of perturbation is bounded i.e., $\|\Delta \boldsymbol{L}\| \le \epsilon$ and $H = \frac{\lambda_2 \lambda_c}{\lambda_1^2}$.

### C.4 PROOF OF THEOREM 6.2

To prove that the representations from CTDG-SSM as permutation equivariant we first show that representations from CTT-HiPPO are equivariant to permutation. Under the permutation the features signal and Laplacian modifies as $\hat{\boldsymbol{X}} = \boldsymbol{\Pi} \boldsymbol{X}$, $\hat{\boldsymbol{L}} = \boldsymbol{\Pi} \boldsymbol{L} \boldsymbol{\Pi}^\top$. Let $\hat{\boldsymbol{H}}_\tau$ be representations obtained

under permutation, then we have

$$
\begin{aligned}
\hat{\boldsymbol{H}}_\tau &= p(\hat{\boldsymbol{L}}_\tau)^{-1} \int_0^\tau \hat{\boldsymbol{X}}(t)\boldsymbol{g}(t)^\top dw(t), \\
&= p(\boldsymbol{\Pi}\boldsymbol{L}_\tau\boldsymbol{\Pi}^\top)^{-1} \int_0^\tau \boldsymbol{\Pi}\boldsymbol{X}(t)\boldsymbol{g}(t)^\top dw(t), \\
&= \boldsymbol{\Pi}p(\boldsymbol{L}_\tau)^{-1}\boldsymbol{\Pi}^\top \int_0^\tau \boldsymbol{\Pi}\boldsymbol{X}(t)\boldsymbol{g}(t)^\top dw(t), \\
&= \boldsymbol{\Pi}p(\boldsymbol{L}_\tau)^{-1} \int_0^\tau \boldsymbol{X}(t)\boldsymbol{g}(t)^\top dw(t) \\
&= \boldsymbol{\Pi}\boldsymbol{H}_\tau,
\end{aligned}
\tag{22}
$$

equation 22 implies the representations obtained from `CTT-HiPPO` are permutation equivariant. Now, to prove the equivariance for the representations from `CTDG-SSM` layer we first evaluate state matrix $\bar{\boldsymbol{A}}$ and system matrix $\bar{\boldsymbol{B}}$ under permutation as

$$
\begin{aligned}
\bar{\boldsymbol{A}}^s_{\hat{\boldsymbol{L}}[k]} &= \exp(-p(\hat{\boldsymbol{L}}[k])^{-1}(p(\hat{\boldsymbol{L}}[k-1]) - p(\hat{\boldsymbol{L}}[k])s), \\
&= \exp(-\boldsymbol{\Pi}p(\boldsymbol{L}[k])^{-1}\boldsymbol{\Pi}^\top\boldsymbol{\Pi}(p(\boldsymbol{L}[k]) - p(\boldsymbol{L}[k-1])\boldsymbol{\Pi}^\top s), \\
&= \exp(-\boldsymbol{\Pi}p(\boldsymbol{L}[k])^{-1}(p(\boldsymbol{L}[k]) - p(\boldsymbol{L}[k-1])\boldsymbol{\Pi}^\top s), \\
&= \boldsymbol{\Pi} \exp(-p(\boldsymbol{L}[k])^{-1}(p(\boldsymbol{L}[k]) - p(\boldsymbol{L}[k-1])s)\boldsymbol{\Pi}^\top, \\
&= \boldsymbol{\Pi}\bar{\boldsymbol{A}}^s_{\boldsymbol{L}[k]}\boldsymbol{\Pi}^\top,
\end{aligned}
\tag{23}
$$

where $\bar{\boldsymbol{B}}$ modifies as

$$
\begin{aligned}
\bar{\boldsymbol{B}}(\hat{\boldsymbol{L}}[k], \hat{\boldsymbol{X}}[k]) &= \int_0^1 \bar{\boldsymbol{A}}^s_{\hat{\boldsymbol{L}}[k]} p(\hat{\boldsymbol{L}}[k])^{-1}\hat{\boldsymbol{X}}[k]\boldsymbol{B}^\top\bar{\boldsymbol{A}}^s\,\Delta[k]\,ds, \\
&= \int_0^1 \boldsymbol{\Pi}\bar{\boldsymbol{A}}^s_{\boldsymbol{L}[k]}\boldsymbol{\Pi}^\top\,\boldsymbol{\Pi}p(\boldsymbol{L}[k])^{-1}\boldsymbol{\Pi}^\top\boldsymbol{\Pi}\boldsymbol{X}[k]\boldsymbol{B}^\top\bar{\boldsymbol{A}}^s\,\Delta[k]\,ds, \\
&= \boldsymbol{\Pi}\int_0^1 \bar{\boldsymbol{A}}^s_{\boldsymbol{L}[k]} p(\boldsymbol{L}[k])^{-1}\boldsymbol{X}[k]\boldsymbol{B}^\top\bar{\boldsymbol{A}}^s\,\Delta[k]\,ds. \\
&= \boldsymbol{\Pi}\bar{\boldsymbol{B}}(\boldsymbol{L}[k], \boldsymbol{X}[k])
\end{aligned}
\tag{24}
$$

Now we show that the updates from `CTDG-SSM` are permutation equivariant. Consider

$$
\begin{aligned}
\hat{\boldsymbol{H}}[k+1] &= \bar{\boldsymbol{A}}_{\hat{\boldsymbol{L}}[k]}\hat{\boldsymbol{H}}[k]\bar{\boldsymbol{A}} + \bar{\boldsymbol{B}}(\hat{\boldsymbol{L}}[k], \hat{\boldsymbol{X}}[k]), \\
&\overset{(a)}{=} \bar{\boldsymbol{A}}_{\hat{\boldsymbol{L}}[k]}(\boldsymbol{\Pi}\boldsymbol{H}[k])\bar{\boldsymbol{A}} + \boldsymbol{\Pi}\bar{\boldsymbol{B}}(\boldsymbol{L}[k], \boldsymbol{X}[k]), \\
&\overset{(b)}{=} \boldsymbol{\Pi}\bar{\boldsymbol{A}}_{\boldsymbol{L}[k]}\boldsymbol{H}[k]\bar{\boldsymbol{A}} + \boldsymbol{\Pi}\bar{\boldsymbol{B}}(\boldsymbol{L}[k], \boldsymbol{X}[k]), \\
&= \boldsymbol{\Pi}\left(\bar{\boldsymbol{A}}_{\boldsymbol{L}[k]}\boldsymbol{H}[k]\bar{\boldsymbol{A}} + \bar{\boldsymbol{B}}(\boldsymbol{L}[k], \boldsymbol{X}[k])\right), \\
&= \boldsymbol{\Pi}\boldsymbol{H}[k+1],
\end{aligned}
\tag{25}
$$

where equation 25(a) follows by recursion. Recall $k = 0$ we have $\hat{\boldsymbol{H}}[1] = \bar{\boldsymbol{B}}(\hat{\boldsymbol{L}}[0], \hat{\boldsymbol{X}}[0])$ as $\boldsymbol{H}[0] = \boldsymbol{0}$, hence $\hat{\boldsymbol{H}}[1] = \boldsymbol{\Pi}\bar{\boldsymbol{B}}(\boldsymbol{L}[0], \hat{\boldsymbol{X}}[0]) = \boldsymbol{\Pi}\boldsymbol{H}[1]$ from equation 24 which is propogated through $k$ layers. Then equation 25(b) follows from equation 23 and equation 24.

# D  NUMERICAL EXPERIMENTS AND ADDITIONAL RESULTS

In this section, we discuss the dataset details, hyperparameters, and the additional results on the dynamic link prediction task.

## D.1  DATASET DETAILS

We provide a detailed description of the datasets considered for experimentation in Table 4. In all the datasets, LastFM, Enron and MOOC are mainly considered for evaluating the LRT task. In

Table 4: Statistics of the datasets used in our experiments. #N & L feat corresponds to the dimension of node and link features, where - represents the unavailability of node features.

| Dataset | Domain | #Nodes | #Links | #N&L Feat | Bipartite | Duration | Unique Steps | Time Granularity |
|---|---|---|---|---|---|---|---|---|
| Wikipedia | Social | 9,227 | 157,474 | – & 172 | True | 1 month | 152,757 | Unix timestamps |
| Reddit | Social | 10,984 | 672,447 | – & 172 | True | 1 month | 669,065 | Unix timestamps |
| MOOC | Interaction | 7,144 | 411,749 | – & 4 | True | 17 months | 345,600 | Unix timestamps |
| LastFM | Interaction | 1,980 | 1,293,103 | – & – | True | 1 month | 1,283,614 | Unix timestamps |
| Enron | Social | 184 | 125,235 | – & – | False | 3 years | 22,632 | Unix timestamps |
| UCI | Social | 1,899 | 59,835 | – & – | False | 196 days | 58,911 | Unix timestamps |
| Social Evo. | Proximity | 74 | 2,099,519 | – & 2 | False | 8 months | 565,932 | Unix timestamps |
| Flights | Transport | 13,169 | 1,927,145 | - & 1 | False | 4 months | 122 | days |
| Can. Parl. | Politics | 734 | 74,478 | - & 1 | False | 14 years | 14 | years |
| US Legis. | Politics | 225 | 60,396 | - & 1 | False | 12 congresses | 12 | congresses |
| UN Trade | Economics | 255 | 507,497 | - & 1 | False | 32 years | 32 | years |
| UN Vote | Politics | 201 | 1,035,742 | - & 1 | False | 72 years | 72 | years |
| Contact | Proximity | 692 | 2,426,279 | - & 1 | False | 1 month | 8,064 | 5 minutes |
| tgbl-wiki | Interaction | 9,227 | 157,474 | - & 1 | True | 1 month | 152,757 | Unix timestamps |
| tgbl-coin | Economics | 638,486 | 22,809,486 | - & 1 | False | 7 month | 1,295,720 | Unix timestamps |

particular, The LastFM dataset corresponds to data from a music streaming platform that records user listening behaviors, where users and songs are nodes and links denote listening events (Celma, 2010). The Enron dataset is an email communication dataset among employees of the Enron Corporation, recorded over a three-year period (Klimt & Yang, 2004). Whereas the MOOC dataset captures student interactions on an online course platform, where links represent students accessing course content such as videos or problem sets (Kizilcec et al., 2013), other DTDG datasets used for evaluation include Flights, Can. Parl, US Legis., UN Trade, UN Vote, and Contact include. For all datasets used in data processing, we employ the same pipeline described in (Yu et al., 2023). Additionally, datasets including `tgbl-wiki` and `tgbl-coin` from (Huang et al., 2023) were also utilized.

Table 5: AUC-ROC for transductive dynamic link prediction under. RNS: Random Negative Sampling, HNS: Historical Negative Sampling, INS : Inductive Negative Sampling.

| NSS | Datasets | JODIE | DyRep | TGAT | TGN | CAWN | TCL | GraphMixer | DyGFormer | CTAN | DyGmamba | CTDG-SSM |
|---|---|---|---|---|---|---|---|---|---|---|---|---|
| RNS | LastFM | 70.89 ± 1.97 | 71.40 ± 2.12 | 71.47 ± 0.14 | 76.64 ± 4.66 | 85.92 ± 0.16 | 71.09 ± 1.48 | 73.51 ± 0.14 | 93.03 ± 0.11 | 85.12 ± 0.77 | 93.31 ± 0.18 | **93.79 ± 0.22** |
| | Enron | 87.77 ± 2.43 | 83.09 ± 2.20 | 68.57 ± 1.46 | 88.72 ± 0.95 | 90.34 ± 0.23 | 83.33 ± 0.93 | 84.16 ± 0.34 | 93.20 ± 0.12 | 87.09 ± 1.51 | 93.34 ± 0.23 | **94.98 ± 2.92** |
| | MOOC | 84.50 ± 0.60 | 84.50 ± 0.87 | 87.01 ± 0.16 | 91.91 ± 0.82 | 80.48 ± 0.41 | 84.02 ± 0.59 | 84.04 ± 0.12 | 88.08 ± 0.50 | 85.40 ± 2.67 | 89.58 ± 0.12 | **99.00 ± 0.33** |
| | Reddit | 98.29 ± 0.05 | 98.13 ± 0.04 | 98.50 ± 0.01 | 98.61 ± 0.05 | 99.02 ± 0.00 | 97.67 ± 0.01 | 97.17 ± 0.02 | 99.15 ± 0.01 | 97.24 ± 0.75 | 99.27 ± 0.01 | **99.48 ± 0.02** |
| | Wikipedia | 96.36 ± 0.14 | 94.43 ± 0.32 | 96.60 ± 0.07 | 98.37 ± 0.10 | 98.54 ± 0.01 | 97.27 ± 0.06 | 96.89 ± 0.04 | 98.92 ± 0.03 | 97.00 ± 0.21 | 99.08 ± 0.02 | **99.33 ± 0.08** |
| | UCI | 90.35 ± 0.51 | 69.46 ± 2.66 | 78.76 ± 1.10 | 92.03 ± 0.69 | 93.81 ± 0.23 | 85.49 ± 0.82 | 91.62 ± 0.52 | 94.45 ± 0.22 | 76.25 ± 2.83 | **94.77 ± 0.18** | 89.24 ± 0.43 |
| | Social Evo. | 92.13 ± 0.20 | 90.37 ± 0.52 | 94.93 ± 0.06 | 95.31 ± 0.27 | 87.34 ± 0.10 | 95.45 ± 0.21 | 95.21 ± 0.07 | 96.25 ± 0.04 | Timeout | 96.38 ± 0.02 | **99.10 ± 0.49** |
| | **Avg. Rank** | 7.93 | 9.36 | 7.86 | 4.57 | 5.71 | 8.00 | 7.71 | 3.00 | 7.50 | 2.00 | **1.86** |
| HNS | LastFM | 75.65 ± 4.43 | 70.63 ± 2.56 | 64.23 ± 0.45 | 78.00 ± 2.97 | 67.92 ± 0.32 | 60.53 ± 2.54 | 64.06 ± 0.34 | 78.80 ± 0.02 | 79.50 ± 0.82 | 79.82 ± 0.27 | **89.55 ± 0.57** |
| | Enron | 75.21 ± 1.27 | 76.36 ± 1.42 | 62.36 ± 1.07 | 76.75 ± 1.40 | 65.62 ± 0.49 | 71.72 ± 1.24 | 74.82 ± 2.04 | 77.35 ± 0.64 | 81.95 ± 1.64 | 77.73 ± 0.61 | **95.86 ± 2.18** |
| | MOOC | 82.38 ± 1.75 | 80.71 ± 2.08 | 81.53 ± 0.79 | 86.59 ± 2.03 | 71.74 ± 0.88 | 73.22 ± 1.21 | 77.09 ± 0.83 | 87.26 ± 0.83 | 73.87 ± 2.77 | 87.91 ± 0.93 | **95.22 ± 1.65** |
| | Reddit | 80.70 ± 0.20 | 79.96 ± 0.23 | 79.60 ± 0.09 | 81.04 ± 0.23 | 80.42 ± 0.20 | 76.83 ± 0.12 | 77.83 ± 0.33 | 80.61 ± 0.48 | 90.63 ± 2.28 | 81.71 ± 0.49 | **97.49 ± 0.17** |
| | Wikipedia | 80.71 ± 0.64 | 77.49 ± 0.72 | 82.83 ± 0.27 | 83.28 ± 0.26 | 65.74 ± 3.46 | 85.55 ± 0.47 | 87.47 ± 0.20 | 72.78 ± 6.65 | 95.43 ± 0.07 | 78.99 ± 1.24 | **99.02 ± 0.17** |
| | UCI | 78.21 ± 3.18 | 58.65 ± 3.58 | 57.12 ± 0.98 | 78.48 ± 1.79 | 57.67 ± 1.11 | 65.42 ± 2.62 | 77.46 ± 1.63 | 75.71 ± 0.57 | 75.05 ± 0.13 | 75.43 ± 1.99 | **87.86 ± 0.59** |
| | Social Evo. | 91.83 ± 1.52 | 92.81 ± 0.60 | 93.63 ± 0.48 | 94.27 ± 1.33 | 87.61 ± 0.06 | 93.05 ± 0.82 | 94.65 ± 0.28 | 97.16 ± 0.06 | Timeout | 97.27 ± 0.30 | **98.89 ± 0.56** |
| | **Avg. Rank** | 6.00 | 7.71 | 8.43 | 4.43 | 9.57 | 8.14 | 6.86 | 5.00 | 5.14 | 3.71 | **1.00** |
| INS | LastFM | 61.59 ± 5.72 | 60.62 ± 2.20 | 63.96 ± 0.41 | 65.48 ± 4.13 | 67.90 ± 0.44 | 54.75 ± 1.31 | 59.98 ± 0.20 | 67.87 ± 0.53 | 78.70 ± 0.87 | 68.74 ± 0.55 | **94.17 ± 0.22** |
| | Enron | 70.75 ± 0.69 | 67.37 ± 2.21 | 59.88 ± 1.12 | 73.22 ± 0.42 | 75.29 ± 0.66 | 69.74 ± 1.19 | 70.72 ± 1.08 | 74.67 ± 0.30 | 75.40 ± 1.92 | 75.47 ± 1.41 | **95.80 ± 1.96** |
| | MOOC | 67.53 ± 1.76 | 62.60 ± 1.27 | 74.44 ± 0.81 | 76.89 ± 2.13 | 70.08 ± 0.33 | 71.80 ± 1.09 | 72.25 ± 0.57 | 80.78 ± 0.89 | 68.17 ± 3.73 | 81.08 ± 0.82 | **99.08 ± 0.35** |
| | Reddit | 83.40 ± 0.33 | 82.75 ± 0.36 | 87.46 ± 0.10 | 84.57 ± 0.19 | 88.19 ± 0.20 | 84.41 ± 0.18 | 82.24 ± 0.24 | 86.25 ± 0.64 | 91.42 ± 2.18 | 86.35 ± 0.52 | **99.51 ± 0.03** |
| | Wikipedia | 70.41 ± 0.39 | 67.57 ± 0.94 | 81.54 ± 0.31 | 81.21 ± 0.30 | 68.48 ± 3.64 | 73.51 ± 1.88 | 84.20 ± 0.36 | 64.09 ± 9.75 | 93.67 ± 0.11 | 75.64 ± 2.42 | **99.36 ± 0.07** |
| | UCI | 64.14 ± 1.25 | 54.10 ± 2.74 | 59.60 ± 0.61 | 63.76 ± 0.99 | 57.85 ± 0.59 | 65.46 ± 2.07 | 74.25 ± 0.71 | 64.92 ± 0.83 | 66.51 ± 0.25 | 66.83 ± 2.83 | **89.75 ± 0.32** |
| | Social Evo. | 91.81 ± 1.69 | 92.77 ± 0.64 | 93.54 ± 0.48 | 94.86 ± 1.25 | 90.10 ± 0.11 | 95.13 ± 0.83 | 94.50 ± 0.26 | 95.01 ± 0.15 | Timeout | 97.37 ± 0.26 | **99.24 ± 0.47** |
| | **Avg. Rank** | 8.29 | 9.86 | 6.71 | 5.86 | 6.86 | 7.14 | 6.57 | 5.71 | 3.67 | 3.29 | **1.00** |

Table 6: AUC-ROC of inductive dynamic link prediction.

| NSS | Datasets | JODIE | DyRep | TGAT | TGN | CAWN | TCL | GraphMixer | DyGFormer | CTAN | DyGmamba | CTDG-SSM |
|---|---|---|---|---|---|---|---|---|---|---|---|---|
| RNS | LastFM | 83.13 ± 1.19 | 83.47 ± 1.06 | 78.40 ± 0.30 | 81.18 ± 3.27 | 89.33 ± 0.06 | 81.38 ± 1.53 | 82.07 ± 0.31 | 94.17 ± 0.10 | 60.40 ± 3.01 | 94.42 ± 0.21 | **94.49 ± 0.27** |
| | Enron | 78.97 ± 1.59 | 73.97 ± 3.00 | 66.67 ± 1.07 | 78.76 ± 1.69 | 86.30 ± 0.56 | 82.61 ± 0.61 | 75.55 ± 0.81 | 89.62 ± 0.27 | 74.61 ± 1.64 | 89.67 ± 0.27 | **93.66 ± 4.67** |
| | MOOC | 80.57 ± 0.52 | 80.50 ± 0.68 | 85.28 ± 0.30 | 88.01 ± 1.48 | 81.32 ± 0.42 | 82.28 ± 0.99 | 81.38 ± 0.17 | 87.05 ± 0.51 | 64.99 ± 2.24 | 88.64 ± 0.08 | **98.67 ± 0.46** |
| | Reddit | 96.43 ± 0.16 | 95.89 ± 0.26 | 97.13 ± 0.04 | 97.41 ± 0.10 | 98.62 ± 0.01 | 95.01 ± 0.10 | 95.24 ± 0.08 | 98.83 ± 0.02 | 80.07 ± 2.53 | 98.97 ± 0.01 | **99.13 ± 0.03** |
| | Wikipedia | 94.91 ± 0.32 | 92.21 ± 0.29 | 96.26 ± 0.12 | 97.81 ± 0.18 | 98.27 ± 0.02 | 97.48 ± 0.06 | 96.61 ± 0.04 | 98.58 ± 0.01 | 93.58 ± 0.65 | 98.77 ± 0.03 | **99.06 ± 0.10** |
| | UCI | 79.73 ± 1.48 | 62.97 ± 2.38 | 80.40 ± 0.77 | 82.81 ± 1.32 | 92.61 ± 0.96 | 84.19 ± 1.37 | 91.17 ± 0.29 | 94.45 ± 0.35 | 79.78 ± 5.02 | 94.76 ± 0.19 | **87.43 ± 0.79** |
| | Social Evo. | 91.72 ± 0.66 | 89.10 ± 1.90 | 91.47 ± 0.10 | 90.74 ± 1.40 | 79.83 ± 0.14 | 92.51 ± 0.11 | 91.89 ± 0.05 | 93.05 ± 0.10 | Timeout | 93.13 ± 0.05 | **98.60 ± 0.14** |
| | **Avg. Rank** | 7.29 | 9.00 | 8.00 | 6.00 | 5.29 | 6.57 | 6.71 | 3.00 | 10.57 | 1.86 | **1.71** |
| INS | LastFM | 69.85 ± 1.70 | 68.14 ± 1.61 | 69.89 ± 0.41 | 67.01 ± 5.77 | 67.72 ± 0.20 | 63.15 ± 1.17 | 69.93 ± 0.17 | 69.86 ± 0.80 | 57.85 ± 3.67 | 70.59 ± 0.57 | **94.77 ± 0.26** |
| | Enron | 65.95 ± 1.27 | 62.20 ± 2.15 | 56.52 ± 0.84 | 64.21 ± 0.94 | 62.07 ± 0.72 | 67.56 ± 1.34 | 67.39 ± 1.33 | 66.07 ± 0.65 | 68.70 ± 1.82 | 68.98 ± 1.00 | **94.59 ± 3.37** |
| | MOOC | 65.37 ± 0.96 | 62.97 ± 2.05 | 74.94 ± 0.80 | 76.36 ± 2.91 | 71.18 ± 0.54 | 71.30 ± 1.21 | 72.15 ± 0.65 | 80.42 ± 0.72 | 58.06 ± 0.89 | 81.12 ± 0.63 | **98.71 ± 0.47** |
| | Reddit | 61.84 ± 0.44 | 60.35 ± 0.53 | 64.92 ± 0.08 | 65.24 ± 0.08 | 65.37 ± 0.12 | 61.85 ± 0.11 | 64.56 ± 0.26 | 64.80 ± 0.53 | 81.70 ± 4.71 | 64.93 ± 0.89 | **99.15 ± 0.03** |
| | Wikipedia | 61.66 ± 0.30 | 56.34 ± 0.67 | 78.40 ± 0.77 | 75.86 ± 0.50 | 68.99 ± 4.33 | 71.45 ± 2.23 | 75.72 ± 0.70 | 64.37 ± 0.98 | 91.12 ± 0.13 | 67.92 ± 2.23 | **99.09 ± 0.10** |
| | UCI | 60.66 ± 1.82 | 51.50 ± 2.08 | 61.27 ± 0.78 | 62.07 ± 0.67 | 55.60 ± 1.22 | 65.87 ± 1.90 | 75.72 ± 0.70 | 64.37 ± 0.98 | 51.68 ± 2.60 | 66.95 ± 2.22 | **87.86 ± 0.73** |
| | Social Evo. | 88.98 ± 0.81 | 86.43 ± 1.48 | 92.37 ± 0.50 | 91.66 ± 2.14 | 83.84 ± 0.21 | 95.50 ± 0.31 | 93.88 ± 0.22 | 94.97 ± 0.36 | Timeout | 96.65 ± 0.29 | **98.90 ± 0.14** |
| | **Avg. Rank** | 8.00 | 9.71 | 6.14 | 6.14 | 8.14 | 6.14 | 4.57 | 5.71 | 7.14 | 3.29 | **1.00** |

Table 7: Model Hyperparameters. N/A: Not Applicable, OHE: One-hot encoding, LR: Learnable.

| Dataset | Latent dimension | Time embedding dimension | $N_u$ | Batch size | Static embedding |
|---|---|---|---|---|---|
| Enron | 32 | 16 | 10 | 128 | OHE |
| UCI | 32 | 16 | 10 | 128 | N/A |
| MOOC | 32 | 16 | 10 | 128 | N/A |
| Wikipedia | 128 | 16 | 10 | 128 | N/A |
| Reddit | 128 | 16 | 10 | 128 | N/A |
| Lastfm | 32 | 16 | 10 | 128 | N/A |
| Flights | 32 | 16 | 10 | 128 | N/A |
| Can. Parl. | 32 | 16 | 10 | 128 | N/A |
| US Legis. | 32 | 16 | 10 | 128 | N/A |
| UN Trade | 32 | 16 | 10 | 128 | N/A |
| UN Vote | 32 | 16 | 10 | 128 | N/A |
| Contact | 32 | 16 | 10 | 128 | N/A |
| tgbl-wiki | 128 | 16 | 10 | 128 | N/A |
| tgbl-coin | 32 | 16 | 10 | 128 | LR |
| Sequence Classification | 32 | N/A | 10 | 128 | N/A |

## D.2 ADDITIONAL RESULTS AND HYPERPARAMETER DETAILS

In this section, we provide additional results for the dynamic link prediction task. Specifically, we report performance using average precision (AP) as an evaluation metric. Furthermore, we present AUC-ROC results under both inductive and transductive settings, comparing different sampling strategies. In Table 5, 15 and Table 8, 14, 13 we report AUC-ROC and AP scores under the transductive setting with different sampling techniques. The results clearly demonstrate that the proposed model outperforms state-of-the-art algorithms on LRT datasets, primarily due to its ability to jointly encode structural information via graph polynomials that capture multi-hop neighborhood interactions and temporal evolution through a state-space formulation. In Table 6, 17, and Table 9, 16, 18 we report results under the inductive setting, where the task is more challenging since the test set includes nodes unseen during training. Additionally, we report the mean reciprocal rank (MRR) in Table 19 using the evaluation mechanism proposed in (Huang et al., 2023) (values close to 1 are better). The proposed model not only outperforms existing approaches but also exhibits only a minor performance drop compared to the transductive setting, highlighting its ability to effectively capture global structural and temporal patterns instead of learning local structural patterns.

**Hyperparameter Details**: In Table D.1, we report the hyperparameters used in all experiments. The latent dimension corresponds to the size of the memory representations, the batch size denotes the number of events in each batch, and OHE refers to one-hot encoding.

Table 8: AP of transductive dynamic link prediction.

| NSS | Datasets | JODIE | DyRep | TGAT | TGN | CAWN | TCL | GraphMixer | DyGFormer | CTAN | DyGmamba | CTDG−SSM |
|---|---|---|---|---|---|---|---|---|---|---|---|---|
| RNS | LastFM | 70.95 ± 2.94 | 71.85 ± 2.44 | 73.30 ± 0.18 | 75.31 ± 5.62 | 86.60 ± 0.11 | 76.62 ± 1.83 | 75.56 ± 0.19 | 92.95 ± 0.14 | 86.44 ± 0.80 | 93.35 ± 0.20 | **93.40 ± 0.49** |
| | Enron | 84.85 ± 3.13 | 79.80 ± 2.28 | 70.76 ± 1.05 | 86.98 ± 1.05 | 89.50 ± 0.10 | 85.41 ± 0.71 | 82.13 ± 0.30 | 92.42 ± 0.11 | 92.52 ± 1.20 | 92.65 ± 0.12 | **94.46 ± 4.73** |
| | MOOC | 81.04 ± 0.83 | 81.50 ± 0.77 | 85.71 ± 0.20 | 89.15 ± 1.69 | 80.30 ± 0.43 | 83.89 ± 0.86 | 82.80 ± 0.15 | 87.66 ± 0.48 | 84.71 ± 2.85 | 89.21 ± 0.08 | **98.85 ± 0.35** |
| | Reddit | 98.31 ± 0.06 | 98.18 ± 0.03 | 98.57 ± 0.01 | 98.65 ± 0.04 | 99.11 ± 0.01 | 97.78 ± 0.02 | 97.31 ± 0.01 | 99.22 ± 0.01 | 97.21 ± 0.84 | 99.32 ± 0.01 | **99.53 ± 0.02** |
| | Wikipedia | 96.51 ± 0.22 | 94.88 ± 0.29 | 96.88 ± 0.06 | 98.45 ± 0.10 | 98.77 ± 0.01 | 97.75 ± 0.04 | 97.22 ± 0.02 | 99.03 ± 0.03 | 96.61 ± 0.79 | 99.15 ± 0.02 | **99.40 ± 0.09** |
| | UCI | 89.28 ± 1.02 | 66.11 ± 2.75 | 79.40 ± 0.61 | 92.33 ± 0.64 | 95.13 ± 0.23 | 86.63 ± 1.30 | 93.15 ± 0.41 | 95.74 ± 0.17 | 76.64 ± 4.11 | **95.91 ± 0.15** | 90.18 ± 0.98 |
| | Social Evo. | 89.88 ± 0.40 | 88.39 ± 0.69 | 93.33 ± 0.06 | 93.45 ± 0.29 | 84.90 ± 0.11 | 93.82 ± 0.19 | 93.36 ± 0.06 | 94.63 ± 0.07 | Timeout | 94.77 ± 0.01 | **98.65 ± 0.65** |
| | **Avg. Rank** | 8.71 | 9.71 | 7.86 | 5.29 | 5.86 | 6.71 | 7.29 | 3.14 | 7.86 | 1.86 | **1.71** |
| HNS | LastFM | 74.38 ± 6.27 | 71.85 ± 2.91 | 71.60 ± 0.36 | 75.03 ± 6.90 | 69.93 ± 0.33 | 71.02 ± 2.07 | 72.28 ± 0.37 | 81.51 ± 0.14 | 82.29 ± 0.94 | 83.02 ± 0.16 | **88.91 ± 0.93** |
| | Enron | 69.13 ± 1.66 | 72.58 ± 1.83 | 64.24 ± 1.24 | 74.31 ± 1.99 | 65.40 ± 0.36 | 72.39 ± 0.61 | 77.35 ± 1.22 | 76.93 ± 0.76 | 77.24 ± 1.53 | 77.77 ± 1.32 | **95.80 ± 3.33** |
| | MOOC | 78.62 ± 2.43 | 75.14 ± 2.86 | 82.83 ± 0.71 | 85.65 ± 2.32 | 74.46 ± 0.53 | 78.51 ± 0.84 | 77.09 ± 0.83 | 86.43 ± 0.38 | 67.73 ± 2.08 | 85.89 ± 0.94 | **94.76 ± 1.76** |
| | Reddit | 79.96 ± 0.30 | 79.40 ± 0.00 | 79.78 ± 0.25 | 81.05 ± 0.32 | 80.96 ± 0.28 | 77.38 ± 0.02 | 78.39 ± 0.36 | 83.81 ± 1.08 | 89.77 ± 2.28 | 88.81 ± 1.52 | **97.55 ± 0.22** |
| | Wikipedia | 81.16 ± 0.73 | 79.44 ± 0.95 | 87.31 ± 0.36 | 87.31 ± 0.25 | 66.77 ± 6.62 | 86.12 ± 1.69 | 90.74 ± 0.06 | 70.13 ± 11.02 | 95.91 ± 0.10 | 81.77 ± 1.20 | **98.99 ± 0.32** |
| | UCI | 74.77 ± 5.35 | 55.89 ± 2.83 | 66.78 ± 0.77 | 81.32 ± 1.26 | 64.69 ± 1.78 | 74.62 ± 2.70 | 83.88 ± 1.06 | 80.44 ± 1.16 | 76.62 ± 0.33 | 81.03 ± 1.09 | **88.87 ± 1.28** |
| | Social Evo. | 91.26 ± 2.47 | 92.86 ± 0.90 | 95.31 ± 0.30 | 93.84 ± 1.68 | 85.65 ± 0.11 | 95.93 ± 0.63 | 95.30 ± 0.34 | 97.05 ± 0.16 | Timeout | 97.35 ± 0.52 | **98.20 ± 0.81** |
| | **Avg. Rank** | 7.43 | 8.71 | 6.43 | 4.93 | 9.71 | 7.71 | 5.57 | 4.71 | 5.57 | 3.29 | **1.00** |
| INS | LastFM | 62.63 ± 6.89 | 62.49 ± 3.04 | 71.16 ± 0.33 | 65.09 ± 7.05 | 67.38 ± 0.57 | 62.76 ± 0.81 | 67.87 ± 0.37 | 72.60 ± 0.06 | 80.06 ± 0.85 | 73.63 ± 0.54 | **93.81 ± 0.44** |
| | Enron | 69.51 ± 1.06 | 66.78 ± 2.21 | 73.27 ± 0.58 | 73.27 ± 0.58 | 75.08 ± 0.81 | 70.98 ± 0.96 | 74.12 ± 0.65 | 78.22 ± 0.80 | 72.02 ± 2.64 | 80.86 ± 1.24 | **95.81 ± 2.99** |
| | MOOC | 66.56 ± 1.49 | 61.48 ± 0.96 | 76.96 ± 0.89 | 77.59 ± 1.83 | 73.55 ± 0.36 | 76.35 ± 1.41 | 74.24 ± 0.75 | 80.99 ± 0.88 | 64.93 ± 3.31 | 81.11 ± 0.63 | **99.03 ± 0.38** |
| | Reddit | 86.93 ± 0.21 | 86.06 ± 0.36 | 89.93 ± 0.10 | 88.12 ± 0.13 | 91.89 ± 0.18 | 86.97 ± 0.26 | 85.37 ± 0.26 | 91.06 ± 0.09 | 90.99 ± 2.19 | 91.15 ± 0.54 | **99.58 ± 0.02** |
| | Wikipedia | 74.78 ± 0.56 | 70.55 ± 1.22 | 86.77 ± 0.29 | 85.80 ± 0.15 | 69.27 ± 7.07 | 72.54 ± 4.69 | 88.54 ± 0.20 | 62.00 ± 14.00 | 94.15 ± 0.08 | 79.86 ± 2.18 | **99.45 ± 0.06** |
| | UCI | 66.02 ± 1.28 | 54.64 ± 2.52 | 67.63 ± 0.51 | 70.34 ± 0.72 | 64.08 ± 1.06 | 73.49 ± 2.21 | 79.57 ± 0.61 | 70.51 ± 1.83 | 66.25 ± 0.51 | 71.95 ± 2.51 | **91.44 ± 0.50** |
| | Social Evo. | 91.08 ± 3.29 | 92.84 ± 0.98 | 95.20 ± 0.30 | 94.58 ± 1.52 | 88.50 ± 0.13 | 96.14 ± 0.63 | 95.11 ± 0.32 | 97.62 ± 0.12 | Timeout | 97.68 ± 0.42 | **98.88 ± 0.63** |
| | **Avg. Rank** | 8.86 | 10.00 | 6.14 | 6.14 | 7.29 | 6.57 | 5.71 | 4.71 | 6.43 | 3.14 | **1.00** |

Table 9: AP of inductive dynamic link prediction.

| NSS | Datasets | JODIE | DyRep | TGAT | TGN | CAWN | TCL | GraphMixer | DyGFormer | CTAN | DyGmamba | CTDG-SSM |
|---|---|---|---|---|---|---|---|---|---|---|---|---|
| RNS | LastFM | 83.13 ± 1.19 | 83.47 ± 1.06 | 78.40 ± 0.30 | 81.18 ± 3.27 | 89.33 ± 0.06 | 81.38 ± 1.53 | 82.07 ± 0.31 | 94.17 ± 0.10 | 60.40 ± 3.01 | **94.42 ± 0.21** | 93.65 ± 0.62 |
| | Enron | 78.97 ± 1.59 | 73.97 ± 3.00 | 66.67 ± 1.07 | 78.76 ± 1.69 | 86.30 ± 0.56 | 82.61 ± 0.61 | 75.55 ± 0.81 | 89.62 ± 0.27 | 74.61 ± 1.64 | 89.67 ± 0.27 | **93.02 ± 7.25** |
| | MOOC | 80.57 ± 0.52 | 80.50 ± 0.68 | 85.28 ± 0.30 | 88.01 ± 1.48 | 81.32 ± 0.42 | 82.28 ± 0.99 | 81.38 ± 0.17 | 87.05 ± 0.51 | 64.99 ± 2.24 | 88.64 ± 0.08 | **98.49 ± 0.48** |
| | Reddit | 96.43 ± 0.16 | 95.89 ± 0.26 | 97.13 ± 0.04 | 97.41 ± 0.12 | 98.62 ± 0.01 | 95.01 ± 0.18 | 95.24 ± 0.08 | 98.83 ± 0.02 | 80.07 ± 2.53 | 98.97 ± 0.01 | **99.28 ± 0.03** |
| | Wikipedia | 94.91 ± 0.32 | 92.21 ± 0.29 | 96.26 ± 0.12 | 97.81 ± 0.18 | 98.27 ± 0.02 | 97.48 ± 0.06 | 96.61 ± 0.04 | 98.58 ± 0.01 | 93.58 ± 0.65 | 98.77 ± 0.03 | **99.19 ± 0.09** |
| | UCI | 79.73 ± 1.48 | 58.39 ± 2.38 | 79.10 ± 0.49 | 87.81 ± 1.32 | 92.61 ± 0.35 | 84.19 ± 1.67 | 91.17 ± 0.29 | 94.45 ± 0.13 | 49.78 ± 5.02 | **94.76 ± 0.19** | 89.12 ± 1.02 |
| | Social Evo. | 91.72 ± 0.66 | 89.10 ± 1.90 | 91.47 ± 0.10 | 90.74 ± 1.40 | 79.83 ± 0.14 | 92.51 ± 0.11 | 91.89 ± 0.05 | **93.05 ± 0.10** | Timeout | 93.13 ± 0.05 | 97.56 ± 0.45 |
| | **Avg. Rank** | 7.29 | 9.00 | 8.00 | 6.14 | 5.29 | 6.57 | 6.71 | 2.86 | 10.57 | **1.71** | 1.86 |
| INS | LastFM | 71.37 ± 3.45 | 69.75 ± 2.73 | 76.26 ± 0.34 | 68.47 ± 6.07 | 71.28 ± 0.43 | 68.79 ± 0.93 | 76.27 ± 0.37 | 75.07 ± 1.45 | 55.60 ± 3.91 | 76.76 ± 0.43 | **94.08 ± 0.57** |
| | Enron | 66.99 ± 1.15 | 62.64 ± 2.33 | 59.95 ± 1.00 | 64.51 ± 1.66 | 60.61 ± 0.63 | 68.93 ± 1.34 | 71.71 ± 1.33 | 67.21 ± 0.72 | 68.66 ± 2.31 | 68.77 ± 0.60 | **94.56 ± 5.02** |
| | MOOC | 64.67 ± 1.18 | 62.05 ± 2.11 | 77.43 ± 0.81 | 76.81 ± 2.83 | 74.36 ± 0.78 | 75.95 ± 1.46 | 73.87 ± 0.99 | 80.66 ± 0.94 | 57.49 ± 1.34 | 80.75 ± 1.00 | **98.64 ± 0.51** |
| | Reddit | 62.54 ± 0.52 | 61.07 ± 0.86 | 63.96 ± 0.25 | 65.27 ± 0.57 | 64.10 ± 0.22 | 61.45 ± 0.25 | 64.82 ± 0.30 | 65.03 ± 1.20 | 78.35 ± 5.03 | 65.30 ± 1.05 | **99.32 ± 0.03** |
| | Wikipedia | 68.22 ± 0.36 | 61.07 ± 0.82 | 84.19 ± 0.96 | 81.96 ± 0.62 | 62.34 ± 6.79 | 71.46 ± 4.95 | 87.47 ± 0.25 | 57.90 ± 11.05 | 92.61 ± 0.90 | 71.14 ± 2.44 | **99.23 ± 0.09** |
| | UCI | 63.57 ± 2.15 | 52.63 ± 1.87 | 69.77 ± 0.43 | 69.94 ± 0.50 | 63.44 ± 1.52 | 74.39 ± 1.81 | 81.40 ± 0.52 | 70.25 ± 2.02 | 52.31 ± 2.67 | 72.17 ± 2.20 | **90.34 ± 0.74** |
| | Social Evo. | 89.06 ± 1.23 | 87.30 ± 1.55 | 94.24 ± 0.36 | 90.67 ± 2.41 | 80.30 ± 0.21 | 95.94 ± 0.37 | 94.56 ± 0.24 | 96.73 ± 0.11 | Timeout | 96.83 ± 0.56 | **98.15 ± 0.27** |
| | **Avg. Rank** | 7.86 | 9.57 | 6.29 | 6.43 | 8.43 | 5.86 | 4.14 | 5.43 | 7.57 | 3.43 | **1.00** |

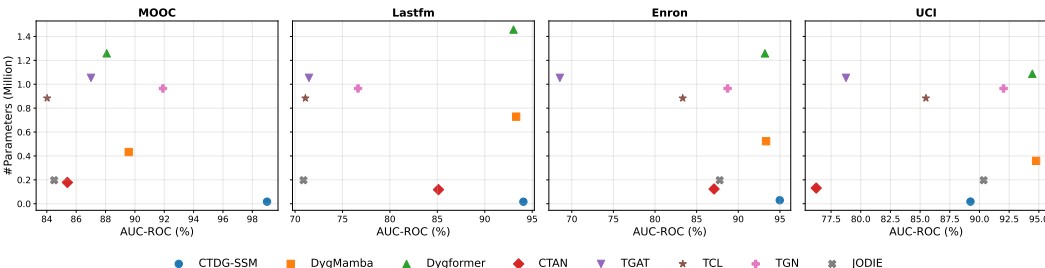

Figure 5: Model size vs. AUC-ROC under a transductive setting with random negative sampling.

# E MODEL EFFICIENCY

## E.1 BATCH LEVEL SUBGRAPH SAMPLING

The discrete update equation involves computing $p(\boldsymbol{L})^{-1}$, which incurs a cost of $\mathcal{O}(N_\tau^3)$, where $N_\tau$ denotes the number of nodes in $\mathcal{G}_\tau$. To implement this update efficiently, we operate on a subset of nodes from $\mathcal{G}_\tau$ whose states are updated, while the remaining node states are kept unchanged. We refer to this subset as the *active batch nodes*. This set includes:

- Nodes appearing in interaction events of the form $(u, v, t)$ within the batch.
- Neighbors of the nodes selected from these interactions.

Neighbor selection depends on the chosen polynomial. For first-order polynomials, we select at most $K$ of the most recent 1-hop neighbors for each node $u$ and $v$ in interaction $(u, v, t)$. For a polynomial of order $m$, we extend this to an $m$-hop neighborhood. All nodes in this $m$-hop ego network are enumerated, and at most $K$ neighbors are chosen based on temporal proximity, using the most recent timestamp along the path. For example, if a node $w$ is connected to $u$ via $v$ through

$$u \xrightarrow{t_1} v \xrightarrow{t_2} w,$$

then the time associated with $w$ when sampling neighbors for interaction $(u, v, t)$ is computed as

$$t_w = (t - t_1) + (t - t_2).$$

The number of active batch nodes $N_B$ for a batch of length $B$ satisfies

$$N_B \le 2BK \ll N_\tau,$$

resulting in a substantial reduction in update cost.

## E.2 LEARNABLE PARAMETERS

In this section, we compare models based on the number of learnable parameters. Recall that the CTDG-SSM layer introduces learnable matrices only through $\bar{\boldsymbol{A}}_{\boldsymbol{L}_B[k]}$, $\bar{\boldsymbol{A}}$ and $\bar{\boldsymbol{B}}(\boldsymbol{L}[k], \boldsymbol{X}[k])$. Figure 5 illustrates the trade-off between parameter count and AUC-ROC. The results show that on

| Models | LastFM | | Enron | | MOOC | | UCI | | Reddit | | Social Evo. | |
|---|---|---|---|---|---|---|---|---|---|---|---|---|
| | Time | Mem | Time | Mem | Time | Mem | Time | Mem | Time | Mem | Time | Mem |
| JODIE | 4.4 | 2.28 | 0.07 | 1.30 | 0.78 | 2.36 | 0.03 | 1.44 | 3.95 | 1.10 | 4.70 | 1.71 |
| DyRep | 6.6 | 2.29 | 0.10 | 1.34 | 0.88 | 2.38 | 0.05 | 1.51 | 5.75 | 1.21 | 7.55 | 1.76 |
| TGAT | 22.75 | 4.15 | 1.28 | 3.46 | 4.08 | 3.64 | 0.60 | 3.42 | 16.33 | 2.98 | 25.50 | 3.89 |
| TGN | 12.14 | 2.21 | 0.15 | 1.45 | 1.03 | 2.54 | 0.08 | 1.51 | 2.05 | 1.67 | 3.83 | 1.78 |
| CAWN | 99.00 | 14.92 | 2.62 | 4.03 | 13.45 | 8.02 | 1.95 | 9.40 | 20.16 | 5.89 | 85.66 | 8.14 |
| TCL | 6.23 | 3.04 | 0.30 | 2.51 | 1.00 | 2.49 | 0.13 | 2.00 | 2.25 | 1.82 | 5.05 | 2.48 |
| GraphMixer | 16.35 | 2.78 | 1.20 | 2.23 | 4.02 | 2.40 | 0.73 | 2.19 | 4.92 | 1.57 | 15.50 | 2.71 |
| DyGFormer | 47.00 | 7.57 | 2.73 | 3.23 | 8.32 | 3.35 | 0.62 | 2.30 | 7.00 | 2.42 | 20.00 | 2.77 |
| CTAN | 3.33 | 1.44 | 0.50 | 1.33 | 3.22 | 2.30 | 0.38 | 1.30 | 0.86 | 1.54 | 2.41 | 0.63 |
| DyGMamba | 28.45 | 4.17 | 2.05 | 2.74 | 4.88 | 2.48 | 0.60 | 1.93 | 6.30 | 2.07 | 17.80 | 2.59 |
| CTDG-SSM | 4.45 | 1.15 | 0.55 | 0.86 | 1.25 | 0.43 | 0.17 | 0.31 | 1.95 | 1.18 | 9.57 | 5.22 |

Table 10: Per-epoch time (minutes) and GPU memory usage (GB) across multiple datasets.

| Models | Enron | | UCI | | Reddit | |
|---|---|---|---|---|---|---|
| | #Epoch | $\mathbf{T}_{tot}$ | #Epoch | $\mathbf{T}_{tot}$ | #Epoch | $\mathbf{T}_{tot}$ |
| CTAN | 173.00 | 86.50 | 236.00 | 89.68 | 327.18 | 173.41 |
| DyGFormer | 32.80 | 89.54 | 34.80 | 21.58 | 24.60 | 104.30 |
| DyGMamba | 33.00 | 67.65 | 28.00 | 16.80 | 26.80 | 88.98 |
| CTDG-SSM | 83.00 | **45.65** | 38.00 | **6.46** | 27.00 | **52.65** |

Table 11: Number of epochs and total time (minutes) across datasets.

long-range datasets such as MOOC and Enron, the proposed model achieves superior performance while being highly parameter-efficient, requiring about one-tenth fewer parameters compared to existing approaches.

### E.3 RUNTIME ANALYSIS

In this section, we compare the proposed model with state-of-the-art approaches using run-time as the performance metric. In Table 10 we report the per-epoch training time (in minutes) and GPU memory consumption (in GB) across all datasets. Notably, it can be observed that `CTDG-SSM` achieves significantly lower per-epoch training time and memory usage compared to `DyGMamba` and `DyGFormer`, both of which are specifically designed for long-range propagation tasks.

In Table 11 we present the total training time, obtained as the product of the per-epoch time and the number of training epochs. In Fig. 6, we analyze the convergence behavior of proposed algorithm where we show the training loss across epochs for multiple datasets. The plots clearly show that the proposed model converges within only a few epochs highlighting its computational efficiency.

### E.4 ROBUSTNESS TO STRUCTURAL PERTURBATIONS

We evaluate the robustness of the proposed algorithm to structural perturbations on the Enron dataset with downstream task as link prediction. In particular, we introduce the perturbations to the true graph as $\bar{\boldsymbol{L}}_B[k] = \boldsymbol{L}_B[k] + \epsilon \Delta \boldsymbol{L}_B[k]$, where $\Delta \boldsymbol{L}_B[k]$ is a perturbation matrix whose entries are sampled from a normal distribution, i.e., $[\Delta \boldsymbol{L}_B]_{ij} \sim \mathcal{N}(0, 1)$ and $\epsilon$ controls the noise level.

| Prediction Node | $p(\boldsymbol{L}) = \boldsymbol{I}$ | $p(\boldsymbol{L}) = \alpha_0 \boldsymbol{I} + \alpha_1 \boldsymbol{L}$ | $p(\boldsymbol{L}) = \alpha_0 \boldsymbol{I} + \alpha_1 \boldsymbol{L} + \alpha_2 \boldsymbol{L}^2$ |
|---|---|---|---|
| First | $1.00 \pm 0.00$ | $1.00 \pm 0.00$ | $1.00 \pm 0.00$ |
| Second | $0.51 \pm 0.06$ | $0.97 \pm 0.02$ | $1.00 \pm 0.00$ |
| Third | $0.47 \pm 0.02$ | $0.96 \pm 0.01$ | $1.00 \pm 0.00$ |
| Second-Last | $0.46 \pm 0.01$ | $0.90 \pm 0.01$ | $0.92 \pm 0.07$ |
| Last | $0.45 \pm 0.02$ | $0.88 \pm 0.18$ | $0.90 \pm 0.06$ |

Table 12: Ablation study with respect to the order of the graph filter.

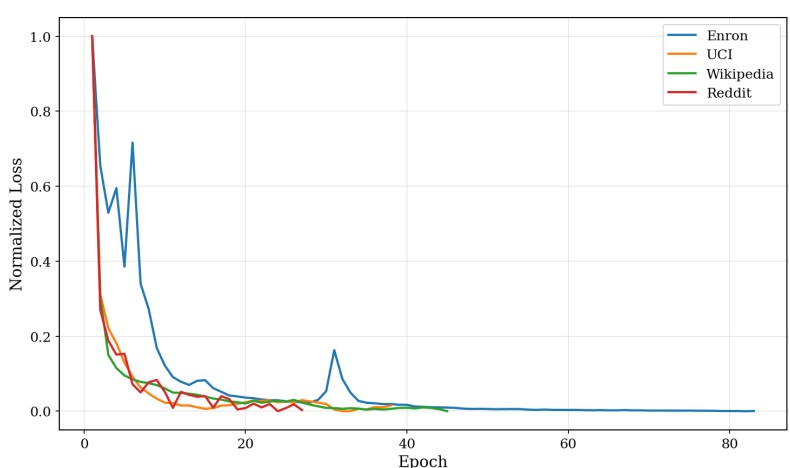

Figure 6: Convergence behavior of `CTDG-SSM` across the datasets

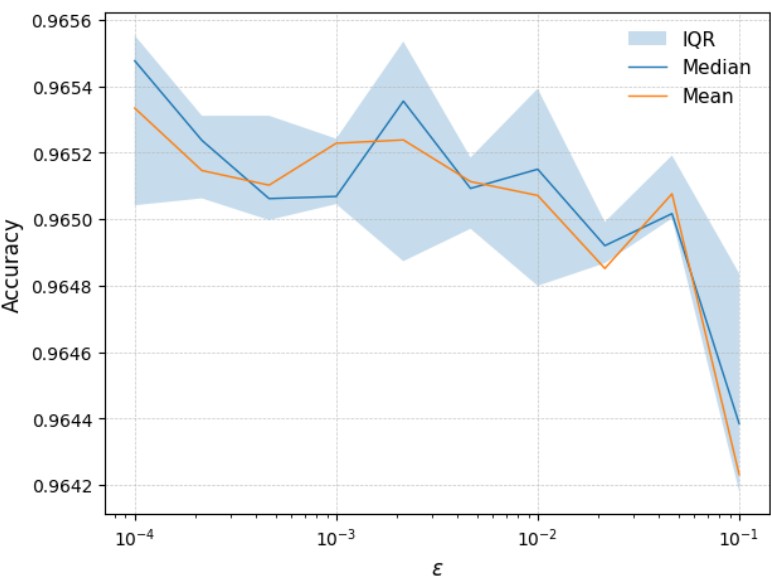

Figure 7: CTDG-SSM link prediction accuracy under noise insertion of form $\boldsymbol{L}_B[k] + \epsilon \frac{\Delta \boldsymbol{L}}{\|\Delta \boldsymbol{L}\|_2}$ at each update.

We then evaluate the proposed algorithm by replacing $\boldsymbol{L}_B[k]$ with $\bar{\boldsymbol{L}}_B[k]$ under different values of $\epsilon$, thereby varying the severity of the structural perturbation. In Fig. 7, we report the accuracy across these noise levels. As expected, accuracy decreases as the noise variance increases; however, for small values of $\epsilon$, the model performs very close to the noise-free setting. This demonstrates that the proposed approach is stable and robust under mild structural perturbations.

### E.5  ABLATION TO STUDY THE IMPORTANCE OF GRAPH FILTERS AND CTDG-SSM MODULE

Considering the downstream task as sequence classification, we conduct an ablation study to understand which components of the model capture long-range information. Specifically, we analyze the role of graph filters and the proposed CTDG-SSM module on the long-range spatial (`LRS`) task and the long-range temporal (`LRT`) task.

To evaluate the `LRT` capabilities of CTDG-SSM, we first set the polynomial $p(\boldsymbol{L}) = \mathbf{I}$. For an event of the form $(u, v, t, x_u, x_v)$ in sequence classification, instead of restricting the model to update only a small subset of nodes (i.e., those in the batch subgraph), we update the state vectors of all nodes. Formally, we define the input signal at time $t$ as:

$$\boldsymbol{X}(t) \in \mathbb{R}^{N_\tau \times 1} \quad \text{such that} \quad \boldsymbol{X}[u](t) = x_u, \boldsymbol{X}[v](t) = x_v, \text{and } 0 \text{ otherwise.}$$

This eliminates the step where the previous states of inactive nodes are carried forward through memory. This carry-forward mechanism could aid the model in LRT, so by removing it, we can evaluate the LRT capability of CTDG-SSM in isolation. In this setup, the model is tasked with predicting the initial feature $\boldsymbol{X}[0](0)$ observed at the first node using the final state vectors of different nodes. A successful LRT would yield strong performance as long as the state vector of node 1 preserves the information of the feature $\boldsymbol{X}[0](0)$. Notably, this model completely lacks LRS capability, as it does not account for the underlying graph structure and updates states solely based on the input at the corresponding nodes.

Next, we evaluate the effect of aggregating multi-hop information by applying graph filters of different orders. In Table 12, we report the mean accuracies obtained from representations at different nodes using filter orders 1and 2. We observe a substantial improvement in prediction accuracy by leveraging the representations from the node $2, \ldots, 31$(the last node), demonstrating the model's enhanced ability to preserve spatial information over longer ranges. In particular, using deeper aggregation-i.e., a filter of order 2-yields a notable gain in accuracy, indicating that incorporating information from larger hop neighborhoods significantly strengthens the model's capacity to capture long-range spatial dependencies.

## F  ADDITIONAL EXPERIMENTS

In this section, we present results on additional temporal datasets-Flights, Contacts, UN Trade, UN Vote, and CanParl (Yu et al., 2023)-using link prediction as the downstream task. We further compare the proposed method with several state-of-the-art approaches, including `Edgebank`, `DyG-Mamba` (Li et al., 2024a), and `FreeDyG` (Tian et al., 2024).

In Tables 14, 15, 16, and 17 we compare the performance of proposed model against the state of the art methods with across these datasets. It is clear that the proposed model consistently outperforms competing methods on most datasets, which we attribute to its ability to jointly model structural and temporal evolution through graph filters and state-space dynamics.

Additionally, in Tables 13 and 18, we provide direct comparisons against `Edgebank`, `DyG-Mamba`, and `FreeDyG`. The results clearly demonstrate that our model consistently achieves superior performance across both transductive and inductive settings.

## G  CTDG-SSM BEYOND NODE/EDGE ADDITION.

The CTDG-SSM state update equation depends on the change in the graph Laplacian, and therefore naturally accommodates both the addition *and* removal of edges.

| NSS | Datasets | Edgebank | DyG-Mamba | FreeDyG | CTDG-SSM |
|---|---|---|---|---|---|
| rnd | Wiki | 90.37±0.00 | 99.08 ± 0.09 | 99.26 ± 0.01 | **99.40 ± 0.00** |
| | Reddit | 94.86±0.00 | 99.27 ± 0.00 | 99.48 ± 0.01 | **99.53 ± 0.00** |
| | MOOC | 57.97±0.00 | 90.25 ± 0.01 | 89.61 ± 0.10 | **98.85 ± 0.00** |
| | LastFM | 79.29±0.00 | **94.23 ± 0.01** | 92.15 ± 0.16 | 93.40±0.49 |
| | Enron | 83.53 ± 0.00 | 93.14 ± 0.08 | 92.51 ± 0.05 | **94.46 ± 4.73** |
| | Social Evo. | 74.95 ± 0.00 | 94.77 ± 0.01 | 94.91 ± 0.01 | **98.65 ± 0.65** |
| | UCI | 76.20 ± 0.00 | 96.14 ± 0.14 | **96.28 ± 0.11** | 90.18 ±0.98 |
| hist | Wiki | 73.35 ± 0.00 | 82.35 ±1.25 | 91.59 ± 0.57 | **98.99 ± 0.32** |
| | Reddit | 73.59 ± 0.00 | 81.02 ± 0.19 | 85.67 ± 1.01 | **97.55 ± 0.22** |
| | MOOC | 60.71 ± 0.00 | 87.42 ± 1.57 | 86.71 ± 0.81 | **94.76 ± 1.76** |
| | LastFM | 73.03 ± 0.00 | 84.08 ± 0.45 | 79.71 ± 0.51 | **88.91 ± 0.93** |
| | Enron | 76.53 ± 0.00 | 77.85 ± 1.20 | 78.87 ± 0.82 | **95.80 ± 3.33** |
| | Social Evo. | 80.57 ± 0.00 | 97.35 ± 0.18 | 77.79 ± 0.23 | **98.20 ± 0.81** |
| | UCI | 65.50 ± 0.00 | 81.36 ± 0.14 | 86.10 ± 1.19 | **88.87 ± 1.28** |
| ind | Wiki | 80.63 ± 0.00 | 87.06 ± 0.86 | 90.05 ± 0.79 | **99.45 ± 0.06** |
| | Reddit | 85.48 ±0.00 | 91.77 ±0.46 | 90.74 ± 0.17 | **99.58 ± 0.02** |
| | MOOC | 49.43 ±0.00 | 81.19 ± 2.02 | 83.01 ± 0.87 | **99.03 ± 0.38** |
| | LastFM | 75.49 ± 0.00 | 75.05 ± 0.40 | 72.19 ± 0.24 | **93.81 ± 0.44** |
| | Enron | 73.89 ± 0.00 | 77.46 ± 0.90 | 77.81 ± 0.65 | **95.81 ± 2.99** |
| | Social Evo. | 83.69±0.00 | 97.78 ± 0.15 | 97.50 ± 0.15 | **98.88 ± 0.63** |
| | UCI | 57.43 ±0.00 | 77.75 ± 1.56 | 82.35 ± 0.73 | **91.44 ±0.50** |

Table 13: Performance comparison with AP on dynamic link prediction under transductive setting.

| NSS | Dataset | JODIE | DyRep | TGAT | TGN | CAWN | Edgebank | TCL | GraphMixer | DyGFormer | DyGMamba | CTDG-SSM |
|---|---|---|---|---|---|---|---|---|---|---|---|---|
| rnd | Flights | 95.60±1.73 | 95.29±0.72 | 94.03±0.18 | 97.95±0.14 | 98.51±0.01 | 89.35±0.00 | 91.23±0.02 | 90.99±0.05 | 98.91±0.01 | **98.95±0.05** | 98.70±0.05 |
| | Can. Parl. | 69.26 ± o.31 | 66.54 ±2.76 | 70.73 ±0.72 | 70.88 ± 2.34 | 69.82 ±2.34 | 64.55 ±0.00 | 68.67 ±2.67 | 77.04 ±0.46 | 97.36±0.45 | **99.57±0.08** | 98.20 ±1.73 |
| | US Legis. | 75.05 ± 1.52 | 75.34 ± 0.39 | 68.52±3.16 | 75.99 ± 0.58 | 70.58 ± 0.48 | 58.39 ± 0.00 | 69.59 ± 0.48 | 70.74 ± 1.02 | 71.11 ± 0.59 | 71.75 ± 0.26 | **82.51 ± 0.00** |
| | UN Trade | 64.94 ± 0.31 | 63.21 ± 0.93 | 61.47 ± 0.18 | 65.03 ± 1.37 | 65.39 ± 0.12 | 60.41 ± 0.00 | 62.21 ± 0.03 | 62.21 ± 0.27 | 66.46 ± 1.29 | 67.50±0.14 | **69.10±0.20** |
| | UN Vote | 63.91 ± 0.81 | 62.81±0.80 | 52.21 ± 0.98 | 65.72 ± 2.17 | 52.84 ± 0.10 | 58.49 ± 0.00 | 51.90 ± 0.30 | 52.11 ± 0.16 | 55.55±0.42 | 56.39±0.18 | **95.31 ± 0.01** |
| | Contact | 95.31±1.33 | 95.98±0.15 | 96.28±0.09 | 96.89±0.56 | 90.26±0.28 | 92.58±0.00 | 92.44±0.12 | 91.92±0.03 | 98.29±0.01 | 98.43±0.12 | **98.90 ± 0.05** |
| hist | Flights | 66.48±2.59 | 67.61±0.99 | 72.38± 0.18 | 66.70±1.64 | 64.72±0.97 | 70.53±0.00 | 70.68± 0.24 | 71.47±0.26 | 66.59±0.49 | 67.80±2.17 | **87.2 ± 1.50** |
| | Can. Parl. | 51.79±0.63 | 63.31±1.23 | 67.13±0.84 | 68.42±3.07 | 66.53±2.77 | 63.84±0.00 | 65.93 ±3.00 | 74.34 ±0.87 | 97.00±0.31 | **99.77±1.00** | 97.8 ± 1.24 |
| | US Legis. | 51.71 ±5.76 | 86.88 ± 2.25 | 62.14 ± 6.60 | 74.00 ±7.57 | 68.82 ± 8.23 | 63.22±0.00 | 80.53 ± 3.95 | 81.65 ± 1.02 | 85.30±3.88 | **86.12±0.26** | 80.02 ± 0.00 |
| | UN Trade | 61.39 ± 1.83 | 59.19 ± 0.17 | 55.74 ± 0.91 | 58.44 ± 5.51 | 55.71 ± 0.38 | 81.32 ± 0.00 | 55.90 ± 1.17 | 57.05± 1.22 | 64.41±1.40 | 66.10±1.02 | **68.4 ± 0.04** |
| | UN Vote | 70.02 ± 0.81 | 69.30± 1.12 | 52.96 ± 2.14 | 69.37 ± 3.93 | 51.26 ± 0.04 | 84.89 ± 0.00 | 52.30 ± 2.35 | 51.20 ± 1.60 | 60.84±1.58 | 61.07±1.39 | **95.29±0.01** |
| | Contact | 95.31 ± 2.13 | 96.39 ± 0.20 | 96.05 ± 0.52 | 93.05 ± 2.35 | 84.16 ± 0.49 | 88.81 ± 0.00 | 93.86 ± 0.01 | 93.36± 0.41 | 97.57 ± 0.06 | 97.61±0.04 | **98.2 ± 0.05** |
| ind | Flights | 69.07 ± 4.02 | 70.57 ± 1.82 | 75.48 ± 0.26 | 71.09 ± 2.72 | 69.18 ± 1.52 | 81.08 ± 0.00 | 74.62 ± 0.18 | 74.87 ± 0.21 | 70.92±1.78 | 73.79±5.69 | **86.50±1.34** |
| | Can. Parl. | 48.42 ± 0.66 | 58.61± 0.86 | 68.82 ± 1.21 | 65.34±2.87 | 67.75±1.00 | 62.16±0.00 | 65.85 ± 1.75 | 69.48 ± 0.63 | **95.44 ± 0.57** | 94.87±0.67 | 94.2 ± 0.50 |
| | US Legis. | 50.27±5.13 | 83.44±1.16 | 61.91 ± 5.82 | 67.57±6.47 | 65.81± 8.52 | 65.74 ± 0.00 | 78.15 ± 3.34 | 79.63 ± 0.84 | 81.25±3.62 | 81.22±1.34 | **81.32 ± 0.00** |
| | UN Trade | 60.42 ± 1.48 | 60.19 ± 1.24 | 60.61 ± 1.24 | 61.04 ± 6.01 | 62.54 ± 0.67 | 72.97±0.00 | 61.06±1.74 | 60.15 ±1.29 | 55.79±1.02 | 58.89±0.59 | **67.92 ± 0.5** |
| | UN Vote | 67.79 ± 1.46 | 67.53 ± 1.98 | 52.89 ± 1.61 | 67.63 ± 2.67 | 52.19 ± 0.34 | 66.30 ± 0.00 | 50.62 ± 0.82 | 51.60 ± 0.73 | 51.91±0.84 | 52.24±0.95 | **95.37 ± 0.01** |
| | Contact | 93.43 ± 1.78 | 94.18 ± 0.10 | 94.35± 0.48 | 90.18 ± 3.28 | 89.31 ± 0.27 | 85.20 ± 0.00 | 91.35 ± 0.21 | 90.87 ± 0.35 | 94.75±0.28 | 95.43±0.17 | **97.60 ± 0.32** |

Table 14: AP for dynamic link prediction under transductive setting

| NSS | Dataset | JODIE | DyRep | TGAT | TGN | CAWN | EdgeBank | TCL | GraphMixer | DyGFormer | DyGMamba | CTDG-SSM |
|---|---|---|---|---|---|---|---|---|---|---|---|---|
| rnd | Flights | 96.21 ± 1.42 | 95.95 ± 0.62 | 94.13 ± 0.17 | 98.22 ± 0.13 | 98.45 ± 0.01 | 90.23 ± 0.00 | 91.21 ± 0.02 | 91.13 ± 0.01 | 98.93 ± 0.01 | **98.98 ± 0.05** | 98.53 ± 0.02 |
| | Can. Parl. | 78.21 ± 0.23 | 73.35 ± 3.67 | 75.69 ±0.78 | 76.99 ± 1.80 | 75.70 ± 3.27 | 64.14 ± 0.00 | 72.46 ± 3.23 | 83.17 ± 0.53 | 97.76 ± 0.41 | **99.69 ± 0.06** | 98.1 ± 0.80 |
| | US Legis. | 82.85 ± 1.07 | 82.28 ± 0.32 | 75.84 ± 1.99 | 83.34 ± 0.43 | 77.16 ± 0.39 | 62.57 ± 0.00 | 76.27 ± 0.63 | 76.96 ± 0.79 | 77.90 ± 0.58 | 79.03 ± 0.26 | **85.92 ± 0.00** |
| | UN Trade | 69.62 ± 0.44 | 67.44 ± 0.83 | 64.01 ± 0.12 | 69.10 ± 1.67 | 68.54 ± 0.18 | 66.75 ± 0.00 | 64.72 ± 0.05 | 65.52 ± 0.51 | 70.20 ± 1.44 | 71.41 ± 0.21 | **73.76 ± 0.30** |
| | UN Vote | 68.53 ± 0.95 | 67.18 ± 1.04 | 52.83 ± 1.12 | 68.71 ± 2.65 | 53.09 ± 0.22 | 62.97 ± 0.00 | 51.88 ± 0.36 | 52.46 ± 0.27 | 58.48 ± 0.12 | **97.41 ± 0.00** | 97.41 ± 0.00 |
| | Contact | 96.66 ± 0.89 | 96.48 ± 0.14 | 96.95 ± 0.08 | 97.54 ± 0.35 | 89.99 ± 0.34 | 94.34 ± 0.00 | 94.15 ± 0.09 | 93.94 ± 0.02 | 98.53 ± 0.01 | 98.68 ± 0.02 | **98.70 ± 0.01** |
| hist | Flights | 68.97 ± 1.87 | 69.43 ± 0.90 | 72.20 ± 0.16 | 68.39 ± 0.95 | 66.11 ± 0.71 | 74.64 ± 0.00 | 70.57 ± 3.01 | 70.37 ± 0.23 | 68.09 ± 0.43 | 68.98 ± 1.81 | **90.1 ± 1.50** |
| | Can. Parl. | 62.44 ± 1.11 | 70.16 ± 1.70 | 70.86 ± 0.94 | 73.23 ± 3.08 | 72.06 ± 3.94 | 63.04 ± 0.00 | 69.95 ± 3.70 | 79.03 ± 1.01 | 97.61 ± 0.40 | **99.82 ± 0.10** | 97.5 ± 0.05 |
| | US Legis. | 67.47 ± 6.40 | **91.44 ± 1.18** | 73.47 ± 5.25 | 83.53 ± 4.53 | 78.62 ± 7.46 | 67.41 ± 0.00 | 83.97 ± 3.71 | 85.17 ± 0.70 | 90.77 ± 1.96 | 88.36 ± 1.78 | 85.55 ± 0.00 |
| | UN Trade | 68.92 ± 1.40 | 64.36 ± 1.40 | 60.37 ± 0.68 | 63.93 ± 5.41 | 63.09 ± 0.74 | **86.61 ± 0.00** | 61.43 ± 1.04 | 63.20 ± 1.54 | 73.86 ± 1.13 | 74.10 ± 2.02 | 73.3 ± 0.7 |
| | UN Vote | 76.84 ± 1.01 | 74.72 ± 1.43 | 53.95 ± 3.15 | 73.40 ± 5.20 | 51.27 ± 0.03 | 89.62 ± 0.00 | 52.29 ± 2.39 | 52.61 ± 1.44 | 64.27 ± 1.78 | 65.17 ± 1.24 | **97.22 ± 0.00** |
| | Contact | 96.35 ± 0.92 | 96.00 ± 0.23 | 95.39 ± 0.43 | 93.76 ± 1.29 | 83.06 ± 0.32 | 92.17 ± 0.00 | 93.34 ± 0.19 | 94.14 ± 0.34 | 97.17 ± 0.05 | 97.27 ± 0.06 | **97.78 ± 0.04** |
| ind | Flights | 69.99 ± 3.10 | 71.13 ± 1.55 | 73.47 ± 0.18 | 71.63 ± 1.72 | 69.70 ± 0.75 | 81.10 ± 0.00 | 72.54 ± 0.19 | 72.21 ± 0.21 | 69.53 ± 1.17 | 71.16 ± 3.24 | **89.25 ± 1.24** |
| | Can. Parl. | 52.88 ± 0.80 | 63.53 ± 0.65 | 72.47 ± 1.18 | 69.57 ± 2.81 | 72.93 ± 1.78 | 61.41 ± 0.00 | 69.47 ± 2.12 | 70.52 ± 0.94 | 96.70 ± 0.59 | **99.82 ± 0.10** | 96.87 ± 0.05 |
| | US Legis. | 59.05 ± 5.52 | 89.44 ± 0.71 | 71.62 ± 5.42 | 78.12 ± 4.46 | 76.45 ± 7.02 | 68.66 ± 0.00 | 82.54 ± 3.91 | 84.22 ± 0.91 | **87.96 ± 1.80** | 86.08 ± 2.27 | 86.06 ± 0.00 |
| | UN Trade | 66.82 ± 1.27 | 65.60 ± 1.28 | 66.13 ± 0.78 | 66.37 ± 5.39 | 71.73 ± 0.74 | 74.06 ± 0.00 | 67.80 ± 1.21 | 62.56 ± 1.51 | 67.60 ± 0.64 | 72.65 ± 0.45 | — |
| | UN Vote | 73.73 ± 1.61 | 72.80 ± 2.16 | 53.04 ± 2.58 | 72.69 ± 3.72 | 52.75 ± 0.90 | 72.85 ± 0.00 | 52.02 ± 1.22 | 52.56 ± 1.51 | 51.89 ± 0.74 | 53.37 ± 1.26 | **97.45 ± 0.01** |
| | Contact | 94.47 ± 1.08 | 94.23 ± 0.18 | 94.10 ± 0.41 | 91.64 ± 1.72 | 87.68 ± 0.24 | 85.87 ± 0.00 | 91.23 ± 0.19 | 90.96 ± 0.27 | 95.01 ± 0.15 | 95.68 ± 0.20 | **97.50 ± 0.45** |

Table 15: AUC-ROC for dynamic link prediction under transductive setting

| NSS | Dataset | JODIE | DyRep | TGAT | TGN | CAWN | TCL | GraphMixer | DyGFormer | DyGMamba | CTDG-SSM |
|---|---|---|---|---|---|---|---|---|---|---|---|
| rnd | Flights | 94.74 ± 0.37 | 92.88 ± 0.73 | 88.73 ± 0.33 | 95.03 ± 0.60 | 97.06 ± 0.02 | 83.41 ± 0.07 | 83.03 ± 0.05 | 97.79 ± 0.02 | **97.85 ± 0.22** | 97.15±0.04 |
| | Can. Parl. | 53.92 ± 0.94 | 54.02 ± 0.76 | 55.18 ± 0.79 | 54.10 ± 0.93 | 55.80 ± 0.69 | 54.30 ± 0.66 | 55.91 ± 0.82 | 87.74 ± 0.71 | **93.46 ± 2.62** | 88.65 ± 0.70 |
| | US Legis. | 54.93 ± 2.29 | 57.28 ± 0.71 | 51.00 ± 3.11 | 58.63 ± 0.37 | 53.17 ± 1.20 | 52.59 ± 0.97 | 50.71 ± 0.76 | 54.28 ± 2.87 | 55.95 ± 1.16 | **76.94 ± 0.01** |
| | UN Trade | 59.65 ± 0.77 | 57.02 ± 0.69 | 61.03 ± 0.18 | 58.31 ± 3.15 | 65.24 ± 0.21 | 62.21 ± 0.12 | 62.17 ± 0.31 | 64.55 ± 0.62 | 70.55 ± 0.04 | **72.42 ± 0.02** |
| | UN Vote | 56.64 ± 0.96 | 54.62 ± 2.22 | 52.24 ± 1.46 | 58.85 ± 2.51 | 49.94 ± 0.45 | 51.60 ± 0.97 | 50.68 ± 0.44 | 55.93 ± 0.39 | 56.61 ± 0.13 | **95.79±0.01** |
| | Contact | 94.34 ± 1.45 | 92.18 ± 0.41 | 95.87 ± 0.11 | 93.82 ± 0.99 | 89.55 ± 0.30 | 91.11 ± 0.12 | 90.59 ± 0.05 | 98.03 ± 0.02 | 98.16 ± 0.03 | **98.42 ± 0.01** |
| ind | Flights | 61.01 ± 1.66 | 62.83 ± 1.31 | 64.72 ± 0.37 | 59.32 ± 1.45 | 56.82 ± 0.56 | 64.50 ± 0.25 | 65.29 ± 0.24 | 57.11 ± 0.20 | 57.76 ± 2.06 | **92.24 ± 1.05** |
| | Can. Parl. | 52.58 ± 0.86 | 52.24 ± 0.28 | 56.46 ± 0.50 | 54.18 ± 0.73 | 57.06 ± 0.08 | 55.46 ± 0.69 | 55.76 ± 0.65 | 87.22 ± 0.82 | **92.68 ± 0.97** | 88.42 ± 0.65 |
| | US Legis. | 52.94 ± 2.11 | 62.10 ± 1.41 | 51.83 ± 3.95 | 61.18 ± 1.10 | 55.56 ± 1.71 | 53.87 ± 1.41 | 52.03 ± 1.02 | 56.31 ± 3.46 | 57.85 ± 0.23 | **75.64 ± 0.01** |
| | UN Trade | 55.43 ± 1.20 | 55.42 ± 0.87 | 55.58 ± 0.68 | 52.80 ± 3.24 | 54.97 ± 0.38 | 55.66 ± 0.98 | 54.88 ± 1.01 | 52.56 ± 1.70 | 52.81 ± 0.18 | **69.24 ± 1.02** |
| | UN Vote | 61.17 ± 1.33 | 60.29 ± 1.79 | 53.08 ± 3.10 | 63.71 ± 2.97 | 48.01 ± 0.82 | 54.13 ± 2.16 | 48.10 ± 0.40 | 52.61 ± 1.25 | 53.70 ± 2.40 | **95.77 ± 0.00** |
| | Contact | 90.43 ± 2.33 | 89.22 ± 0.65 | 94.14 ± 0.45 | 88.12 ± 1.50 | 74.19 ± 0.81 | 90.43 ± 0.17 | 89.91 ± 0.36 | 93.55 ± 0.52 | 94.05 ± 0.32 | **96.78± 0.72** |

Table 16: AP for dynamic link prediction under inductive setting

| NSS | Dataset | **JODIE** | DyRep | TGAT | TGN | CAWN | TCL | GraphMixer | DyGFormer | DyGMamba | CTDG-SSM |
|---|---|---|---|---|---|---|---|---|---|---|---|
| rnd | Flights | 95.21 ± 0.32 | 93.56 ± 0.70 | 88.64 ± 0.35 | 95.92 ± 0.43 | 96.86 ± 0.02 | 82.48 ± 0.01 | 97.80 ± 0.06 | 99.33 ± 0.48 | **97.98 ± 0.25** | 97.36±0.04 |
| | Can. Parl. | 53.81 ± 1.14 | 55.27 ± 0.49 | 56.51 ± 0.75 | 55.86 ± 0.75 | 58.83 ± 1.13 | 55.83 ± 1.07 | 58.32 ± 1.08 | 89.33 ± 0.48 | **94.02 ± 3.42** | 89.78±0.78 |
| | US Legis. | 58.12 ± 2.35 | 61.07 ± 0.56 | 48.27 ± 3.50 | 62.38 ± 0.48 | 51.49 ± 1.13 | 50.43 ± 1.48 | 47.20 ± 0.89 | 53.21 ± 3.04 | 57.17 ± 0.20 | **81.17 ± 0.00** |
| | UN Trade | 62.28 ± 0.50 | 58.82 ± 0.98 | 62.72 ± 0.12 | 59.99 ± 3.50 | 67.05 ± 0.21 | 63.76 ± 0.07 | 63.48 ± 0.37 | 67.25 ± 1.05 | 68.26 ± 0.26 | **73.76±0.45** |
| | UN Vote | 58.13 ± 1.43 | 55.13 ± 3.46 | 51.83 ± 1.35 | 61.23 ± 2.71 | 48.34 ± 0.76 | 50.51 ± 1.05 | 50.04 ± 0.86 | 56.73 ± 0.69 | 56.91 ± 0.12 | **97.72±0.01** |
| | Contact | 95.37 ± 0.92 | 91.89 ± 0.38 | 96.53 ± 0.10 | 94.84 ± 0.75 | 89.07 ± 0.34 | 93.05 ± 0.09 | 92.83 ± 0.05 | 98.30 ± 0.02 | 98.44±0.05 | **98.70 ± 0.65** |
| ind | Flights | 60.72 ± 1.29 | 61.99 ± 1.39 | 63.40 ± 0.26 | 59.66 ± 1.05 | 56.58 ± 0.44 | 63.49 ± 0.23 | 63.32 ± 0.09 | 57.35 ± 0.20 | 56.58±2.12 | **91.36 ± 1.87** |
| | Can. Parl. | 51.61 ± 0.98 | 52.35 ± 0.52 | 58.15 ± 0.62 | 55.43 ± 0.42 | 60.01 ± 0.47 | 56.88 ± 0.93 | 56.63 ± 1.09 | 88.51 ± 0.73 | **92.37 ± 0.18** | 89.56 ± 0.69 |
| | US Legis. | 58.12 ± 2.94 | 67.94 ± 0.98 | 49.99 ± 4.88 | 64.87 ± 1.65 | 54.41 ± 1.31 | 52.12 ± 2.13 | 49.28 ± 0.86 | 56.57 ± 3.22 | 57.91 ± 3.41 | **81.46 ± 0.01** |
| | UN Trade | 58.71 ± 1.20 | 57.87 ± 1.36 | 59.98 ± 0.59 | 55.62 ± 3.59 | 60.88 ± 0.79 | 61.01 ± 0.93 | 59.71 ± 1.17 | 57.28 ± 3.06 | 57.58 ± 0.20 | **71.43 ± 0.04** |
| | UN Vote | 65.29 ± 1.30 | 64.10 ± 2.10 | 51.78 ± 4.14 | 68.58 ± 3.08 | 48.04 ± 1.76 | 54.65 ± 2.20 | 45.57 ± 0.41 | 53.87 ± 2.01 | 54.83 ± 2.17 | **97.73 ± 0.01** |
| | Contact | 90.80 ± 1.18 | 88.87 ± 0.67 | 93.76 ± 0.40 | 88.85 ± 1.39 | 74.79 ± 0.38 | 90.37 ± 0.16 | 90.04 ± 0.29 | 94.14 ± 0.26 | 94.35 ± 0.29 | **96.98 ± 0.56** |

Table 17: AUC-ROC for dynamic link under inductive setting.

| NSS | Datasets | Edgebank | DyG-Mamba | FreeDyG | CTDG-SSM |
|---|---|---|---|---|---|
| rnd | Wiki | N/A | 98.65 ± 0.03 | 98.97 ± 0.01 | **99.19 ± 0.09** |
| | Reddit | N/A | 98.88 ± 0.00 | 98.91 ± 0.01 | **99.28 ± 0.00** |
| | MOOC | N/A | 90.20 ± 0.06 | 87.75 ± 0.62 | **98.49 ± 0.48** |
| | LastFM | N/A | 95.13 ± 0.08 | **94.89 ± 0.01** | 93.65 ± 0.62 |
| | Enron | N/A | 91.14 ± 0.07 | 89.69 ± 0.17 | **93.02 ± 7.25** |
| | Social Evo. | N/A | 93.23 ± 0.01 | 94.76 ± 0.05 | **97.56 ± 0.45** |
| | UCI | N/A | 94.15 ± 0.04 | **94.85 ± 0.10** | 89.12 ± 1.02 |
| ind | Wiki | N/A | 79.44 ± 2.78 | 87.54 ± 0.26 | **99.23 ± 0.09** |
| | Reddit | N/A | 65.61 ± 0.01 | 64.98 ± 0.20 | **99.32 ± 0.03** |
| | MOOC | N/A | 81.67 ± 1.08 | 81.41 ± 0.31 | **98.64 ± 0.51** |
| | LastFM | N/A | 79.60 ± 0.28 | 77.01 ± 0.43 | **94.08 ± 0.57** |
| | Enron | N/A | 68.44 ± 1.85 | 72.85 ± 0.81 | **94.56 ± 5.02** |
| | Social Evo. | N/A | 96.93 ± 0.21 | 96.91 ± 0.12 | **98.15 ± 0.27** |
| | UCI | N/A | 79.27 ± 1.03 | 82.06 ± 0.58 | **90.34 ± 0.74** |

Table 18: Performance comparison with AP on dynamic link prediction under inductive setting.

| Dataset | CTDG-SSM | DyGMamba | DyGFormer | CTAN | TGN |
|---|---|---|---|---|---|
| tgbl-wiki | **0.817 ± 0.027** | 0.739 ± 0.009 | 0.798 ± 0.004 | 0.668 ± 0.007 | 0.396 ± 0.060 |
| tgbl-coin | **0.862 ± 0.003** | — | 0.752 ± 0.004 | 0.748 ± 0.004 | 0.586 ± 0.037 |

Table 19: MRR on the tgbl-wiki and tgbl-coin datasets.

To handle edge deletions within the subgraph, one can simply invert the construction process described in the paper. Specifically, if an edge is removed in batch $B$, we construct batch Laplacian $\boldsymbol{L}_B[k]$ without this edge, while $\boldsymbol{L}_B[k-1]$ includes it. The resulting difference $\boldsymbol{L}_B[k] - \boldsymbol{L}_B[k-1]$ correctly captures the effect of edge removal.

Node deletion can be treated analogously by removing all edges incident to that node. In this case, $\boldsymbol{L}_B[k]$ contains no edges between the removed node and its neighbors, while $\boldsymbol{L}_B[k-1]$ retains these edges. This ensures that the update mechanism captures the effective removal of the node.

## H    LIMITATIONS AND FUTURE RESEARCH DIRECTIONS.

In the current model, we implement a polynomial of the Laplacian using simple graph filters, which provide an efficient linear approximation to the underlying differential operator. While effective, this design restricts the expressiveness of the operator. An important direction for future work is to explore learning these operators and their inverses directly through graph neural networks, potentially enabling more adaptive and data-dependent approximations. Also, the current framework is primarily evaluated on CTDG datasets, where all node and edge features are fully observed. Extending the framework to handle scenarios with missing features in sampled events, or to accommodate interleaved and partially observed dynamic graphs, presents a promising direction for future research.

## I    CTDG-SSM PSEUDO CODE AND TIME COMPLEXITY

The CTDG-SSM model consists of two primary components: the online update and the inference. In this section, we will provide the algorithm for both of these parts.

### I.1    ONLINE UPDATE

From a stream of events, we form a batch of $B$ concurrent events. Using subgraph sampling, we construct the corresponding batch Laplacian $\boldsymbol{L}_B[k]$. By removing the edges associated with the events in the current batch from $\boldsymbol{L}_B[k]$, we obtain the previous-step Laplacian $\boldsymbol{L}_B[k-1]$. Algorithm 1 summarizes this procedure. For active batch nodes $N_B$, state vectors of dimension $d$, and a polynomial of highest order $m$, the state update has a time complexity of $\mathcal{O}(mN_B^3 + dN_B^2)$, it is to be noted that $N_B << N$.

### I.2    INFERENCE

The query provided to the model for a downstream task may take the form $(u, v, t)$, where the model must determine whether this constitutes a valid link or classify node $u$ based on the interaction and its historical context. Alternatively, the query may be of the form $(u)$, in which case the model retrieves the stored state of node $u$ and processes it according to the downstream task. In this section, Algorithm 2 specifies the procedure for link prediction queries, and Algorithm 3 details the procedure for node classification queries. The inference time complexity depends on the task:

- for link prediction and node classification, it is $\mathcal{O}(\deg(u))$ due to the computation of $\Delta t$;
- for sequence classification, it is $\mathcal{O}(1)$ per query.

**Algorithm 1** CTDG-SSM ZOH Update

**Require:** Batch Laplacian $\boldsymbol{L}_B[k]$, Batch Laplacian $\boldsymbol{L}_B[k-1]$, events $\{u_i, v_i, t_i, \boldsymbol{x}_u[t_i], \boldsymbol{x}_v[t_i], \boldsymbol{x}_{u,v}[t_i]\}_{i=1}^{B}$ and batch active node indices $\{\hat{u}_i\}_{i=1}^{N_B}$. learnable polynomial $p_\alpha$, Gaussian quadrature node $\boldsymbol{q}_{\text{nodes}} \in \mathbb{R}^{8\times1}$, $\boldsymbol{q}_{\text{weights}} \in \mathbb{R}^{8\times1}$. State matrices parameters $\boldsymbol{A}_{\log}^{(0)} \in \mathbb{R}^d$, $\boldsymbol{A}_{\log}^{(1)} \in \mathbb{R}^d$, hidden states $\boldsymbol{H}^{(0)}[k]$ and $\boldsymbol{H}^{(1)}[k]$.

1: $\boldsymbol{I} \leftarrow \boldsymbol{I}_{\dim(\boldsymbol{L}[k])}$             {Identity matrix}
2: $\Delta p_\alpha(\boldsymbol{L}_B[k]) \leftarrow p_\alpha(\boldsymbol{L}_B[k]) - p_\alpha(\boldsymbol{L}_B[k-1])$
3: $\bar{\boldsymbol{A}}_{\boldsymbol{L}_B[k]} \leftarrow \exp\left(-p_\alpha(\boldsymbol{L}_B[k])^{-1}\Delta p_\alpha(\boldsymbol{L}_B[k])\right)$
4: LHS $\leftarrow \boldsymbol{O}_{\dim(N_B \times N_B \times 8)}$
5: **for** $i = 0$ to 7 **do**
6:      LHS[:,:,i] $\leftarrow \exp\left(-p_\alpha(\boldsymbol{L}_B[k])^{-1}\Delta p_\alpha(\boldsymbol{L}_B[k])\boldsymbol{q}_{\text{node}}[i]\right)$
7: **end for**
8: Construct SSM Input $\boldsymbol{X}$:
9: $\boldsymbol{N}_{\text{St}} \leftarrow \boldsymbol{0}_{dim(N_\tau \times 2d_s)}$        {Zero matrix, $d_s$: Static embedding dimensions}
10: **for** $i = 0$ to $B - 1$ **do**
11:      $\boldsymbol{N}_{\text{St}}[i, 1{:}d] \leftarrow$ Static-Embedding$(u_i)$
12:      $\boldsymbol{N}_{\text{St}}[i, d{+}1{:}2d] \leftarrow$ Static-Embedding$(v_i)$
13:      $\boldsymbol{N}_{\text{St}}[i + B, 1{:}d] \leftarrow$ Static-Embedding$(v_i)$
14:      $\boldsymbol{N}_{\text{St}}[i + B, d{+}1{:}2d] \leftarrow$ Static-Embedding$(u_i)$
15: **end for**
16: **for** $i = 1$ to $B$ **do**
17:      $\boldsymbol{X}[i,:] \leftarrow [\boldsymbol{x}_u[t_i] \mid \boldsymbol{x}_v[t_i] \mid \boldsymbol{x}_{u,v}[t_i] \mid \phi(\Delta[t_i]) \mid \boldsymbol{N}_{st}[i,:]]$    {$\boldsymbol{V}$ has $N_\tau$ rows}
18:      $\boldsymbol{X}[i + B,:] \leftarrow [\boldsymbol{x}_u[t_i] \mid \boldsymbol{x}_v[t_i] \mid \boldsymbol{x}_{u,v}[t_i] \mid \phi(\Delta[t_i]) \mid \boldsymbol{N}_{st}[i+B,:]]$    {rest filled with 0.}
19: **end for**

20: **CTDG-SSM 1$^{\text{st}}$ Layer**:
21: $\tilde{\boldsymbol{X}}^{(0)}[k] \leftarrow h_\theta(\boldsymbol{X})$             { encoder $h_\theta$}
22: $\boldsymbol{X}_n^{(0)}[k] \leftarrow \text{RMS}_0(\tilde{\boldsymbol{X}}^{(0)}[k])$
23: $\boldsymbol{B}_{x,0} \leftarrow \boldsymbol{B}_0(\boldsymbol{X}_n^{(0)}[k])$
24: $\Delta_0 \leftarrow \tau_\Delta^{(0)}(\boldsymbol{X}_n^{(0)}[k])$             $(N_\tau \times 1)$
25: $\boldsymbol{A}_c^{(0)} \leftarrow -\exp(\boldsymbol{A}_{\log}^{(0)})$
26: $\bar{\boldsymbol{A}}^{(0)} \leftarrow \exp\left(\Delta_0 \odot \boldsymbol{A}_c^{(0)}[\text{None}, :]\right)$
27: $\boldsymbol{C}_0 \leftarrow (p_\alpha(\boldsymbol{L}_B[k])^{-1}(\Delta_0 \odot \boldsymbol{B}_{x,0}))[:,:,\text{None}] \odot \boldsymbol{q}_{\text{weights}}[\text{None}, \text{None}, :]$
28: $RHS_0 \leftarrow \exp((\Delta_0 \odot \boldsymbol{A}_c^{(0)}[\text{None}, :])[:,:,\text{None}] \odot \boldsymbol{q}_{\text{nodes}}[\text{None}, \text{None}, :])$    $(N_\tau \times d \times 8)$
29: $\bar{\boldsymbol{B}}(\boldsymbol{L}_B[k], \boldsymbol{X}_n^{(0)}[k]) \leftarrow \sum_{q=0}^{7} LHS[:,:,q]\, C_0[:,:,q]\, RHS_0[:,:,q]$
30: $\hat{\boldsymbol{H}}^{(0)}[k+1] \leftarrow \bar{\boldsymbol{A}}_{\boldsymbol{L}_B[k]}(\boldsymbol{H}^{(0)}[k][\{\hat{u}_i\}_{i=1}^{N_B}] \odot \bar{\boldsymbol{A}}^{(0)}) + \bar{\boldsymbol{B}}(\boldsymbol{L}_B[k], \boldsymbol{X}_n^{(0)}[k])$
31: $\tilde{\boldsymbol{X}}^{(1)}[k] \leftarrow \tilde{\boldsymbol{X}}^{(0)}[k] + \text{GeLU}(\hat{\boldsymbol{H}}^{(0)}[k+1])$

32: **CTDG-SSM 2$^{\text{nd}}$ Layer**:
33: $\boldsymbol{X}_n^{(1)}[k] \leftarrow \text{RMS}_1(\tilde{\boldsymbol{X}}^{(1)})$
34: $\boldsymbol{B}_{x,1} \leftarrow \boldsymbol{B}_1(\boldsymbol{X}_n^{(1)}[k])$
35: $\Delta_1 \leftarrow \tau_\Delta^{(1)}(\boldsymbol{X}_n^{(1)}[k])$
36: $\boldsymbol{A}_c^{(1)} \leftarrow -\exp(\boldsymbol{A}_{\log}^{(1)})$
37: $\bar{\boldsymbol{A}}^{(1)} \leftarrow \exp(\Delta_1 \odot \boldsymbol{A}_c^{(1)}[:,\text{None}])$
38: $\boldsymbol{C}_1 \leftarrow (p_\alpha(\boldsymbol{L}_B[k])^{-1}(\boldsymbol{B}_{x,2} \odot \Delta_1))[:,:,\text{None}] \odot \boldsymbol{q}_{\text{weights}}[\text{None}, \text{None}, :]$
39: $RHS_1 \leftarrow \exp((\Delta_1 \odot \boldsymbol{A}_c^{(1)}[\text{None}, :])[:,:,\text{None}] \odot \boldsymbol{q}_{\text{nodes}}[\text{None}, \text{None}, :])$
40: $\bar{\boldsymbol{B}}(\boldsymbol{L}[k], \boldsymbol{X}_n^{(1)}[k]) \leftarrow \sum_{q=0}^{7} LHS[:,:,q]\, C_1[:,:,q]\, RHS_1[:,:,q]$
41: $\hat{\boldsymbol{H}}^{(1)}[k+1] \leftarrow \bar{\boldsymbol{A}}_{\boldsymbol{L}_B[k]}(\boldsymbol{H}^{(1)}[k][\{\hat{u}_i\}_{i=1}^{N_B}] \odot \bar{\boldsymbol{A}}^{(1)}) + \bar{\boldsymbol{B}}(\boldsymbol{L}_B[k], \boldsymbol{X}_n^{(1)}[k])$
42: $\hat{\boldsymbol{X}}^{(2)}[k] \leftarrow \tilde{\boldsymbol{X}}^{(1)}[k] + \text{GeLU}(\hat{\boldsymbol{H}}^2[k+1])$
43: $\boldsymbol{H}^{(0)}[k+1] \leftarrow \text{MeanAgg}(\hat{\boldsymbol{H}}^{(0)}[k+1], \{\hat{u}_i\}_{i=1}^{N_B})$    {Only update for batch active node
44: $\boldsymbol{H}^{(1)}[k+1] \leftarrow \text{MeanAgg}(\hat{\boldsymbol{H}}^{(1)}[k+1], \{\hat{u}_i\}_{i=1}^{N_B})$    rest of the node retain old values.}
45: $\tilde{\boldsymbol{X}}^{(2)}[k] \leftarrow \text{MeanAgg}(\hat{\boldsymbol{X}}^{(2)}[k], \{\hat{u}_i\}_{i=1}^{N_B})$
46: **return**

---

**Algorithm 2** CTDG-SSM Inference (link-prediction)

---

**Require:** Link prediction queries $\{(u_i, v_i, t_i)\}_i$
1: $\Delta t_i = \ln(1 + t_i - t_{\text{last},u_i v_i})$            $\{t_{\text{last},u_i v_i}$ is the last interaction time of node $u_i$ and $v_i.\}$
2: $\hat{\boldsymbol{y}}_{(link)}(u_i, v_i, t_i) = [\tilde{\boldsymbol{X}}^{(2)}[u_i] \mid \tilde{\boldsymbol{X}}^{(2)}[v_i] \mid \psi(\Delta t_i)]$        $\{\psi$ is temporal encoding function$\}$
3: $p_i = \boldsymbol{w}^\top \hat{\boldsymbol{y}}_{u_i}$                                        $\{$logit for link prediction$\}$

---

**Algorithm 3** CTDG-SSM Inference (node-classification)

---

**Require:** Interaction $\{(u_i, v_i, t_i)\}_i$
1: $\Delta t_i = \ln(1 + t_i - t_{\text{last},u_i v_i})$            $\{t_{\text{last},u_i v_i}$ is the last interaction time of node $u_i$ and $v_i.\}$
2: $\hat{\boldsymbol{y}}_{u_i} = [\tilde{\boldsymbol{X}}^{(2)}[u_i] \mid \psi(\Delta t_i)]$                      $\{\psi$ is temporal encoding function$\}$
3: $\boldsymbol{p}_i = \boldsymbol{W}\hat{\boldsymbol{y}}_{(link)}(u_i, v_i, t_i)$                          $\{$Multiclass logits$\}$
4: **return**

---

