# OpenReview forum: "CTDG-SSM: Continuous-time Dynamic Graph State Space Models for Long Range Propagation"
_ICLR.cc/2026/Conference — ICLR 2026 Conference Withdrawn Submission_

### Official Review · Reviewer_2WD4 · 2025-10-28

**Soundness:** 2
**Presentation:** 1
**Contribution:** 2
**Rating:** 4
**Confidence:** 3

**Summary:**

This work considers the graph (representation) learning on continuous-time dynamic graphs (CTDGs). Aiming to capture both long-range temporal and spatial dependencies, a new CTDG-SSM (continuous-time dynamic graph state space model) method was then proposed with theoretical guarantees and high scalability, based on some designs of HiPPO-based memory mechanism for CTDGs and ZOH discretization of SSM. Evaluation on public benchmarks of 3 tasks (i.e., dynamic link prediction, dynamic node classification, and sequence classification) demonstrate the effectiveness of CTDG-SSM.

**Strengths:**

**S1**. This work provide a series of rigorous theoretical analysis for the proposed method.

**S2**. The proposed method was evaluated on 3 tasks (i.e., dynamic link prediction, dynamic node classification, and sequence classification) with different purposes.

**S3**. This work anonymously provides its code to ensure reproducibility of experiments.

**Weaknesses:**

**W1**. The overall presentation of this paper is hard to read, which need significant improvement.

Some abbreviations (e.g., HiPPO, LRT, and RMS) were first used without giving their full names.

In lines 43-44, the possible applications of CTDGs (e.g., finance, e-commerce, and social network analysis) were described without giving any citations.

The main paper contains many lengthy paragraphs (especially Section 1 and Section 2), which are hard to read and understand. It is suggested to split them into shorter sub-paragraphs and summarize some key conclusions in tables/figures.


***
**W2**. There are several unclear statements with weak motivations, which need further clarification.

For the problem statement in Section 3, the availability of graph attributes (e.g., node and edge attributes) were not clearly stated. Most related methods have optional inputs for both node and edge attributes. It is also unclear for CTDG-SSM how to incorporate these attributes combined with the induced subgraph adjacency matrix ${\bf{A}}_{\tau}$ and Laplacian matrix ${\bf{L}}_{\tau}$.

According to the problem statement in Section 3, it seems that the subgraph adjacency matrix and the proposed method can only handle the addition of new edges and nodes but cannot tackle the deletion of them. Such a limitation should be clearly stated in the main paper.

In lines 184-186, the quadratic Laplacian regularizer and classic HiPPO formulation have the same definition (i.e., $p({\bf{L}}_{\tau}) = {\bf{I}}$), which seem to be inconsistent.

In lines 194-195, it was claimed that $p({\bf{L})}^{-1}$ is well-defined, but its formal definition (i.e., how to derive $p({\bf{L})}^{-1}$) was not given.

The toy example in Fig. 2 cannot fully demonstrate the overall subgraph sampling procedures described in lines 266-293. As a result, it is hard to understand how the proposed method exactly work by just reading the lengthy text descriptions.

In Section 5, some necessary details about how to train the proposed model (e.g., training loss, training algorithm, optimizer, etc.) are missing.


***
**W3**. There is no pseudo-code to summarize the overall training and inference procedures of CTDG-SSM. As a result, it is hard to check some details about the proposed method.


***
**W4**. While high efficiency and scalability is one of the highlighted advantages of CTDG-SSM, there is no formal analysis about the (space and time) complexity of CTDG-SSM as well as the comparison with complexities of other baselines, which can theoretically validate this advantage.


***
**W5**. Current experiment setups may not fully validate the effectiveness of CTDG-SSM. Some related details are also missing.

There are no descriptions about the experiment environments.

As summarized in Table 4, all the datasets cannot be considered as large-scale dynamic graphs in terms of the number of nodes, which cannot fully verify the high efficiency and scalability of CTDG-SSM. It is suggested to include results from some larger public benchmarks (e.g., TGB).

For details of datasets summarized in Table 4, there is no information about the node classification task (e.g., the number of classes).

It is still unclear why the 3 evaluation tasks (e.g., dynamic link prediction, dynamic node classification, and sequence classification) can measure the ability to capture LRS and LRT as stated in the main paper, i.e., due to what mechanisms? It is also unclear what does the toy example in Fig. 4 mean by just reading the short description in lines 429-431.

As efficiency is usually highly related to scalability, it is also suggested to compare CTDG-SSM's training and inference time with other baselines.


***
**W6**. There are no discussions about the limitations of this work and possible solutions as future research directions.

**Questions:**

See **W1**-**W6**.

---

> ### Author Response · Authors · 2025-11-21
>
> Thank you for your helpful reviews. Following the review, we have incorporated changes to improve the presentation of the work. Additionally, we have conducted experiments on the TGB datasets, including pseudo-code and added details regarding the experiment setup and datasets. we hope our answers have sufficiently addressed your questions. We are happy to provide more information if you have any further concerns.
>
> # W1.  The overall presentation of this paper is hard to read, which need significant improvement.
>
> # Some abbreviations (e.g., HiPPO, LRT, and RMS) were used without their full names being provided.
>
> # In lines 43-44, the possible applications of CTDGs (e.g., finance, e-commerce, and social network analysis) were described without giving any citations.
>
> # The main paper contains many lengthy paragraphs (especially Section 1 and Section 2), which are hard to read and understand. It is suggested to split them into shorter sub-paragraphs and summarize some key conclusions in tables/figures.
>
> We will incorporate the suggested changes in the revised manuscript, including citation (Altman et al., 2023) for LRS in financial networks.
>
> # W2. There are several unclear statements with weak motivations, which need further clarification. or the problem statement in Section 3, the availability of graph attributes (e.g., node and edge attributes) were not clearly stated. Most related methods have optional inputs for both node and edge attributes. It is also unclear for CTDG-SSM how to incorporate these attributes combined with the induced subgraph adjacency matrix
>
>
> We thank the Reviewer for the comment. While the architecture section describes how node features are constructed, we acknowledge that this was not stated explicitly in the problem formulation. To ensure consistency and clarity, we have revised Section 3 to explicitly describe how node and edge attributes are incorporated into CTDG-SSM as follows.
> For each event $(u, v, t_k)$, we construct the feature vector for every involved node by concatenating all available attributes. Specifically, the feature for node $(u)$ at timestamp $t_{k}$ is defined as
>
> $$
> \mathbf{X} [u,:] [t _k] = \big[\tilde{ \mathbf{x}} _u \| \tilde{\mathbf{x}} _v \| \mathbf{x} _{uv} \| \phi(\Delta t _k)\big], \tag 1
> $$
> where $\tilde{\mathbf{x}} _u = [ \mathbf{x} _u ,|, \mathbf{s} _u]$ combines raw node attributes $\mathbf{x} _u$ with static embeddings $\mathbf{s} _u$, $ \mathbf{x} _{uv}$ denotes edge attributes (when available), and $\phi(\Delta t_k)$ encodes temporal information for the event.
>
> The node and edge features are not incorporated with the adjacency or Laplacian matrix; they are used as input to the CTDG-SSM update equation.
>
> # W2. (cont.) According to the problem statement in Section 3, it seems that the subgraph adjacency matrix and the proposed method can only handle the addition of new edges and nodes but cannot tackle the deletion of them. Such a limitation should be clearly stated in the main paper.
> We thank the Reviewer for pointing this out. We clarify that the core CTDG-SSM update equation depends on the change in the graph Laplacian, and therefore naturally accommodates both the addition *and* removal of edges.
>
> To handle edge deletions within the subgraph, one can simply invert the construction process described in the paper. Specifically, if an edge is removed in batch $B$, we construct batch Laplacian $\mathbf{L}_B[k]$ without this edge, while $\mathbf{L}_B[k-1]$ includes it. The resulting difference $\mathbf{L}_B[k] - \mathbf{L}_B[k-1]$ correctly captures the effect of edge removal.
>
> Node deletion can be treated analogously by removing all edges incident to that node. In this case, $\mathbf{L}_B[k]$ contains no edges between the removed node and its neighbors, while $\mathbf{L}_B[k-1]$ retains these edges. This ensures that the update mechanism captures the effective removal of the node.
>
> We did not explicitly discuss this in the main paper because standard CTDG datasets primarily involve edge additions or interactions. However, we appreciate the Reviewer’s observation and will include a clear statement on handling edge and node removals in the revised version.

---

> ### Author Response · Authors · 2025-11-21
>
> # W2. (cont.) In lines 184-186, the quadratic Laplacian regularizer and classic HiPPO formulation have the same definition (i.e., $p(\mathbf L) =  \mathbf I$), which seems to be inconsistent.
>
> We thank the Reviewer for pointing out this issue and appreciate the opportunity to clarify the statement in the main paper. We intended to contrast the classical HiPPO formulation and the GraphSSM formulation with the CTDG-SSM problem setup.
>
> The expression shown in the paper,
>
> $$
> \min _{\mathbf{H} _{i,\tau}} \int _0 ^\tau \left\| \mathbf{X} [:,i] (t) - p(\mathbf{L} _\tau) \mathbf{H} _{i,\tau} \mathbf{g}(t) \right\| _2 ^2\, d \mu (t),
> $$
>
> is the CTDG-SSM formulation, where the polynomial operator $p(\mathbf{L}_\tau)$ specifies how the graph structure influences the reconstruction.
>
> When $p(\mathbf{L}_\tau)=\mathbf{I}$, the Laplacian dependence vanishes, yielding the classical HiPPO objective:
>
> $$
> \min_ {\mathbf{H} _{i,\tau}} \int _0 ^\tau  \left\| \mathbf{X} [:,i] (t) - \mathbf{H} _{i,\tau} \mathbf{g} (t) \right\| _2 ^2 d \mu (t),
> $$
>
> which matches the standard HiPPO setting.
>
> However, when a quadratic Laplacian regularizer is introduced to enforce smoothness over the reconstructed signal, we obtain the GraphSSM objective:
>
> $$
> \min_ {\mathbf{H} _{i,\tau}} \int _0 ^\tau  \left\| \mathbf{X} [:,i] (t) - \mathbf{H} _{i,\tau} \mathbf{g} (t) \right\| _2 ^2\, d \mu (t) + \int _0 ^\tau \big(\mathbf{H} _{i,\tau}\mathbf{g} (t) \big) ^\top  \mathbf{L} _t \big( \mathbf{H} _{i,\tau} \mathbf{g} (t) \big) d \mu (t).
> $$
>
> We will revise the corresponding paragraph in the paper to make this distinction explicit and avoid any ambiguity.
>
> # W2. (cont.) In lines 194-195, it was claimed that $p(\mathbf L) ^{-1}$ is well-defined, but its formal definition (i.e., how to derive $p(\mathbf L) ^{-1}$) was not given.
>
> Recall that the normalized graph Laplacian is
> $$
> \mathbf{L} _{\tau} = \mathbf{I} - \mathbf{D} _{\tau} ^{-1/2} \mathbf{A} _{\tau} \mathbf{D} _{\tau} ^{-1/2},
> $$
> whose eigenvalues satisfy $\lambda _{\tau} \in [0,2]$.
> In our framework, we adopt a finite–impulse–response (FIR) graph filter of the form:
> $$
> p(\mathbf{L} _{\tau})
> = \sum _{k=0}^{K-1} \alpha _k \mathbf{L} _{\tau} ^k
> = \alpha _0 \mathbf{I} + \alpha _1 \mathbf{L} _{\tau} + \cdots + \alpha _{K-1} \mathbf{L} _{\tau} ^{K-1}.
> $$
>
> By parameterizing the coefficients such that $\alpha _i > 0$ for all $i$, we can ensure that
> $$
> p(\lambda) > 0 \quad \text{for all } \lambda \in [0,2],
> $$
> which implies that $p(\mathbf{L} _{\tau})$ is positive definite.  Consequently, $p(\mathbf{L} _{\tau})$ is invertible.
>
> Formally, the operator $p(\mathbf{L})^{-1}$ is simply the matrix inverse of the polynomial filter:
> $$
> p(\mathbf{L})^{-1}
> = \left( \sum _{k=0}^{K-1} \alpha _k \mathbf{L} ^k \right)^{-1}.
> $$
> This is well defined whenever $p(\mathbf{L})$ is positive definite, as guaranteed by the coefficient choice above.
>
> # W2. (cont.) The toy example in Fig. 2 cannot fully demonstrate the overall subgraph sampling procedures described in lines 266-293. As a result, it is hard to understand how the proposed method exactly works by just reading the lengthy text descriptions.
>
>
> To construct the induced subgraph adjacency matrix $\mathbf{A} _B$ for a batch of events, we follow the sampling mechanism described in the architecture:
>
> 1. We gather all nodes participating in the sampled events and reindex them when a node appears multiple times across events.
> 2. For these reindexed nodes, we sample local neighbors to form the *batch active node set*.
> 3. Using the sampled events, we place edges among the active batch nodes and construct $\mathbf{A} _B$, from which the corresponding Laplacian $\mathbf{L} _B$ is derived.
> 4. Finally, for every node in the active set, we construct its feature vector according to Eq. (1).
>
> We will add this clarification to the revised manuscript.
>
> # W2. (cont.)  In Section 5, some necessary details about how to train the proposed model (e.g., training loss, training algorithm, optimizer, etc.) are missing.
>
>  We train our model using the DyGLib (Yu et al., 2023) pipeline, which supports both transductive/inductive settings, as well as negative sampling. For each query, we concatenate the nodes’ state vectors with the temporal embedding, pass them through a linear layer to compute the logit, and calculate the loss using the binary cross-entropy loss with the Adam optimizer. We use Validation AUC for early stopping. We will add these training details to the main paper for completeness.
>
> # W3. There is no pseudo-code to summarize the overall training and inference procedures of CTDG-SSM. As a result, it is hard to check some details about the proposed method.
> Thank you for your feedback regarding the pseudo-code. We have added all the relevant pseudo-codes to the Appendix of the revised manuscript.

---

> ### Author Response · Authors · 2025-11-21
>
> # W4. While high efficiency and scalability are among the highlighted advantages of CTDG-SSM, there is no formal analysis about the (space and time) complexity of CTDG-SSM, as well as the comparison with the complexities of other baselines, which can theoretically validate this advantage.
>
> For active batch nodes $N_B$, state vectors of dimension $d$, and a polynomial of highest order $m$, the state update has a time complexity of   $\mathcal{O}( m N _B ^{3} + d N _B ^{2} )$, it is to be noted that $N_B << N$.
>
> The inference time depends on the task:
> - for **link prediction** and **node classification**, it is $\mathcal{O}(\mathrm{deg}(u))$ due to the computation of $\Delta t$;
> - for **sequence classification**, it is $\mathcal{O}(1)$ per query.
>
> We will include this formal complexity analysis along with comparisons to other baselines in the supplementary material. Additionally, in Tables 1 and 2, where the downstream task is a dynamic link prediction task, we report the per-epoch training time (in minutes), GPU memory usage (in GB), and total training time, which is calculated as the product of per-epoch training time and the number of epochs.
>
> **Table 1: Comparison of model per-epoch time (minutes) and GPU memory usage (GB) across multiple datasets**
> | Models      | LastFM (Time / Mem) | Enron (Time / Mem) | MOOC (Time / Mem) | UCI (Time / Mem) | Reddit (Time / Mem) | Social Evo. (Time / Mem) |
> |------------|-------------------|------------------|-----------------|----------------|-------------------|--------------------------|
> | JODIE      | 4.4 / 2.28        | 0.07 / 1.30      | 0.78 / 2.36     | 0.03 / 1.44    | 3.95 / 1.10       | 4.70 / 1.71             |
> | DyRep      | 6.6 / 2.29        | 0.10 / 1.34      | 0.88 / 2.38     | 0.05 / 1.51    | 5.75 / 1.21       | 7.55 / 1.76             |
> | TGAT       | 22.75 / 4.15      | 1.28 / 3.46      | 4.08 / 3.64     | 0.60 / 3.42    | 16.33 / 2.98      | 25.50 / 3.89            |
> | TGN        | 12.14 / 2.21      | 0.15 / 1.45      | 1.03 / 2.54     | 0.08 / 1.51    | 2.05 / 1.67       | 3.83 / 1.78             |
> | CAWN       | 99.00 / 14.92     | 2.62 / 4.03      | 13.45 / 8.02    | 1.95 / 9.40    | 20.16 / 5.89      | 85.66 / 8.14            |
> | TCL        | 6.23 / 3.04       | 0.30 / 2.51      | 1.00 / 2.49     | 0.13 / 2.00    | 2.25 / 1.82       | 5.05 / 2.48             |
> | GraphMixer | 16.35 / 2.78      | 1.20 / 2.23      | 4.02 / 2.40     | 0.73 / 2.19    | 4.92 / 1.57       | 15.50 / 2.71            |
> | DyGFormer  | 47.00 / 7.57      | 2.73 / 3.23      | 8.32 / 3.35     | 0.62 / 2.30    | 7.00 / 2.42       | 20.00 / 2.77            |
> | CTAN       | 3.33 / 1.44       | 0.50 / 1.33      | 3.22 / 2.30     | 0.38 / 1.30    | 0.86 / 1.54       | 2.41 / 0.63             |
> | DyGMamba   | 28.45 / 4.17      | 2.05 / 2.74      | 4.88 / 2.48     | 0.60 / 1.93    | 6.30 / 2.07       | 17.80 / 2.59            |
> | CTDG-SSM   | 4.45 / 1.15       | 0.55 / 0.86      | 1.25 / 0.43     | 0.17 / 0.31   | 1.95 / 1.18       | 9.57 / 5.2            |
>
> ---
> ---
>
>
>
> **Table 2: Training epochs and total time (minutes) across datasets**
>
> | Models      | Enron (#Epoch / T_tot) | UCI (#Epoch / T_tot) | Reddit (#Epoch / T_tot) |
> |------------|-----------------------|---------------------|------------------------|
> | CTAN       | 173.00 / 86.50        | 236.00 / 89.68      | 327.18 / 173.41        |
> | DyGFormer  | 32.80 / 89.54         | 34.80 / 21.58       | 24.60 / 104.30         |
> | DyGMamba   | 33.00 / 67.65         | 28.00 / 16.80       | 26.80 / 88.98          |
> | CTDG-SSM   | 83.00 / **45.65**     | 38.00 / **6.46**    | 27.00 / **52.65**      |
>
> ---

---

> ### Author Response · Authors · 2025-11-21
>
> # W5. Current experiment setups may not fully validate the effectiveness of CTDG-SSM. Some related details are also missing.
>
> # There are no descriptions about the experiment environments.
>
> We run all experiments on two machines: (1) Intel Xeon Gold 5220R CPU, 251 GB RAM, and an NVIDIA RTX A6000 (48 GB); and (2) Intel Xeon Gold 5218 CPU, 376 GB RAM, and an NVIDIA Quadro RTX 8000 (48 GB). The full Python environment and dependencies are provided in the project repository for reproducibility.
>
> # W5. (cont.) As summarized in Table 4, all the datasets cannot be considered as large-scale dynamic graphs in terms of the number of nodes, which cannot fully verify the high efficiency and scalability of CTDG-SSM. It is suggested to include results from some larger public benchmarks (e.g., TGB).
>
> We thank the reviewer for this helpful suggestion. Our primary dataset selection followed the benchmarks provided in DyGLib. To further evaluate the scalability of CTDG-SSM, we additionally include experiments on larger public temporal graph datasets, including tgbl-wiki and tgbl-coin from the TGB benchmark suite. The corresponding results are reported in Table 4.
>
> # W5. (cont.) For details of datasets summarized in Table 4, there is no information about the node classification task (e.g., the number of classes).
> We thank the reviewer for the comment. For the node classification task, we follow the setup provided in DyGLib (Yu et al., 2023), which defines a binary classification task with two classes. We
> will add this information to the main paper for further clarification.
>
> # W5. (cont.) It is still unclear why the 3 evaluation tasks (e.g., dynamic link prediction, dynamic node classification, and sequence classification) can measure the ability to capture LRS and LRT as stated in the main paper, i.e., due to what mechanisms? It is also unclear what does the toy example in Fig. 4 mean by just reading the short description in lines 429-431.
>
> Dynamic link prediction and node classification mainly require LRT, as short single-hop neighbors are often sufficient to capture the evolution dynamics in these datasets. To specifically evaluate LRS, we utilize the sequence classification task from (Gravina et al. , 2024), where the class of a sequence (based on the first node) is predicted from the state vector of the last node, forcing the model to propagate long-range structural information. The toy example in Fig. 4 illustrates this propagation: even distant node features are preserved in state vectors, demonstrating the model’s ability to capture
> LRS, which baselines like DyGMamba and DyGFormer cannot.
>
> # W5. (cont.) As efficiency is usually highly related to scalability, it is also suggested to compare CTDG-SSM's training and inference time with other baselines.
>
> We present the per-epoch training-time comparison in Table 1 and the total train time calculated as the number of  Epochs (for best performance) $ \times $ per-epoch time in Table 2. For inference, CTDG-SSM retrieves node states and applies a linear layer. For sequence classification, this is $O(1)$. For link prediction and node classification, computing $∆t$ by scanning neighbors adds a $O(deg(u))$ cost before the linear layer
>
> # W6. There are no discussions about the limitations of this work and possible solutions as future research directions.
>
> In the current model, we implement a polynomial of the Laplacian using simple graph filters, which provide an efficient linear approximation to the underlying differential operator. While effective, this design restricts the expressiveness of the operator. An important direction for future work is to explore learning these operators and their inverses directly through graph neural networks, potentially enabling more adaptive and data-dependent approximations. Also, the current framework is primarily evaluated on CTDG datasets, where all node and edge features are fully observed. Extending the framework to handle scenarios with missing features in sampled events, or to accommodate interleaved and partially observed dynamic graphs, presents a promising direction for future research.
>
>
> ---
>
> **Table 4: MRR on the `tgbl-wiki` and `tgbl-coin` datasets**
>
> | Dataset       | CTDG-SSM             | DyGMamba        | DyGFormer       | CTAN            | TGN             |
> |---------------|--------------------|----------------|----------------|----------------|----------------|
> | tgbl-wiki     | **0.817 ± 0.027**  | 0.739 ± 0.009  | 0.798 ± 0.004  | 0.668 ± 0.007  | 0.396 ± 0.060  |
> | tgbl-coin     | **0.862 ± 0.003**  |  $\quad \quad$ ---            | 0.752 ± 0.004  | 0.748 ± 0.004  | 0.586 ± 0.037  |

---

> > ### Author Response · Authors · 2025-11-26
> > **Awaiting for feedback on the Rebuttal**
> >
> > Dear Reviewer 2WD4,
> >
> > We kindly request you to review our detailed rebuttal, which thoroughly addresses the concerns you raised.
> >
> > As requested, the rebuttal provides clear clarifications regarding computational complexity and subgraph sampling. In the revised manuscript, we have added dedicated sections explaining how feature attributes are handled and detailing the procedure for obtaining subgraphs. To further strengthen the clarity, we have included a comprehensive ablation study that demonstrates the importance of the proposed higher-order graph filters and CTDG-SSM in effectively capturing long-range spatial and temporal dependencies.
> >
> > We have also incorporated all your suggestions regarding presentation. In particular, we now include pseudocode that clearly outlines the steps of the proposed method.
> >
> > If our rebuttal satisfactorily resolves your concerns, we respectfully request you to consider raising the score.

---

### Official Review · Reviewer_Ks8e · 2025-10-29

**Soundness:** 2
**Presentation:** 2
**Contribution:** 2
**Rating:** 4
**Confidence:** 5

**Summary:**

This work jointly models temporal dynamics and graph structure for dynamic graphs. It integrates the Mamba state-space architecture with HiPPO-based memory to compress historical information, enabling long-term and long-range sequence modeling. Extensive experiments across multiple benchmarks demonstrate state-of-the-art performance.

**Strengths:**

1. This paper introduces high-order topological information into dynamic graph representation learning and achieves strong performance on long-sequence classification tasks.

2. This paper extends the SSM framework to dynamic graph modeling with a solid theoretical foundation.

3. This paper conducts extensive experiments, showing strong results on dynamic link prediction, dynamic node classification, and long-sequence classification.

**Weaknesses:**

1. Time complexity is our primary concern. For example, matrix inversion costs O(N^3); each batch requires constructing/updating the Laplacian; and a K-order polynomial filter entails K matrix multiplications. The paper does not analyze time complexity or provide comparative runtime experiments. We believe the computational cost is substantial and may hinder real-world deployment. Moreover, the absence of experiments on large-scale graphs further undermines confidence in practical applicability.

2. The proposed HiPPO matrix appears very close to JinTang Li et al. (NeurIPS 2024), seemingly as a direct extension to dynamic graphs. In addition, the adopted Mamba structure looks like a straightforward application of Mamba, without a detailed comparison to existing Mamba-based frameworks. This weakens the claimed architectural novelty.

3. The paper lacks essential ablations. For instance: How do we know the proposed high-order graph filters are effective? How do we verify that the model truly handles and benefits from long-range dependencies? How is robustness demonstrated?

4. The dynamic link prediction benchmarks are selectively chosen. Common datasets such as Flights, Can. Parl., US Legis., UN Trade, UN Vote, and Contact are missing. Comparisons with the latest baselines are also absent—for example: [1] DyG-Mamba: Continuous State Space Model for Dynamic Graphs; [2] Towards Better Evaluation for Dynamic Link Prediction; [3] FreeDyG: Frequency-Enhanced Continuous-Time Dynamic Graph Model for Link Prediction.

5. There is no clear investigation of input sequence length or of higher-order graph structure. It remains unclear whether gains come from long-sequence modeling or from high-order structural information. More experiments are needed to disentangle these factors. Prior studies (e.g., GraphMixer) have reported that longer sequences may not help, which the authors should address explicitly.

6. The results on MOOC are surprisingly high, and our reproduction raises a potential data-leakage issue. Specifically, the code calls ssm_utlis.py::get_delta_t(...) with the default parameter default=1e+11. This may leak information for negative samples. The authors should clarify this choice and whether it preserves fairness with DyGLib and other baselines.

**Questions:**

Please refer to the above-mentioned weaknesses.

---

> ### Author Response · Authors · 2025-11-21
> **Response to Reviewer Ks8e**
>
> We thank the Reviewer for his time and feedback.  We take his opportunity to clarify the reviewers questions on novelty in the proposed CTDG-SSM framework,  robustness and long range propogation,
>
> # W1. Time complexity is our primary concern. For example, matrix inversion costs $O(N^3)$; each batch requires constructing/updating the Laplacian; and a K-order polynomial filter entails K matrix multiplications. The paper does not analyze time complexity or provide comparative runtime experiments. We believe the computational cost is substantial and may hinder real-world deployment. Moreover, the absence of experiments on large-scale graphs further undermines confidence in practical applicability.
>
> We thank the reviewer for this comment. Although computing memory representations in $\texttt{CTDG-SSM}$ involves inverting a polynomial of the graph Laplacian, we emphasize that this operation is performed only on the subset of nodes involved in the current mini-batch. Specifically, the algorithm constructs a Laplacian $\mathbf{L} _{B}$ over the nodes from the current batch and their sampled neighbors $N _u$. If $N _{B}$ denotes the number of nodes in this combined set, then the cost of inverting $p(\mathbf{L} _{B})$ is $O(N _{B} ^{3})$. In practice, $N _{B} << N$, where $N$ is the size of the full graph, meaning that the inversion is applied to a very small matrix and is therefore computationally lightweight. Since $N _{B}$ is determined by the batch size, it behaves like a tunable hyperparameter. Empirically, this design yields efficient training, as evidenced by our shorter runtimes.
>
> In Tables 1 and 2, we report the per-epoch training time (in minutes), GPU memory usage (in GB), and total training time, which is calculated as the product of per-epoch training time and the number of epochs on link prediction task. Notably, CTDG-SSM achieves substantially lower runtime and memory usage compared to DyGMamba and DyGFormer, both of which are designed for long-range propagation tasks. As explained earlier, although the update involves operations such as matrix inversion, these operators are applied to matrices of very small dimensionality, making the computation lightweight in practice.
>
> We also evaluate CTDG-SSM on other large-scale temporal datasets, such as tgbl-wiki and tgbl-coin.  The results are presented in Table 4 clearly emphasizes the proposed model ability in efficiency (interms of capturing LRS and LRT information) and scalabity.

---

> ### Author Response · Authors · 2025-11-21
>
> **Table 1: Comparison of model per-epoch time (minutes) and GPU memory usage (GB) across multiple datasets**
> | Models      | LastFM (Time / Mem) | Enron (Time / Mem) | MOOC (Time / Mem) | UCI (Time / Mem) | Reddit (Time / Mem) | Social Evo. (Time / Mem) |
> |------------|-------------------|------------------|-----------------|----------------|-------------------|--------------------------|
> | JODIE      | 4.4 / 2.28        | 0.07 / 1.30      | 0.78 / 2.36     | 0.03 / 1.44    | 3.95 / 1.10       | 4.70 / 1.71             |
> | DyRep      | 6.6 / 2.29        | 0.10 / 1.34      | 0.88 / 2.38     | 0.05 / 1.51    | 5.75 / 1.21       | 7.55 / 1.76             |
> | TGAT       | 22.75 / 4.15      | 1.28 / 3.46      | 4.08 / 3.64     | 0.60 / 3.42    | 16.33 / 2.98      | 25.50 / 3.89            |
> | TGN        | 12.14 / 2.21      | 0.15 / 1.45      | 1.03 / 2.54     | 0.08 / 1.51    | 2.05 / 1.67       | 3.83 / 1.78             |
> | CAWN       | 99.00 / 14.92     | 2.62 / 4.03      | 13.45 / 8.02    | 1.95 / 9.40    | 20.16 / 5.89      | 85.66 / 8.14            |
> | TCL        | 6.23 / 3.04       | 0.30 / 2.51      | 1.00 / 2.49     | 0.13 / 2.00    | 2.25 / 1.82       | 5.05 / 2.48             |
> | GraphMixer | 16.35 / 2.78      | 1.20 / 2.23      | 4.02 / 2.40     | 0.73 / 2.19    | 4.92 / 1.57       | 15.50 / 2.71            |
> | DyGFormer  | 47.00 / 7.57      | 2.73 / 3.23      | 8.32 / 3.35     | 0.62 / 2.30    | 7.00 / 2.42       | 20.00 / 2.77            |
> | CTAN       | 3.33 / 1.44       | 0.50 / 1.33      | 3.22 / 2.30     | 0.38 / 1.30    | 0.86 / 1.54       | 2.41 / 0.63             |
> | DyGMamba   | 28.45 / 4.17      | 2.05 / 2.74      | 4.88 / 2.48     | 0.60 / 1.93    | 6.30 / 2.07       | 17.80 / 2.59            |
> | CTDG-SSM   | 4.45 / 1.15       | 0.55 / 0.86      | 1.25 / 0.43     | 0.17 / 0.31   | 1.95 / 1.18       | 9.57 / 5.2            |
>
> ---
> ---
>
>
>
> **Table 2: Training epochs and total time (minutes) across datasets**
>
> | Models      | Enron (#Epoch / T_tot) | UCI (#Epoch / T_tot) | Reddit (#Epoch / T_tot) |
> |------------|-----------------------|---------------------|------------------------|
> | CTAN       | 173.00 / 86.50        | 236.00 / 89.68      | 327.18 / 173.41        |
> | DyGFormer  | 32.80 / 89.54         | 34.80 / 21.58       | 24.60 / 104.30         |
> | DyGMamba   | 33.00 / 67.65         | 28.00 / 16.80       | 26.80 / 88.98          |
> | CTDG-SSM   | 83.00 / **45.65**     | 38.00 / **6.46**    | 27.00 / **52.65**      |
>
> ---
>
> **Table 4: MRR on the `tgbl-wiki` and `tgbl-coin` datasets**
>
> | Dataset       | CTDG-SSM             | DyGMamba        | DyGFormer       | CTAN            | TGN             |
> |---------------|--------------------|----------------|----------------|----------------|----------------|
> | tgbl-wiki     | **0.817 ± 0.027**  | 0.739 ± 0.009  | 0.798 ± 0.004  | 0.668 ± 0.007  | 0.396 ± 0.060  |
> | tgbl-coin     | **0.862 ± 0.003**  |  $\quad \quad$ ---            | 0.752 ± 0.004  | 0.748 ± 0.004  | 0.586 ± 0.037  |

---

> ### Author Response · Authors · 2025-11-21
>
> # W.2 The proposed HiPPO matrix appears very close to JinTang Li et al. (NeurIPS 2024), seemingly as a direct extension to dynamic graphs. In addition, the adopted Mamba structure appears to be a straightforward application of Mamba, without a detailed comparison to existing Mamba-based frameworks. This weakens the claimed architectural novelty.
>
> The fact that GraphSSM cannot be applied to CTDGs makes the proposed model and development novel. As such, the proposed extension is not straightforward. Below, we summarize the key challenges involved in adapting GraphSSM to the continuous-time setting:
>
> -  **Deriving a State-Space Update That Accounts for Event-Driven Changes in Graph Topology**
>   This is crucial because the CTDGs model fine-grained temporal evolution, where interactions occur at arbitrary, irregular time intervals. Extending the discrete-time update used in GraphSSM to a continuous-time, event-driven form requires a non-trivial modification of the underlying SSM dynamics, which must account for structural changes as captured through  $\frac{dp(\mathbf{L}_s)}{ds}$ term in the proposed state update equation. We further emphasize that obtaining such a recursive update requires a new framework that models time-varying node features as *joint time-vertex signals*, as clearly explained in the paper.
> - **Algorithm that can scale to large-scale datasets with significantly lower runtime**:
>   The proposed algorithm, developed to update the states of active nodes in sampled events, is novel, and this stands in contrast to existing representation-learning architectures for CTDGs, where updates are typically fully event-driven but not restricted to specific participating nodes. More importantly, a computationally tractable discrete state-space update is novel and the first of its kind in SSM-based frameworks.
> - **Theoretical Analysis**:
>   To the best of our knowledge, this is the first work to provide a theoretical analysis establishing robustness and permutation equivariance properties. Their inclusion requires deriving the continuous-time update rules from first principles.
>
> We agree that residual connections and memory modules are simply architectural mechanisms for stability (preserving input data for large depths) for the storage of hidden states, and not the conceptual contribution.
>
> We further emphasize that our framework differs substantially from existing Mamba-based graph models. In particular, unlike Graph-Mamba (Behrouz & Hashemi, 2024), which is primarily designed for static graphs and applies Mamba layers without modeling temporal evolution, our proposed CTDG-SSM block introduces a new state-space update tailored specifically for continuous-time dynamic graphs (CTDGs). The core contribution lies in a computationally tractable state update
> equation that incorporates time-varying graph structure, as well as a dedicated memory module that
> stores and evolves node representations over time. These components enable CTDG-SSM to capture both temporal and structural dynamics capabilities that static Graph-Mamba architectures do
> not support.

---

> ### Author Response · Authors · 2025-11-21
>
> # W3. The paper lacks essential ablations. For instance: How do we know the proposed high-order graph filters are effective? How do we verify that the model truly handles and benefits from long-range dependencies? How is robustness demonstrated?
>
> We thank the reviewer for their feedback and have included additional ablations. In addition to the sequence classification task presented in the main paper for evaluating the effectiveness of different polynomials on LRS and LRT, we have considered another sequence classification task.  This ablation study is used to understand which components of the model capture long-range information. Specifically, we analyze the role of graph filters and the proposed CTDG-SSM module on the long-range spatial ($\texttt{LRS}$) task and the long-range temporal ($\texttt{LRT}$) task.
>
> To evaluate the $\texttt{LRT}$ capabilities of CTDG-SSM, we first set the polynomial $p(\mathbf{L}) = \mathbf{I}$. For an event of the form $(u,v,t,x_u,x_v)$ in sequence classification, instead of restricting the model to update only a small subset of nodes (i.e., those in the batch subgraph), we update the state vectors of all nodes. Formally, we define the input signal at time $t$ as:
>
> $$
> \mathbf{X}(t) \in \mathbb{R}^{N _\tau \times 1} \quad \text{such that} \quad \mathbf{X} [u] (t) = x _u, \mathbf{X} [v] (t) = x _v, \text{and 0 otherwise}.
> $$
>
> This eliminates the step where the previous states of inactive nodes are carried forward through memory. This carry-forward mechanism could aid the model in LRT, so by removing it, we can evaluate the LRT capability of CTDG-SSM in isolation.
> In this setup, the model is tasked with predicting the initial feature $\mathbf{X} [0] (0)$ observed at the first node using the final state vectors of different nodes. A successful LRT would yield strong performance as long as the state vector of node 1 preserves the information of the feature $\mathbf{X} [0] (0)$. Notably, this model completely lacks LRS capability, as it does not account for the underlying graph structure and updates states solely based on the input at the corresponding nodes.
>
> In Table 3  [1st column], we report the mean accuracy using the representations from each node. It can be observed that in the absence of the graph Laplacian and the memory module-based carry-forward, the model retains strong long-range temporal information, as indicated by the high accuracy at the first node. However, the accuracy for predictions originating from other nodes is substantially lower, demonstrating that the model fails to capture long-range spatial dependencies. This setup highlights that while the proposed SSM module alone effectively captures temporal long-range information, it fails to preserve long-range spatial information.
>
> Next, we evaluate the effect of aggregating multi-hop information by applying graph filters of different orders. and report the mean accuracies obtained from representations at different nodes using a filter of order 1 (Table 3 [2nd column]) and a filter of order 2 (Table 3 [3rd column]). We observe a substantial improvement in prediction accuracy based on the representations from the second node (Table 3 [2nd row]),..., the last node (Table 3 [last row]), demonstrating the model’s enhanced ability to preserve spatial information over longer ranges. In particular, using deeper aggregation-i.e., a filter of order 2-yields a notable gain in accuracy, indicating that incorporating information from larger hop neighborhoods significantly strengthens the model’s capacity to capture
> long-range spatial dependencies.
>
> **Comment on Robustness**
>
> In the revised manuscript we present a clear study on the robustness of the proposed model against structural perturbations. In particular we present the detailed perturbation analysis on the Enron dataset with the link prediction task, where in we  perturb the graph Laplacian via  perturbation matrix whose elements are generated from a normal distribution with $[∆\mathbf{L}] _{ij}$ ∼$N (0, 1)$.  Then we evaluate the proposed algorithm with perturbed graph Laplacian and report accuracy across different noise levels.  We observed that accuracy decreases as the noise variance increases; however, for
> small values of noise variance, the model performs very close to the noise-free setting. This demonstrates that the
> proposed approach is stable and robust under mild structural perturbations. Please refer the section E.3 and Fig.7 in the  revised draft for the detailed discussion.

---

> ### Author Response · Authors · 2025-11-21
>
> **Table 3: Ablation study with respect to the order of the graph filter and memory-based carry-forward.**
> | Prediction Node | $p(\mathbf{L}) = \mathbf{I}$ | $p(\mathbf{L}) = \alpha _0 \mathbf{I} + \alpha _1 \mathbf{L}$ | $p(\mathbf{L}) = \alpha _0 \mathbf{I} + \alpha _1 \mathbf{L} + \alpha _2 \mathbf{L}^2$ |
> |-----------------|-------------------------------|-----------------------------------------------|------------------------------------------------------|
> | First           | 1.00 ± 0.00                   | 1.00 ± 0.00                                   | 1.00 ± 0.00                                         |
> | Second          | 0.51 ± 0.06                   | 0.97 ± 0.02                                   | 1.00 ± 0.00                                         |
> | Third           | 0.47 ± 0.02                   | 0.96 ± 0.01                                   | 1.00 ± 0.00                                         |
> | Second-Last     | 0.46 ± 0.01                   | 0.90 ± 0.01                                   | 0.92 ± 0.07                                         |
> | Last            | 0.45 ± 0.02                   | 0.88 ± 0.18                                   | 0.90 ± 0.06                                         |

---

> ### Author Response · Authors · 2025-11-21
>
> # W.4 The dynamic link prediction benchmarks are selectively chosen. Common datasets such as Flights, Can. Parl., US Legis., UN Trade, UN Vote, and Contact are missing. Comparisons with the latest baselines are also absent—for example: [1] DyG-Mamba: Continuous State Space Model for Dynamic Graphs; [2] Towards Better Evaluation for Dynamic Link Prediction; [3] FreeDyG: Frequency-Enhanced Continuous-Time Dynamic Graph Model for Link Prediction.
>
> We have included experiments on all the mentioned datasets under both transductive and inductive settings, using different sampling strategies. In Table 5, we present the results, which clearly show that the proposed model consistently outperforms on most of the aforementioned datasets due to its ability to jointly encode the structural and temporal evolution using graph filters and state space models. In Table 6, we compare our model with the aforementioned baselines, showing that the proposed model consistently outperforms them across the datasets.
>
> Due to character limit we have included the additional results with AUC ROC and AP under tranductive and inductive settings in the revised mansucript.
>
> ---
>
> **Table 5: AP  for dynamic link prediction under transductive setting**
>
> | **NSS** | **Dataset**     | `JODIE` | `DyRep` | `TGAT` | `TGN` | `CAWN` | `EdgeBank` | `TCL` | `GraphMixer` | `DyGFormer` | `DyGMamba` | `CTDG-SSM` |
> |---------|-----------------|---------|---------|--------|-------|--------|------------|-------|---------------|-------------|------------|------------|
> | rnd     | Flights         | 96.21 ± 1.42  | 95.95 ± 0.62  | 94.13 ± 0.17  | 98.22 ± 0.13  | 98.45 ± 0.01  | 90.23 ± 0.00  | 91.21 ± 0.02  | 91.13 ± 0.01  | 98.93 ± 0.01  | **98.98 ± 0.05**  | 98.53 ± 0.02 |
> | rnd     | Can. Parl.      | 78.21 ± 0.23  | 73.35 ± 3.67  | 75.69 ± 0.78  | 76.99 ± 1.80  | 75.70 ± 3.27  | 64.14 ± 0.00  | 72.46 ± 3.23  | 83.17 ± 0.53  | 97.76 ± 0.41  | **99.69 ± 0.06**  | 98.10 ± 0.80 |
> | rnd     | US Legis.       | 82.85 ± 1.07  | 82.28 ± 0.32  | 75.84 ± 1.99  | 83.34 ± 0.43  | 77.16 ± 0.39  | 62.57 ± 0.00  | 76.27 ± 0.63  | 76.96 ± 0.79  | 77.90 ± 0.58  | 79.03 ± 0.26  | **85.92 ± 0.00** |
> | rnd     | UN Trade        | 69.62 ± 0.44  | 67.44 ± 0.83  | 64.01 ± 0.12  | 69.10 ± 1.67  | 68.54 ± 0.18  | 66.75 ± 0.00  | 64.72 ± 0.05  | 65.52 ± 0.51  | 70.20 ± 1.44  | 71.41 ± 0.21  | **73.76 ± 0.30** |
> | rnd     | UN Vote         | 68.53 ± 0.95  | 67.18 ± 1.04  | 52.83 ± 1.12  | 68.71 ± 2.65  | 53.09 ± 0.22  | 62.97 ± 0.00  | 51.88 ± 0.36  | 52.46 ± 0.27  | 57.12 ± 0.62  | 58.48 ± 0.12  | **97.41 ± 0.00** |
> | rnd     | Contact         | 96.66 ± 0.89  | 96.48 ± 0.14  | 96.95 ± 0.08  | 97.54 ± 0.35  | 89.99 ± 0.34  | 94.34 ± 0.00  | 94.15 ± 0.09  | 93.94 ± 0.02  | 98.53 ± 0.01  | 98.68 ± 0.02  | **98.70 ± 0.01** |
>
> ---
>
> **Table 6: Performance comparison with AP on dynamic link prediction models across datasets**
>
> | **NSS** | **Datasets**   | `Edgebank`       | `DyG-Mamba`      | `FreeDyG`        | `CTDG-SSM`       |
> |---------|----------------|-----------------|-----------------|-----------------|-----------------|
> | 1       | Wiki           | 90.37 ± 0.00    | 99.08 ± 0.09    | 99.26 ± 0.01    | **99.40 ± 0.00** |
> | 2       | Reddit         | 94.86 ± 0.00    | 99.27 ± 0.00    | 99.48 ± 0.01    | **99.53 ± 0.00** |
> | 3       | MOOC           | 57.97 ± 0.00    | 90.25 ± 0.01    | 89.61 ± 0.10    | **98.85 ± 0.00** |
> | 4       | LastFM         | 79.29 ± 0.00    | **94.23 ± 0.01**| 92.15 ± 0.16    | 93.40 ± 0.49    |
> | 5       | Enron          | 83.53 ± 0.00    | 93.14 ± 0.08    | 92.51 ± 0.05    | **94.46 ± 4.73** |
> | 6       | Social Evo.    | 74.95 ± 0.00    | 94.77 ± 0.01    | 94.91 ± 0.01    | **98.65 ± 0.65** |
> | 7       | UCI            | 76.20 ± 0.00    | 96.14 ± 0.14    | **96.28 ± 0.11**| 90.18 ± 0.98    |

---

> ### Author Response · Authors · 2025-11-21
>
> # W5. There is no clear investigation of input sequence length or of higher-order graph structure. It remains unclear whether gains come from long-sequence modeling or from high-order structural information. More experiments are needed to disentangle these factors. Prior studies (e.g., GraphMixer) have reported that longer sequences may not help, which the authors should address explicitly.
>
> We thank the reviewers for bringing this important point to our attention. We clarify that, in our model, *sequence length is not a hyperparameter* and does not play the same role as in prior temporal graph architectures.
>
> Prior studies, such as GraphMixer, DyGFormer, and DyGMamba, retrieve historical interactions from the dataset at inference time. For each query $(u, v, t)$, these models collect a sequence of past events and process it to produce the prediction. In such frameworks, the sequence length directly determines how much temporal history is examined and therefore becomes a critical design choice. This is precisely why these works study the effects of longer or shorter sequences, and why some report diminishing returns from very long histories.
>
> Our model operates under a fundamentally different paradigm. We $\textbf{do not}$ retrieve historical temporal data for each new query. Instead, each node maintains a \textit{persistent state vector} that is updated online as interactions occur. These state vectors serve as compressed, continuously updated representations of each node’s temporal neighborhood and structural context.
>
> Concretely:
>
> - When the model encounters a new node during deployment, it initializes its state vector to zero.
> - Whenever an interaction involving this node (or its neighbors) occurs, the corresponding state vectors are updated once.
> - When a query $(u, v, t)$ is presented, the model simply reads the *latest* state vectors of the involved nodes; it **does not** recompute or traverse historical sequences.
>
> Thus, the model never processes variable-length historical sequences during inference, eliminating the notion of a tunable temporal “sequence length”. The performance gains we observe, therefore, do not stem from increasing sequence length, but rather from how the model’s state-space dynamics propagate and maintain high-order structural and temporal information over time.
>
> We will clarify this distinction more explicitly in the manuscript.

---

> ### Author Response · Authors · 2025-11-21
>
> # W6. The results on MOOC are surprisingly high, and our reproduction raises a potential data-leakage issue. Specifically, the code calls ssm_utlis.py::get_delta_t(...) with the default parameter default=1e+11. This may leak information for negative samples. The authors should clarify this choice and whether it preserves fairness with DyGLib and other baselines.
>
> We thank the Reviewer for the observation. We clarify that while ' get_delta_t ' assigns a default value of $10^{11}$ for first-time pairs, this does not introduce data leakage. Under *Random Negative Sampling (RNS)*, a pair may indeed appear for the first time as a negative sample and thus receive this default $\Delta t$. However, our method also achieves consistently strong performance under *Historical Negative Sampling (HNS)*, where all negative edges correspond to node pairs that have previously appeared in the graph's history. In this setting, negative samples never receive the default $\Delta t$. If the model were unintentionally exploiting the default value, we would observe a performance drop under HNS, which is not the case. This clearly indicates that the model is not relying on this value.
>
> Additionally, the model does not directly utilize raw $\Delta t$ values. Before being input to the model, each $\Delta t$ undergoes a logarithmic compression, $\ln(1 + \Delta t)$, which significantly reduces the influence of extreme values ( including the default $10^{11}$ ) and prevents them from dominating the temporal representation. This further mitigates any risk of leakage.
>
> Finally, assigning a large initial $\Delta t$ for unseen pairs is a standard practice in prior temporal graph models, most notably *DyGMamba*, to represent unknown or effectively infinite time gaps. We follow this convention to ensure fairness and maintain comparability with DyGLib and other baselines.

---

> > ### Author Response · Authors · 2025-11-26
> >
> > Dear Reviewer ks8e,
> >
> > We kindly request you to review our detailed rebuttal, which thoroughly addresses the concerns you raised.
> >
> > As requested, the rebuttal provides clear clarifications regarding the novelty of our approach and its computational complexity. In the revised manuscript, we have also included a comprehensive ablation study that highlights the importance of the proposed higher-order graph filters and CTDG-SSM in effectively capturing long-range spatial and temporal dependencies. To further support the theoretical analysis on robustness, we have added new experimental results.
> >
> > Additionally, we now present experiments on all the previously mentioned datasets and compare our method against the relevant state-of-the-art baselines.
> >
> > If our rebuttal satisfactorily resolves your concerns, we respectfully request you to consider raising the score.

---

### Official Review · Reviewer_Dwin · 2025-10-30

**Soundness:** 3
**Presentation:** 3
**Contribution:** 2
**Rating:** 4
**Confidence:** 4

**Summary:**

This paper introduces a new state-space modeling framework for continuous-time dynamic graphs that jointly captures temporal and structural dependencies. It proposes CTT-HiPPO, a topology-aware memory formulation that projects classical HiPPO solutions through polynomials of the graph Laplacian to encode both temporal evolution and multi-hop structural context, leading to the unified CTDG-SSM formulation. The framework is theoretically grounded with robustness and permutation-equivariance guarantees, discretized for efficient implementation, and achieves state-of-the-art performance on link prediction, node classification, and sequence classification tasks that require long-range temporal and spatial reasoning.

**Strengths:**

1. The paper is generally well written with strong theoretical support.
2. The proposed method shows strong results and can capture long range temporal and structural dependencies.

**Weaknesses:**

1. The contribution is somewhat limited. Authors start from DyGMamba and propose an advanced version of SSM-based temporal graph reasoning model. The modification of HiPPO is a good point, but the authors didn’t explain why they wanted to develop their method based on SSM and which characteristics drive them build method on top of it. For other modules like residual connection and memory components, these are for me just some combination of common practices in model design.
2. Lack of important experiments. I really wish to empirically see which part of the model enables long range dependencies being effectively captured. Currently this is not shown clearly. It would be better to have related experiments and detailed analysis.

**Questions:**

1. For kth order filter, is $L_\tau^k$ the laplacian matrix for k hop neighbor? Or is it the kth order of $L_\tau$?
2. Do you think sequence classification is way too artificial? Could you share where can sequence classification, i.e., preserving bode label, be critical in real world applications?
3. Do you think the long range dependency of your model comes from the memory module rather than the your design of SSM? Lets say, even if you employ multihop and temporal-aware graph filters, it is still not guaranteed that the model would remember LRT and LRS. Could you provide some kind of analysis to show that the contribution actually comes from your SSM design? This is very important in determining the quality of the proposed method.
4. I saw model size performance comparison. What about inference time and training time/convergence?

---

> ### Author Response · Authors · 2025-11-21
> **Response to Reviewer Dwin**
>
> We thank the Reviewer for the time and effort in reviewing the manuscript. Below we provide a point to point response
>
> # W1. The contribution is somewhat limited. Authors start from DyGMamba and propose an advanced version of SSM-based temporal graph reasoning model. The modification of HiPPO is a good point, but the authors didn’t explain why they wanted to develop their method based on SSM and which characteristics drive them build method on top of it. For other modules like residual connection and memory components, these are for me just some combination of common practices in model design.
>
> We thank the Reviewer for the comment. We clarify and emphasize that our approach does not originate from $\texttt{DyGMamba}$. $\texttt{DyGMamba}$ primarily adapts the existing $\texttt{Mamba}$ architecture to CTDGs, leveraging its ability to capture long-range temporal dependencies while restricting spatial propagation to one hop. In contrast, $\texttt{CTDG-SSM}$ is a state space model for CTDGs developed from first principles, without imposing one-hop spatial constraints, and is based on HiPPO. As already mentioned in the paper, our framework is motivated by SSMs due to its key characteristics that are directly aligned with the natural requirements of CTDGs:
>
>   (a)  Superior long-range temporal (LRT) modeling with linear complexity: SSMs (e.g., S4, Mamba) are empirically and theoretically strong at capturing dependencies over very long sequences while remaining efficient.
>  (b) HiPPO-initialized SSMs achieve expressive dynamics with low parameter overhead. HiPPO matrices offer a principled approach to continuous-time memory, enabling stable representations of streaming temporal processes.
>
> The fact that GraphSSM cannot be applied to CTDGs makes the proposed model and development novel. As such, the proposed extension is not straightforward. Below, we summarize the key challenges involved in adapting GraphSSM to the continuous-time setting:
>
> - **Deriving a state-space update that accounts for event-driven changes in graph topology:**
> This is crucial because the CTDGs model fine-grained temporal evolution, where interactions occur at arbitrary, irregular time intervals. Extending the discrete-time update used in GraphSSM to a continuous-time, event-driven form requires a non-trivial modification of the underlying SSM dynamics, which accounts for the underlying structural changes as captured by   $\frac{dp(\mathbf{L} _s)}{ds}$  in the proposed state space equation. We further emphasize that obtaining such a recursive update requires a new framework that models time-varying node features as joint time-vertex signals, as clearly explained in the paper.
>
> - **Algorithm that can scale to large-scale datasets with significantly lower runtime:**
> The proposed algorithm, developed to update the states of active nodes in sampled events, is novel. This stands in contrast to existing representation-learning architectures for CTDGs, where updates are typically fully event-driven but not restricted to specific participating nodes. More importantly, a computationally tractable discrete state-space update is novel and first of its kind in SSM-based frameworks.
>
> - **Theoretical Analysis:**
> To the best of our knowledge, this is the first work to provide a theoretical analysis establishing robustness and permutation equivariance properties. Their inclusion requires deriving the continuous-time update rules from first principles.

---

> ### Author Response · Authors · 2025-11-21
>
> # W2. Lack of important experiments.  I really wish to empirically see which part of the model enables long range dependencies being effectively captured. Currently this is not shown clearly. It would be better to have related experiments and detailed analysis.
>
> Considering the downstream task as sequence classification, we conduct an ablation study to understand which components of the model capture long-range information. Specifically, we analyze the role of graph filters and the proposed CTDG-SSM module on the long-range spatial ($\texttt{LRS}$) task and the long-range temporal ($\texttt{LRT}$) task.
>
> To evaluate the $\texttt{LRT}$ capabilities of CTDG-SSM, we first set the polynomial $p(\mathbf{L}) = \mathbf{I}$. For an event of the form $(u,v,t,x_u,x_v)$ in sequence classification, instead of restricting the model to update only a small subset of nodes (i.e., those in the batch subgraph), we update the state vectors of all nodes. Formally, we define the input signal at time $t$ as:
>
> $$
> \mathbf{X}(t) \in \mathbb{R}^{N _\tau \times 1} \quad \text{such that} \quad \mathbf{X} [u] (t) = x _u, \mathbf{X} [v] (t) = x _v, \text{and 0 otherwise}.
> $$
>
> This eliminates the step where the previous states of inactive nodes are carried forward through memory. This carry-forward mechanism could aid the model in LRT, so by removing it, we can evaluate the LRT capability of CTDG-SSM in isolation.
> In this setup, the model is tasked with predicting the initial feature $\mathbf{X} [0] (0)$ observed at the first node using the final state vectors of different nodes. A successful LRT would yield strong performance as long as the state vector of node 1 preserves the information of the feature $\mathbf{X} [0] (0)$. Notably, this model completely lacks LRS capability, as it does not account for the underlying graph structure and updates states solely based on the input at the corresponding nodes.
>
> In Table 3  [1st column], we report the mean accuracy using the representations from each node. It can be observed that in the absence of the graph Laplacian and the memory module-based carry-forward, the model retains strong long-range temporal information, as indicated by the high accuracy at the first node. However, the accuracy for predictions originating from other nodes is substantially lower, demonstrating that the model fails to capture long-range spatial dependencies. This setup highlights that while the proposed SSM module alone effectively captures temporal long-range information, it fails to preserve the long-range information spatially.
>
> Next, we evaluate the effect of aggregating multi-hop information by applying graph filters of different orders. and report the mean accuracies obtained from representations at different nodes using a filter of order 1 (Table 3 [2nd column]) and a filter of order 2 (Table 3 [3rd column]). We observe a substantial improvement in prediction accuracy based on the representations from the second node (Table 3 [2nd row]),..., the last node (Table 3 [last row]), demonstrating the model’s enhanced ability to preserve spatial information over longer ranges. In particular, using deeper aggregation-i.e., a filter of order 2-yields a notable gain in accuracy, indicating that incorporating information from larger hop neighborhoods significantly strengthens the model’s capacity to capture
> long-range spatial dependencies.
>
> ---
>
> **Table 3: Ablation study with respect to the order of the graph filter and memory-based carry-forward.**
> | Prediction Node | $p(\mathbf{L}) = \mathbf{I}$ | $p(\mathbf{L}) = \alpha _0 \mathbf{I} + \alpha _1 \mathbf{L}$ | $p(\mathbf{L}) = \alpha _0 \mathbf{I} + \alpha _1 \mathbf{L} + \alpha _2 \mathbf{L}^2$ |
> |-----------------|-------------------------------|-----------------------------------------------|------------------------------------------------------|
> | First           | 1.00 ± 0.00                   | 1.00 ± 0.00                                   | 1.00 ± 0.00                                         |
> | Second          | 0.51 ± 0.06                   | 0.97 ± 0.02                                   | 1.00 ± 0.00                                         |
> | Third           | 0.47 ± 0.02                   | 0.96 ± 0.01                                   | 1.00 ± 0.00                                         |
> | Second-Last     | 0.46 ± 0.01                   | 0.90 ± 0.01                                   | 0.92 ± 0.07                                         |
> | Last            | 0.45 ± 0.02                   | 0.88 ± 0.18                                   | 0.90 ± 0.06                                         |

---

> ### Author Response · Authors · 2025-11-21
>
> # Q1. For the kth order filter, is $L_\tau^k$ the Laplacian matrix for the k-hop neighbor? Or is it the kth order of $L_\tau$?
>
> We clarify that the term $\mathbf{L} ^{k} _{\tau} $ is the $k$-th power of $\mathbf{L} _{\tau}$, and it expands the receptive field so that information is aggregated from nodes up to $k$-hops away, but it is not the Laplacian of a $k$-hop graph.
>
> # Q2. Do you think sequence classification is way too artificial? Could you share where sequence classification, i.e., preserving node label, can be critical in real-world applications?
>
> Although the sequence classification task is synthetically constructed, its purpose is to evaluate the model’s ability to capture long-range statistical (LRS) and long-range temporal (LRT) dependencies. Importantly, the setup mirrors real-world scenarios where interactions naturally come from event-driven chains, with each new entity linking to the previous one, such as information diffusion in social networks or transaction flows in financial systems. In these settings, the temporal path contains rich structural and temporal signals about the origin. Predicting the label or category of the initial node after observing the entire interaction chain is therefore meaningful. For instance, in rumor or fake-news detection, the goal is often to determine whether the source of a long propagation chain is a bot or a human. Likewise, in financial fraud detection, analyzing a long transaction sequence is crucial for determining whether the originating account is fraudulent or high-risk.

---

> ### Author Response · Authors · 2025-11-21
>
> # Q3. Do you think the long-range dependency of your model comes from the memory module rather than your design of SSM? Let's say, even if you employ multihop and temporal-aware graph filters, it is still not guaranteed that the model will remember LRT and LRS. Could you provide an analysis to demonstrate that the contribution actually originates from your SSM design? This is very important in determining the quality of the proposed method.
>
> Considering the downstream task as sequence classification, we conduct an ablation study to understand which components of the model capture long-range information. Specifically, we analyze the role of graph filters and the proposed CTDG-SSM module on the long-range spatial ($\texttt{LRS}$) task and the long-range temporal ($\texttt{LRT}$) task.
>
> To evaluate the $\texttt{LRT}$ capabilities of CTDG-SSM, we first set the polynomial $p(\mathbf{L}) = \mathbf{I}$. For an event of the form $(u,v,t,x_u,x_v)$ in sequence classification, instead of restricting the model to update only a small subset of nodes (i.e., those in the batch subgraph), we update the state vectors of all nodes. Formally, we define the input signal at time $t$ as:
>
> $$
> \mathbf{X}(t) \in \mathbb{R}^{N _\tau \times 1} \quad \text{such that} \quad \mathbf{X} [u] (t) = x _u, \mathbf{X} [v] (t) = x _v, \text{and 0 otherwise}.
> $$
>
> This eliminates the step where the previous states of inactive nodes are carried forward through memory. This carry-forward mechanism could aid the model in LRT, so by removing it, we can evaluate the LRT capability of CTDG-SSM in isolation.
> In this setup, the model is tasked with predicting the initial feature $\mathbf{X} [0] (0)$ observed at the first node using the final state vectors of different nodes. A successful LRT would yield strong performance as long as the state vector of node 1 preserves the information of the feature $\mathbf{X} [0] (0)$. Notably, this model completely lacks LRS capability, as it does not account for the underlying graph structure and updates states solely based on the input at the corresponding nodes.
>
> In Table 3  [1st column], we report the mean accuracy using the representations from each node. It can be observed that in the absence of the graph Laplacian and the memory module-based carry-forward, the model retains strong long-range temporal information, as indicated by the high accuracy at the first node. However, the accuracy for predictions originating from other nodes is substantially lower, demonstrating that the model fails to capture long-range spatial dependencies. This setup highlights that while the proposed SSM module alone effectively captures temporal long-range information, it fails to preserve the long-range information spatially.
>
> Next, we evaluate the effect of aggregating multi-hop information by applying graph filters of different orders. and report the mean accuracies obtained from representations at different nodes using a filter of order 1 (Table 3 [2nd column]) and a filter of order 2 (Table 3 [3rd column]). We observe a substantial improvement in prediction accuracy based on the representations from the second node (Table 3 [2nd row]),..., the last node (Table 3 [last row]), demonstrating the model’s enhanced ability to preserve spatial information over longer ranges. In particular, using deeper aggregation-i.e., a filter of order 2-yields a notable gain in accuracy, indicating that incorporating information from larger hop neighborhoods significantly strengthens the model’s capacity to capture
> long-range spatial dependencies.
>
> ---
>
> **Table 3: Ablation study with respect to the order of the graph filter and memory-based carry-forward.**
> | Prediction Node | $p(\mathbf{L}) = \mathbf{I}$ | $p(\mathbf{L}) = \alpha _0 \mathbf{I} + \alpha _1 \mathbf{L}$ | $p(\mathbf{L}) = \alpha _0 \mathbf{I} + \alpha _1 \mathbf{L} + \alpha _2 \mathbf{L}^2$ |
> |-----------------|-------------------------------|-----------------------------------------------|------------------------------------------------------|
> | First           | 1.00 ± 0.00                   | 1.00 ± 0.00                                   | 1.00 ± 0.00                                         |
> | Second          | 0.51 ± 0.06                   | 0.97 ± 0.02                                   | 1.00 ± 0.00                                         |
> | Third           | 0.47 ± 0.02                   | 0.96 ± 0.01                                   | 1.00 ± 0.00                                         |
> | Second-Last     | 0.46 ± 0.01                   | 0.90 ± 0.01                                   | 0.92 ± 0.07                                         |
> | Last            | 0.45 ± 0.02                   | 0.88 ± 0.18                                   | 0.90 ± 0.06                                         |

---

> ### Author Response · Authors · 2025-11-21
>
> # Q4. I saw a model size performance comparison. What about inference time and training time/convergence?
>
> We report the training times per epoch and total runtimes  in Tables 1 and 2.   CTDG-SSM achieves substantially
> lower per-epoch training time and memory usage compared to DyGMamba and DyGFormer, both
> of which are designed for long-range propagation tasks. As explained in the paper, although the update
> involves operations such as matrix inversion, these operators are applied to matrices of very small
> dimensionality, making the computation lightweight in practice.
> For inference, CTDG-SSM retrieves node states and applies a linear layer. For sequence classification, this is $O(1)$. For link prediction and node classification, computing $∆t$ by scanning neighbors adds a $O(deg(u))$ cost before the linear
> layer.  Recall For inference, CTDG-SSM retrieves node states and applies a linear layer. For sequence classification, this is $O(1)$. For link prediction and node classification, this is $O(deg(u))$, due to the computation of ∆t by scanning neighbors, which incurs an additional cost before the linear layer.
>
> The convergence plots are added in the Appendix of  the revised manuscript.
>
> **Table 1: Comparison of model per-epoch time (minutes) and GPU memory usage (GB) across multiple datasets**
> | Models      | LastFM (Time / Mem) | Enron (Time / Mem) | MOOC (Time / Mem) | UCI (Time / Mem) | Reddit (Time / Mem) | Social Evo. (Time / Mem) |
> |------------|-------------------|------------------|-----------------|----------------|-------------------|--------------------------|
> | JODIE      | 4.4 / 2.28        | 0.07 / 1.30      | 0.78 / 2.36     | 0.03 / 1.44    | 3.95 / 1.10       | 4.70 / 1.71             |
> | DyRep      | 6.6 / 2.29        | 0.10 / 1.34      | 0.88 / 2.38     | 0.05 / 1.51    | 5.75 / 1.21       | 7.55 / 1.76             |
> | TGAT       | 22.75 / 4.15      | 1.28 / 3.46      | 4.08 / 3.64     | 0.60 / 3.42    | 16.33 / 2.98      | 25.50 / 3.89            |
> | TGN        | 12.14 / 2.21      | 0.15 / 1.45      | 1.03 / 2.54     | 0.08 / 1.51    | 2.05 / 1.67       | 3.83 / 1.78             |
> | CAWN       | 99.00 / 14.92     | 2.62 / 4.03      | 13.45 / 8.02    | 1.95 / 9.40    | 20.16 / 5.89      | 85.66 / 8.14            |
> | TCL        | 6.23 / 3.04       | 0.30 / 2.51      | 1.00 / 2.49     | 0.13 / 2.00    | 2.25 / 1.82       | 5.05 / 2.48             |
> | GraphMixer | 16.35 / 2.78      | 1.20 / 2.23      | 4.02 / 2.40     | 0.73 / 2.19    | 4.92 / 1.57       | 15.50 / 2.71            |
> | DyGFormer  | 47.00 / 7.57      | 2.73 / 3.23      | 8.32 / 3.35     | 0.62 / 2.30    | 7.00 / 2.42       | 20.00 / 2.77            |
> | CTAN       | 3.33 / 1.44       | 0.50 / 1.33      | 3.22 / 2.30     | 0.38 / 1.30    | 0.86 / 1.54       | 2.41 / 0.63             |
> | DyGMamba   | 28.45 / 4.17      | 2.05 / 2.74      | 4.88 / 2.48     | 0.60 / 1.93    | 6.30 / 2.07       | 17.80 / 2.59            |
> | CTDG-SSM   | 4.45 / 1.15       | 0.55 / 0.86      | 1.25 / 0.43     | 0.17 / 0.31   | 1.95 / 1.18       | 9.57 / 5.2            |
>
> ---
> ---
>
>
>
> **Table-2: Training epochs and total time (minutes) across datasets**
>
> | Models      | Enron (#Epoch / T_tot) | UCI (#Epoch / T_tot) | Reddit (#Epoch / T_tot) |
> |------------|-----------------------|---------------------|------------------------|
> | CTAN       | 173.00 / 86.50        | 236.00 / 89.68      | 327.18 / 173.41        |
> | DyGFormer  | 32.80 / 89.54         | 34.80 / 21.58       | 24.60 / 104.30         |
> | DyGMamba   | 33.00 / 67.65         | 28.00 / 16.80       | 26.80 / 88.98          |
> | CTDG-SSM   | 83.00 / **45.65**     | 38.00 / **6.46**    | 27.00 / **52.65**      |

---

> ### Author Response · Authors · 2025-11-26
> **Awaiting for feedback on the rebuttal**
>
> Dear Reviewer Dwin,
>
> We kindly request you to review our detailed rebuttal, which thoroughly addresses the concerns you raised.
>
> As requested, the rebuttal provides clear clarifications regarding the novelty of our approach, computational complexity. In addition we have a provided a detailed ablation study that clarifies the importance of higher order graph filters and the CTDG-SSM module in capturing the LRS and LRT tasks.   The current version is supplemented with extensive additional experiments and clarifies the questions raised with more evidences.
>
> If our rebuttal satisfactorily resolves your concerns, we respectfully request you to consider raising the score.

---

> ### Comment · Reviewer_Dwin · 2025-11-27
> **Further Response**
>
> I like how the authors use additional results as backbone for the rebuttal. Most concerns are resolved. Although I am still slightly concerned about the so-called "novelty", I think providing strong theoretical understanding is also a strong contribution. I will raise my score to 6.
>
> I recommend the authors update their PDF to reflect the efforts spent in rebuttal. This is for other reviewers to better understand this work.
>
> Thanks a lot.
>
> PS: It seems the system does not allow me to change score right now. I suppose it is due to the issue happening today. I would do that after the system becomes normal again. Don't worry.

---

> > ### Author Response · Authors · 2025-11-28
> >
> > Dear Reviewer Dwin,
> >
> > We sincerely thank you for the positive feedback on the response and for the score update. We have incorporated all necessary changes into the revised manuscript.

---

### Official Review · Reviewer_qcoL · 2025-11-02

**Soundness:** 2
**Presentation:** 2
**Contribution:** 3
**Rating:** 4
**Confidence:** 4

**Summary:**

The paper proposes CTDG-SSM, a state-space model formulation for continuous-time dynamic graphs (CTDGs). The method builds on HiPPO to derive topology-aware memory representations (CTT-HiPPO), where a polynomial of the graph Laplacian is used to incorporate structural information. A continuous-time formulation is discretized via zero-order hold and evaluated on link prediction, node classification, and a sequence classification benchmark. Results indicate strong performance using a modest number of parameters.

**Strengths:**

* Addresses a relevant problem: maintaining both long-range temporal and long-range spatial dependencies in CTDGs.

* Empirical performance appears strong on benchmarks requiring long-range propagation.

* Architecture is lightweight in parameter size compared to SOTA CTDG methods.

**Weaknesses:**

* **Novelty**. I find novelty relatively limited. Like GraphSSM, CTDG-SSM extends HiPPO to temporal graphs — however the former cannot be directly applied to CTDGs.

* **Imprecise account of prior literature**. Authors provide an imprecise account of prior literature. For instance, the categorization of methods around line 49 is not faithful to CAW — this method does not even have a notion of explicit node embeddings. Regarding CTDGs vs. DTDGs, it is possible to draw equivalences between both — c.f., Prop. 1. of [1].

* **Matrix inverses and numerical stability**. It is not clear to me why $p(L_{\tau})^{-1}$ should exist. The repeated use of matrix inverses also makes me wonder if there is some numerical stability worth disclosing — and how it affects runtime. A brief discussion of how invertibility is ensured, and whether any numerical stability considerations are necessary in practice, would improve clarity.

* **Permutation equivariance**. It is not clear to me that the proposed architecture really is permutation equivariant. It seems Theorem 6.2 ignores the stochasticity in subgraph sampling, which breaks exact permutation equivariance. Clarifying how this affects the theoretical property would be helpful.

* **Efficiency claims**. Efficiency claims are not convincingly supported: only parameter counts are compared, while no runtime, memory-usage, or throughput experiments are provided. Given the use of Laplacian-polynomial inverses and evolving graph operators, the practical computational cost is unclear. Reporting wall-clock training time, inference latency, or events-per-second versus DyGmamba and DyGFormer would make the efficiency argument more credible.


[1] Provably expressive temporal graph networks, NeurIPS 2022

**Questions:**

My questions are directly aligned with the points raised in the weaknesses above.

---

> ### Author Response · Authors · 2025-11-21
> **Response to Reviewer qcoL**
>
> We thank the Reviewer for his time and for the feedback. We provide point  to point response for the comments.
>
>
> # **W1.** Novelty. I find novelty relatively limited. Like GraphSSM, CTDG-SSM extends HiPPO to temporal graphs — however the former cannot be directly applied to CTDGs.
>
> The fact that GraphSSM cannot be applied to CTDGs makes the proposed model and development novel. As such, the proposed extension is not straightforward. Below, we summarize the key challenges involved in adapting GraphSSM to the continuous-time setting:
>
> -  **Deriving a State-Space Update That Accounts for Event-Driven Changes in Graph Topology**
>   This is crucial because the CTDGs model fine-grained temporal evolution, where interactions occur at arbitrary, irregular time intervals. Extending the discrete-time update used in GraphSSM to a continuous-time, event-driven form requires a non-trivial modification of the underlying SSM dynamics, which must account for structural changes as captured by  $\frac{dp(\mathbf{L}_s)}{ds}$ term in the proposed state space equation. We further emphasize that obtaining such a recursive update requires a new framework that models time-varying node features as *joint time-vertex signals*, as clearly explained in the paper.
> - **Algorithm that can scale to large-scale datasets with significantly lower runtime**:
>   The proposed algorithm, developed to update the states of active nodes in sampled events, is novel, and this stands in contrast to existing representation-learning architectures for CTDGs, where updates are typically fully event-driven but not restricted to specific participating nodes. More importantly, a computationally tractable discrete state-space update is novel and the first of its kind in SSM-based frameworks.
> - **Theoretical Analysis**:
>   To the best of our knowledge, this is the first work to provide a theoretical analysis establishing robustness and permutation equivariance properties. Their inclusion requires deriving the continuous-time update rules from first principles.
>
> # **W2.** Imprecise account of prior literature. The authors provide an imprecise account of prior literature. For instance, the categorization of methods around line 49 is not faithful to CAW — this method does not even have a notion of explicit node embeddings. Regarding CTDGs vs. DTDGs, it is possible to draw equivalences between both — c.f., Prop. 1. of [1].
> We thank the Reviewer for pointing this out. We will revise this statement to ensure coherence with the established categorization in the related works section. In particular, we have revised the statement as follows:
>
> > *"Event-driven models update node states at the arrival of each interaction and capture structural context through mechanisms such as temporal random walks and GNN-based message passing (Wang et al., 2021; Rossi et al., 2020; Xu et al., 2020)."*
>
> Regarding the equivalence between CTDGs and DTDGs, in the related works section, we will explicitly mention that the proposed method, primarily developed for CTDGs, can also handle DTDGs, given the equivalence between CTDGs and DTDGs (Souza et al., 2022).

---

> ### Author Response · Authors · 2025-11-21
>
> # **W3.** Matrix inverses and numerical stability. It is not clear to me why $p(\mathbf{L}_{\tau})^{-1}$ should exist. The repeated use of matrix inverses also makes me wonder if there is some numerical instability worth disclosing — and how it affects runtime. A brief discussion on how invertibility is ensured and whether any numerical stability considerations are necessary in practice would improve clarity.
>
> To begin with, recall that the normalized graph Laplacian is defined as
> $$
> \mathbf{L} _\tau = \mathbf{I} - \mathbf{D} _{\tau}^{-1/2}\mathbf{A} _{\tau}\mathbf{D} _{\tau}^{-1/2},
> $$
>
> whose spectrum satisfies $\lambda_{\tau} \in [0,2]$.
>
> As discussed, we adopt a finite impulse response (FIR) graph filter of the form
>
> $$
> p(\mathbf{L} _{\tau}) = \sum _{k=0} ^{K-1} \alpha _k \mathbf{L} _{\tau} ^{k}
> = \alpha _0 \mathbf{I} + \alpha _1 \mathbf{L} _{\tau} + \dots + \alpha _{K-1} \mathbf{L} _{\tau} ^{K-1}.
> $$
>
> Choosing $\alpha_{i} > 0 \,\, \forall i = 0,1,\ldots,K-1$ is sufficient to ensure that  $p(\lambda) > 0$ for all $\lambda \in [0,2]$, which implies that $p(\mathbf{L}_{\tau})$ is positive definite and therefore invertible.
>
> It is also important to note that the inverse operation $p( \mathbf{L} _\tau) ^{-1}$  is not computationally expensive in practice. As described in the architecture section, our model updates only the states of the active nodes within each batch. The corresponding graph Laplacian $\mathbf{L} _{B}$ is constructed over the node set $\mathcal{V} _{B}$, which consists of
> (i) nodes appearing in the current and previous batches, and
> (ii) their sampled neighbors.
>
> If $N _{B} = |\mathcal{V} _{B}|$ denotes the size of this active set, then computing the inverse $p(\mathbf{L} _{B}[k]) ^{-1}$ incurs a cost of $\mathcal{O}(N _{B} ^{3})$.
>
> Crucially, $N_{B} \ll N$, where $N$ is the total number of nodes in the graph. Thus, the inversion is performed on a much smaller matrix than the full-graph Laplacian, significantly reducing computational overhead. Moreover, $N_{B}$ is controlled by the batch size and the sampling strategy, and therefore can be treated as a tunable hyperparameter.
>
> Empirically, this design yields efficient training, as evidenced by the runtimes and per-epoch training times reported in Tables 1 and 2.
>
> #  **W4.**  It is not clear to me that the proposed architecture really is permutation equivariant. It seems Theorem 6.2 ignores the stochasticity in subgraph sampling, which breaks exact permutation equivariance. Clarifying how this affects the theoretical property would be helpful.
>
> We clarify that Theorem 6.2 establishes permutation equivariance of the $\texttt{CTDG-SSM}$ representations with respect to node relabeling, and this property is distinct from the sampling procedure. Sampling is used solely to select a subset of nodes and construct the corresponding subgraph. Once this subgraph is fixed, the associated Laplacian and features are also fixed. Theorem 6.2 applies to this stage: if we apply any permutation to the nodes of this fixed Laplacian and its features, the output representations from $\texttt{CTDG-SSM}$ permute in exactly the same way-thus satisfying permutation equivariance.
>
> # **W5.** Efficiency claims. Efficiency claims are not convincingly supported: only parameter counts are compared, while no runtime, memory-usage, or throughput experiments are provided. Given the use of Laplacian-polynomial inverses and evolving graph operators, the practical computational cost is unclear. Reporting wall-clock training time, inference latency, or events-per-second versus DyGMamba and DyGFormer would make the efficiency argument more credible.
>
> We thank the Reviewer for the comment. For the dynamic link prediction task, we now report per-
> epoch training time (in minutes) and GPU memory usage (in GB) in Table 1, and the total training
> time = per epoch training time × #Epochs in Table 2. Notably, CTDG-SSM achieves substantially
> lower per-epoch training time and memory usage compared to DyGMamba and DyGFormer, both
> of which are designed for long-range propagation tasks. As explained earlier, although the update
> involves operations such as matrix inversion, these operators are applied to matrices of very small
> dimensionality, making the computation lightweight in practice.
> For inference, CTDG-SSM retrieves node states and applies a linear layer. For sequence classification, this is $O(1)$. For link prediction and node classification, computing $∆t$ by scanning neighbors adds a $O(deg(u))$ cost before the linear
> layer.

---

> ### Author Response · Authors · 2025-11-21
>
> **Table 1: Comparison of model per-epoch time (minutes) and GPU memory usage (GB) across multiple datasets**
>
> | Models      | LastFM (Time / Mem) | Enron (Time / Mem) | MOOC (Time / Mem) | UCI (Time / Mem) | Reddit (Time / Mem) | Social Evo. (Time / Mem) |
> |------------|-------------------|------------------|-----------------|----------------|-------------------|--------------------------|
> | JODIE      | 4.4 / 2.28        | 0.07 / 1.30      | 0.78 / 2.36     | 0.03 / 1.44    | 3.95 / 1.10       | 4.70 / 1.71             |
> | DyRep      | 6.6 / 2.29        | 0.10 / 1.34      | 0.88 / 2.38     | 0.05 / 1.51    | 5.75 / 1.21       | 7.55 / 1.76             |
> | TGAT       | 22.75 / 4.15      | 1.28 / 3.46      | 4.08 / 3.64     | 0.60 / 3.42    | 16.33 / 2.98      | 25.50 / 3.89            |
> | TGN        | 12.14 / 2.21      | 0.15 / 1.45      | 1.03 / 2.54     | 0.08 / 1.51    | 2.05 / 1.67       | 3.83 / 1.78             |
> | CAWN       | 99.00 / 14.92     | 2.62 / 4.03      | 13.45 / 8.02    | 1.95 / 9.40    | 20.16 / 5.89      | 85.66 / 8.14            |
> | TCL        | 6.23 / 3.04       | 0.30 / 2.51      | 1.00 / 2.49     | 0.13 / 2.00    | 2.25 / 1.82       | 5.05 / 2.48             |
> | GraphMixer | 16.35 / 2.78      | 1.20 / 2.23      | 4.02 / 2.40     | 0.73 / 2.19    | 4.92 / 1.57       | 15.50 / 2.71            |
> | DyGFormer  | 47.00 / 7.57      | 2.73 / 3.23      | 8.32 / 3.35     | 0.62 / 2.30    | 7.00 / 2.42       | 20.00 / 2.77            |
> | CTAN       | 3.33 / 1.44       | 0.50 / 1.33      | 3.22 / 2.30     | 0.38 / 1.30    | 0.86 / 1.54       | 2.41 / 0.63             |
> | DyGMamba   | 28.45 / 4.17      | 2.05 / 2.74      | 4.88 / 2.48     | 0.60 / 1.93    | 6.30 / 2.07       | 17.80 / 2.59            |
> | CTDG-SSM   | 4.45 / 1.15       | 0.55 / 0.86      | 1.25 / 0.43     | 0.17 / 0.31   | 1.95 / 1.18       | 9.57 / 5.2            |
>
> ---
> ---
> **Table-2 Training epochs and total time (minutes) across datasets**
>
> | Models      | Enron (#Epoch / T_tot) | UCI (#Epoch / T_tot) | Reddit (#Epoch / T_tot) |
> |------------|-----------------------|---------------------|------------------------|
> | CTAN       | 173.00 / 86.50        | 236.00 / 89.68      | 327.18 / 173.41        |
> | DyGFormer  | 32.80 / 89.54         | 34.80 / 21.58       | 24.60 / 104.30         |
> | DyGMamba   | 33.00 / 67.65         | 28.00 / 16.80       | 26.80 / 88.98          |
> | CTDG-SSM   | 83.00 / **45.65**     | 38.00 / **6.46**    | 27.00 / **52.65**      |

---

> > ### Author Response · Authors · 2025-11-26
> > **Awaiting for feedback on the rebuttal**
> >
> > Dear Reviewer qcoL,
> >
> > We kindly request you to review our detailed rebuttal, which thoroughly addresses the concerns you raised.
> >
> > As requested, the rebuttal provides clear clarifications regarding both the novelty of our approach and its computational complexity. Although the questions primarily sought clarifications, we have supplemented our responses with extensive additional experiments. These results further demonstrate the strength of our proposed algorithm in capturing long-range spatial and temporal dependencies while maintaining significantly lower runtimes.
> >
> > If our rebuttal satisfactorily resolves your concerns, we respectfully request you to consider raising the score.

---

### Author Response · Authors · 2025-12-01
**Summary of Rebuttal**

Dear AC,

We would like to provide a summary of the key points addressed in our rebuttal to CTDG-SSM:

1. **Clarifications:** The majority of the reviewer concerns were clarification-related. We have addressed these issues in our responses and incorporated the necessary clarifications directly into the main paper.

2. **Ablation Studies & Robustness:** We added an ablation study examining the importance of $p(\textbf{L})$  in LRS and evaluated the model’s robustness under graph structure perturbations.

3. **Theoretical Contributions:** We highlighted the principal theoretical development of the SSM for CTDG, presenting an updated SSM formulation that explicitly incorporates dynamic topology through $\frac{dp(\textbf{L})}{dt}$ term. This extension is unique to our work and directly addresses reviewers’ concerns regarding novelty.
4. **Additional Experiments:** As requested, we performed experiments on additional datasets, including those from TGBL, and incorporated more baseline comparisons.

5. **Complexity & Sub-Graph Sampling:** We have clarified the computational complexity and the sub-graph sampling procedure used in our method.

These clarifications, along with the additional ablation studies and experiments, strengthen the claims of our proposed approach. Notably, one reviewer explicitly suggested that these changes enhanced the paper’s contribution before the discussion phase was suspended.

We hope these updates will be helpful in your evaluation.

---

### Note · Authors · 2026-01-26

I have read and agree with the venue's withdrawal policy on behalf of myself and my co-authors.

---

### Meta-Review · Area_Chair_gG1y · 2025-12-29

**Summary:**

The main strengths identified by the reviewers are:
* Timeliness and relevance of the problem addresses: Long-range temporal and long-range spatial propagation in CTDGs is an important and timely topic, bringing consolidated concepts from static graph learning to the domain of dynamic graph models.
* Theoretical grounding: the continuous-time formulation is principled, with formal guarantees on robustness and permutation equivariance.
* CTT-HiPPO operator: this is conceptually appealing, offering a clean way to couple temporal memory with multi-hop structural context.
* Empirical performance: the experimental analysis provides good evidence for the model achieving strong performance across dynamic link prediction, node classification, and sequence classification.

On the side of the weaknesses, the most persistent issue is novelty. Multiple reviewers felt the work represents an incremental extension of existing SSM-based or HiPPO-inspired models (e.g., GraphSSM, DyGMamba). References were also given (by Reviewer Ks8e) to similar works extending HiPPO to continuous time (JinTang Li et al, NeurIPS 2024).

Early versions suffered from unclear positioning and imprecise related-work discussion, which led to confusion about what aspects are fundamentally new versus adapted. Reviewer Ks8e also highilighted the lack of empirical comparison with highly related models and relevant real world datasets.

There were also repeated and shared concerns about computational complexity, particularly around Laplacian polynomial inverses and matrix operations, which required extensive clarification and additional experiments to justify scalability.

Finally, Reviewer Ks8e highlighted a potential issue with data leakage on MOOC, with specific reference to the submitted code.

**Reviewer Concerns:**

The rebuttal successfully addressed several technical and experimental concerns raised by reviewers.

First, concerns on computational complexity and scalability were resolved by clarifying that Laplacian polynomial inverses are computed only on small, batch-induced subgraphs. Also clear experimental evidence as regards per-epoch runtime, total training time, and GPU memory comparisons against related models resolved the issue.

Questions regarding numerical stability and invertibility were addressed with clear considerations abour why the Laplacial operator is positive definite and stable in practice.

The issue of permutation equivariance under subgraph sampling was clarified by separating the sampling stage from the equivariant update, with a precise statement of the scope of the theoretical guarantee.

Concerns about missing datasets and selective benchmarking were mitigated by adding experiments in recent long-range CTDG baselines and additional benchmarks.

On the side of concerns that are still outstanding, the most relevant one is novelty, which remains debated. The Authors responded to all concerns by restating that the adaptation to continuous time is non-trivial. The Authors rebuttal on this does not really articulate on which aspects are exactly non trivial and, most importantly, does not provide a clear comparison on the novelty aspects against prior SSM/HiPPO-based models. A clear example of this is the lack of consideration of Reviewer Ks8e concern about the similarities with JinTang Li et al. NeurIPS 2024.

An additional point is the method’s reliance on fixed polynomial graph filters (rather than learned operators). This limits expressiveness, which is an issue acknowledged by the Authors but not resolved.

**Reviewer Scores:**

Among all reviewers only Dwin responded to the rebuttal: their position is that while novelty aspects remain not entirely convincing, the additional theoretical considerations and experimental analyses strenghtened the work. As a result they proposed a score raise to 6.

Reviewer qcoL initially proposed a score of 4 (with confidence 4). Most of the concerns highlighted received a solid response, with the exception of novelty aspects. One could expect a similar behaviour to Dwin with a score being raised to 6.

Reviewer Ks8e was the most confident (5) with a deep review and a score of 4. The rebuttal did not adequately shed light on specific questions related to novelty and data leakage on MOOC. Here one could expect the reviewer to keep the original score.

Reviewer 2WD4 had an articulated list of points on no particular technical depth, consistent with a medium confidence (3) and a score of 4. One could have expected a limited involvement of this reviewer to the discussion, possibly with no change in score.

Overall, one could expect that the paper at the end of the rebuttal would have scored between 5 (with more likelihood) and 5.5. Assuming a decision threshold around 5.5 this would have been a borderline paper, leaning on the negative end due to the unresolved issues about novelty.

---

### Decision · Program_Chairs · 2026-01-26

Reject